# Global Carbon Budget 2018

Corinne Le Quéré[1], Robbie M. Andrew[2], Pierre Friedlingstein[3], Stephen Sitch[4], Judith Hauck[5], Julia Pongratz[6,7], Penelope Pickers[8], Jan Ivar Korsbakken[2], Glen P. Peters[2], Josep G. Canadell[9], Almut Arneth[10], Vivek K. Arora[11], Leticia Barbero[12,13], Ana Bastos[6], Laurent Bopp[14], Frédéric Chevallier[15], Louise P. Chini[16], Philippe Ciais[15], Scott C. Doney[17], Thanos Gkritzalis[18], Daniel S. Goll[15], Ian Harris[19], Vanessa Haverd[20], Forrest M. Hoffman[21], Mario Hoppema[5], Richard A. Houghton[22], Tatiana Ilyina[7], Atul K. Jain[23], Truls Johannessen[24], Chris D. Jones[25], Etsushi Kato[26], Ralph F. Keeling[27], Kees Klein Goldewijk[28,29], Peter Landschützer[7], Nathalie Lefèvre[30], Sebastian Lienert[31], Danica Lombardozzi[32], Nicolas Metzl[30], David R. Munro[33], Julia E. M. S. Nabel[7], Shin-ichiro Nakaoka[34], Craig Neill[35,36], Are Olsen[37], Tsueno Ono[38], Prabir Patra[39], Anna Peregon[15], Wouter Peters[40,41], Philippe Peylin[15], Benjamin Pfeil[24,37] , Denis Pierrot[12,13], Benjamin Poulter[42] , Gregor Rehder[43], Laure Resplandy[44], Eddy Robertson[25], Matthias Rocher[45], Christian Rödenbeck[46], Ute Schuster[4], Jörg Schwinger[37], Roland Séférian[45], Ingunn Skjelvan[37], Tobias Steinhoff[47], Adrienne Sutton[48], Pieter P. Tans[49], Hanqin Tian[50], Bronte Tilbrook[35,36], Francesco N Tubiello[51], Ingrid T. van der Laan-Luijkx[40], Guido R. van der Werf[52], Nicolas Viovy[15], Anthony P. Walker[53], Andrew J. Wiltshire[25], Rebecca Wright[8], Sönke Zaehle[45]

[1]Tyndall Centre for Climate Change Research, University of East Anglia, Norwich Research Park, Norwich NR4 7TJ, UK

[2]CICERO Center for International Climate Research, Oslo 0349, Norway

[3]College of Engineering, Mathematics and Physical Sciences, University of Exeter, Exeter EX4 4QF, UK

[4]College of Life and Environmental Sciences, University of Exeter, Exeter EX4 4RJ, UK

[5]Alfred Wegener Institute Helmholtz Centre for Polar and Marine Research, Postfach 120161, 27515 Bremerhaven, Germany

[6]Ludwig-Maximilans-Universität Munich, Luisenstr. 37, 80333 München, Germany

[7]Max Planck Institute for Meteorology, Hamburg, Germany

[8]Centre for Ocean and Atmospheric Sciences, School of Environmental Sciences, University of East Anglia, Norwich Research Park, Norwich NR4 7TJ, UK

[9]Global Carbon Project, CSIRO Oceans and Atmosphere, GPO Box 1700, Canberra, ACT 2601, Australia

[10]Karlsruhe Institute of Technology (KIT, IMK-IFU)

[11]Canadian Centre for Climate Modelling and Analysis, Climate Research Division, Environment and Climate Change Canada, Victoria, BC, Canada

[12]Cooperative Institute for Marine and Atmospheric Studies, Rosenstiel School for Marine and Atmospheric Science, University of Miami, Miami, FL 33149, USA

[13]National Oceanic & Atmospheric Administration/Atlantic Oceanographic & Meteorological Laboratory (NOAA/AOML), Miami, FL 33149, USA

[14]Laboratoire de Météorologie Dynamique, Institut Pierre-Simon Laplace, CNRS-ENS-UPMC-X, Département de Géosciences, Ecole Normale Supérieure, 24 rue Lhomond, 75005 Paris, France

[15]Laboratoire des Sciences du Climat et de l'Environnement, Institut Pierre-Simon Laplace, CEA-CNRS-UVSQ, CE Orme des Merisiers, 91191 Gif sur Yvette Cedex, France

[16]Department of Geographical Sciences, University of Maryland, College Park, Maryland 20742, USA

[17]University of Virginia, Charlottesville, VA 22904, USA

[18]Flanders Marine Institute (VLIZ), Wanelaarkaai 7, 8400 Ostend, Belgium

[19]NCAS-Climate, Climatic Research Unit, School of Environmental Sciences, University of East Anglia, Norwich Research Park, Norwich, NR4 7TJ, UK

[20]CSIRO Oceans and Atmosphere, GPO Box 1700, Canberra, ACT 2601, Australia

[21]Computational Earth Sciences Group, Oak Ridge National Laboratory, Oak Ridge, Tennessee, USA

[22]Woods Hole Research Centre (WHRC), Falmouth, MA 02540, USA

[23]Department of Atmospheric Sciences, University of Illinois, Urbana, IL 61821, USA

[24]Geophysical Institute, University of Bergen and Bjerknes Centre for Climate Research, Allégaten 70, 5007 Bergen, Norway

[25]Met Office Hadley Centre, FitzRoy Road, Exeter EX1 3PB, UK

[26]Institute of Applied Energy (IAE), Minato-ku, Tokyo 105-0003, Japan

[27]University of California, San Diego, Scripps Institution of Oceanography, La Jolla, CA 92093-0244, USA

[28]PBL Netherlands Environmental Assessment Agency, Bezuidenhoutseweg 30, P.O. Box 30314, 2500 GH, The Hague, the Netherlands

[29]Faculty of Geosciences, Department IMEW, Copernicus Institute of Sustainable Development, Heidelberglaan 2, P.O. Box 80115, 3508 TC, Utrecht, the Netherlands

[30]Sorbonne Universités (UPMC, Univ Paris 06), CNRS, IRD, MNHN, LOCEAN/IPSL Laboratory, 75252 Paris, France

[31]Climate and Environmental Physics, Physics Institute and Oeschger Centre for Climate Change Research, University of Bern, Bern, Switzerland

[32]National Center for Atmospheric Research, Climate and Global Dynamics, Terrestrial Sciences Section, Boulder, CO 80305, USA

[33]Department of Atmospheric and Oceanic Sciences and Institute of Arctic and Alpine Research, University of Colorado, Campus Box 450, Boulder, CO 80309-0450, USA

[34]Center for Global Environmental Research, National Institute for Environmental Studies (NIES), 16-2 Onogawa, Tsukuba, Ibaraki 305-8506, Japan

[35]CSIRO Oceans and Atmosphere, PO Box 1538, Hobart, Tasmania, 7001, Australia

[36]Antarctic Climate and Ecosystem Cooperative Research Centre, University of Tasmania, Hobart, Australia

[37]NORCE Norwegian Research Centre and Bjerknes Centre for Climate Research, Jahnebakken 5, 5007 Bergen, Norway

[38]National Research Institute for Far Sea Fisheries, Japan Fisheries Research and Education Agency, 2-12-4 Fukuura, Kanazawa-Ku, Yokohama 236-8648, Japan

[39]Department of Environmental Geochemical Cycle Research, JAMSTEC, Yokohama, Japan

[40]Department of Meteorology and Air Quality, Wageningen University & Research, PO Box 47, 6700AA Wageningen, The Netherlands

[41]Centre for Isotope Research, University of Groningen, Nijenborgh 6, 9747 AG Groningen, the Netherlands

[42]NASA Goddard Space Flight Center, Biospheric Science Laboratory, Greenbelt, Maryland 20771, USA

[43]Leibniz Institute for Baltic Sea Research Warnemünde, 18119 Rostock, Germany

[44]Princeton University Department of Geosciences and Princeton Environmental Institute Princeton, New Jersey, USA

[45]Centre National de Recherche Météorologique, Unite mixte de recherche 3589 Météo-France/CNRS, 42 Avenue Gaspard Coriolis, 31100 Toulouse, France

[46]Max Planck Institute for Biogeochemistry, P.O. Box 600164, Hans-Knöll-Str. 10, 07745 Jena, Germany

[47]GEOMAR Helmholtz Centre for Ocean Research Kiel, Düsternbrooker Weg 20, 24105, Kiel, Germany

[48]National Oceanic & Atmospheric Administration/Pacific Marine Environmental Laboratory (NOAA/PMEL), 7600 Sand Point Way NE, Seattle, WA 98115, USA

[49]National Oceanic & Atmospheric Administration, Earth System Research Laboratory (NOAA/ESRL), Boulder, CO 80305, USA

[50]School of Forestry and Wildlife Sciences, Auburn University, 602 Ducan Drive, Auburn, AL 36849, USA

[51]Statistics Division, Food and Agriculture Organization of the United Nations, Via Terme di Caracalla, Rome 00153, Italy

[52]Faculty of Science, Vrije Universiteit, Amsterdam, The Netherlands

[53]Environmental Sciences Division & Climate Change Science Institute, Oak Ridge National Laboratory, Oak Ridge, Tennessee, USA

**Abstract**
Accurate assessment of anthropogenic carbon dioxide ($CO_2$) emissions and their redistribution
among the atmosphere, ocean, and terrestrial biosphere – the 'global carbon budget' – is
important to better understand the global carbon cycle, support the development of climate
policies, and project future climate change. Here we describe data sets and methodology to
quantify the five major components of the global carbon budget and their uncertainties. Fossil
$CO_2$ emissions ($E_{FF}$) are based on energy statistics and cement production data, while emissions
from land-use change ($E_{LUC}$), mainly deforestation, are based on land-use and land-use change
data and bookkeeping models. Atmospheric $CO_2$ concentration is measured directly and its
growth rate ($G_{ATM}$) is computed from the annual changes in concentration. The ocean $CO_2$ sink
($S_{OCEAN}$) and terrestrial $CO_2$ sink ($S_{LAND}$) are estimated with global process models constrained by
observations. The resulting carbon budget imbalance ($B_{IM}$), the difference between the estimated
total emissions and the estimated changes in the atmosphere, ocean, and terrestrial biosphere, is
a measure of imperfect data and understanding of the contemporary carbon cycle.  All
uncertainties are reported as ±1σ. For the last decade available (2008-2017), $E_{FF}$ was 9.4 ± 0.5 GtC
$yr^{-1}$, $E_{LUC}$ 1.5 ± 0.7 GtC $yr^{-1}$, $G_{ATM}$ 4.7 ± 0.02 GtC $yr^{-1}$, $S_{OCEAN}$ 2.4 ± 0.5 GtC $yr^{-1}$, and $S_{LAND}$ 3.2 ± 0.8 GtC
$yr^{-1}$, with a budget imbalance $B_{IM}$ of 0.5 GtC $yr^{-1}$ indicating overestimated emissions and/or
underestimated sinks. For year 2017 alone, the growth in $E_{FF}$ was about 1.6% and emissions
increased to 9.9 ± 0.5 GtC $yr^{-1}$. Also for 2017, $E_{LUC}$ was 1.4 ± 0.7 GtC $yr^{-1}$, $G_{ATM}$ was 4.6 ± 0.2 GtC $yr^{-1}$
, $S_{OCEAN}$ was 2.5 ± 0.5 GtC $yr^{-1}$ and $S_{LAND}$ was 3.8 ± 0.8 GtC $yr^{-1}$, with a small $B_{IM}$ of 0.3 GtC. The
global atmospheric $CO_2$ concentration reached 405.0 ± 0.1 ppm averaged over 2017. For 2018,
preliminary data for the first 6-9 months indicate a renewed growth in $E_{FF}$ of +2.7% (range of 1.8%
to 3.7%) based on national emissions projections for China, USA, the EU and India, and projections
of Gross Domestic Product corrected for recent changes in the carbon intensity of the economy
for the rest of the world. The analysis presented here shows that the mean and trend in the five
components of the global carbon budget are consistently estimated over the period 1959-2017,
but discrepancies of up to 1 GtC $yr^{-1}$ persist for the representation of semi-decadal variability in
$CO_2$ fluxes. A detailed comparison among individual estimates and the introduction of a broad
range of observations shows: (1) no consensus in the mean and trend in land-use change
emissions, (2) a persistent low agreement between the different methods on the magnitude of
the land $CO_2$ flux in the northern extra-tropics, and (3) an apparent underestimation of the $CO_2$
variability by ocean models, originating outside the tropics. This living data update documents
changes in the methods and data sets used in this new global carbon budget and the progress in
understanding of the global carbon cycle compared with previous publications of this data set (Le
Quéré et al., 2018, 2016; 2015b; 2015a; 2014; 2013). All results presented here can be
downloaded from https://doi.org/10.18160/GCP-2018.
**1    Introduction**
The concentration of carbon dioxide ($CO_2$) in the atmosphere has increased from approximately
277 parts per million (ppm) in 1750 (Joos and Spahni, 2008), the beginning of the Industrial Era, to
405.0 ± 0.1  ppm in 2017 (Dlugokencky and Tans, 2018; Fig. 1). The atmospheric $CO_2$ increase
above preindustrial levels was, initially, primarily caused by the release of carbon to the
atmosphere from deforestation and other land-use change activities (Ciais et al., 2013). While
emissions from fossil fuels started before the Industrial Era, they only became the dominant
source of anthropogenic emissions to the atmosphere from around 1950 and their relative share
has continued to increase until present. Anthropogenic emissions occur on top of an active natural
carbon cycle that circulates carbon between the reservoirs of the atmosphere, ocean, and
terrestrial biosphere on time scales from sub-daily to millennia, while exchanges with geologic
reservoirs occur at longer timescales (Archer et al., 2009).
The global carbon budget presented here refers to the mean, variations, and trends in the
perturbation of $CO_2$ in the environment, referenced to the beginning of the Industrial Era. It
quantifies the input of $CO_2$ to the atmosphere by emissions from human activities, the growth rate
of atmospheric $CO_2$ concentration, and the resulting changes in the storage of carbon in the land
and ocean reservoirs in response to increasing atmospheric $CO_2$ levels, climate change and
variability, and other anthropogenic and natural changes (Fig. 2). An understanding of this
perturbation budget over time and the underlying variability and trends of the natural carbon
cycle are necessary to understand the response of natural sinks to changes in climate, $CO_2$ and
land-use change drivers, and the permissible emissions for a given climate stabilization target.
The components of the $CO_2$ budget that are reported annually in this paper include separate
estimates for the $CO_2$ emissions from (1) fossil fuel combustion and oxidation from all energy and
industrial processes and cement production ($E_{FF}$; GtC yr$^{-1}$) and (2) the emissions resulting from
deliberate human activities on land, including those leading to land-use change ($E_{LUC}$; GtC yr$^{-1}$);
and their partitioning among (3) the growth rate of atmospheric $CO_2$ concentration ($G_{ATM}$; GtC yr$^{-}$

[1]), and the uptake of $CO_2$ (the 'CO₂ sinks') in (4) the ocean ($S_{OCEAN}$; GtC yr⁻¹) and (5) on land ($S_{LAND}$; GtC yr⁻¹). The $CO_2$ sinks as defined here conceptually include the response of the land (including inland waters and estuaries) and ocean (including coasts and territorial sea) to elevated $CO_2$ and changes in climate, rivers, and other environmental conditions, although in practice not all processes are accounted for (see Section 2.7). The global emissions and their partitioning among the atmosphere, ocean and land are in reality in balance, however due to imperfect spatial and/or temporal data coverage, errors in each estimate, and smaller terms not included in our budget estimate (discussed in Section 2.7), their sum does not necessarily add up to zero. We estimate a budget imbalance ($B_{IM}$), which is a measure of the mismatch between the estimated emissions and the estimated changes in the atmosphere, land and ocean, with the full global carbon budget as follows:

$$E_{FF} + E_{LUC} = G_{ATM} + S_{OCEAN} + S_{LAND} + B_{IM}. \qquad (1)$$

$G_{ATM}$ is usually reported in ppm yr⁻¹, which we convert to units of carbon mass per year, GtC yr⁻¹, using 1 ppm = 2.124 GtC (Table 1). We also include a quantification of $E_{FF}$ by country, computed with both territorial and consumption based accounting (see Sect. 2), and discuss missing terms from sources other than the combustion of fossil fuels (see Sect. 2.7).

The $CO_2$ budget has been assessed by the Intergovernmental Panel on Climate Change (IPCC) in all assessment reports (Ciais et al., 2013;Denman et al., 2007;Prentice et al., 2001;Schimel et al., 1995;Watson et al., 1990), and by others (e.g. Ballantyne et al., 2012). The IPCC methodology has been adapted and used by the Global Carbon Project (GCP, www.globalcarbonproject.org), which has coordinated a cooperative community effort for the annual publication of global carbon budgets up to year 2005 (Raupach et al., 2007; including fossil emissions only), year 2006 (Canadell et al., 2007), year 2007 (published online; GCP, 2007), year 2008 (Le Quéré et al., 2009), year 2009 (Friedlingstein et al., 2010), year 2010 (Peters et al., 2012b), year 2012 (Le Quéré et al., 2013;Peters et al., 2013), year 2013 (Le Quéré et al., 2014), year 2014 (Friedlingstein et al., 2014;Le Quéré et al., 2015b), year 2015 (Jackson et al., 2016;Le Quéré et al., 2015a), year 2016 (Le Quéré et al., 2016), and most recently year 2017 (Le Quéré et al., 2018;Peters et al., 2017). Each of these papers updated previous estimates with the latest available information for the entire time series.

We adopt a range of ±1 standard deviation (σ) to report the uncertainties in our estimates, representing a likelihood of 68% that the true value will be within the provided range if the errors

have a Gaussian distribution and no bias is assumed. This choice reflects the difficulty of characterising the uncertainty in the $CO_2$ fluxes between the atmosphere and the ocean and land reservoirs individually, particularly on an annual basis, as well as the difficulty of updating the $CO_2$ emissions from land-use change. A likelihood of 68% provides an indication of our current capability to quantify each term and its uncertainty given the available information. For comparison, the Fifth Assessment Report of the IPCC (AR5) generally reported a likelihood of 90% for large data sets whose uncertainty is well characterised, or for long time intervals less affected by year-to-year variability. Our 68% uncertainty value is near the 66% which the IPCC characterises as 'likely' for values falling into the ±1σ interval. The uncertainties reported here combine statistical analysis of the underlying data and expert judgement of the likelihood of results lying outside this range. The limitations of current information are discussed in the paper and have been examined in detail elsewhere (Ballantyne et al., 2015;Zscheischler et al., 2017). We also use a qualitative assessment of confidence level to characterise the annual estimates from each term based on the type, amount, quality and consistency of the evidence as defined by the IPCC (Stocker et al., 2013).

All quantities are presented in units of gigatonnes of carbon (GtC, $10^{15}$ gC), which is the same as petagrams of carbon (PgC; Table 1). Units of gigatonnes of $CO_2$ (or billion tonnes of $CO_2$) used in policy are equal to 3.664 multiplied by the value in units of GtC.

This paper provides a detailed description of the data sets and methodology used to compute the global carbon budget estimates for the period preindustrial (1750) to 2017 and in more detail for the period since 1959. It also provides decadal averages starting in 1960 including the last decade (2008-2017), results for the year 2017, and a projection for year 2018. Finally it provides cumulative emissions from fossil fuels and land-use change since year 1750, the preindustrial period, and since year 1870, the reference year for the cumulative carbon estimate used by the IPCC (AR5) based on the availability of global temperature data (Stocker et al., 2013). This paper is updated every year using the format of 'living data' to keep a record of budget versions and the changes in new data, revision of data, and changes in methodology that lead to changes in estimates of the carbon budget. Additional materials associated with the release of each new version will be posted at the Global Carbon Project (GCP) website (http://www.globalcarbonproject.org/carbonbudget), with fossil fuel emissions also available through the Global Carbon Atlas (http://www.globalcarbonatlas.org). With this approach, we aim

to provide the highest transparency and traceability in the reporting of $CO_2$, the key driver of
climate change.
**2    Methods**
Multiple organizations and research groups around the world generated the original
measurements and data used to complete the global carbon budget. The effort presented here is
thus mainly one of synthesis, where results from individual groups are collated, analysed and
evaluated for consistency. We facilitate access to original data with the understanding that
primary data sets will be referenced in future work (see Table 2 for how to cite the data sets).
Descriptions of the measurements, models, and methodologies follow below and in depth
descriptions of each component are described elsewhere.
This is the 13[th] version of the global carbon budget and the seventh revised version in the format
of a living data update. It builds on the latest published global carbon budget of Le Quéré et
al.(2018). The main changes are: (1) the inclusion of data to year 2017 (inclusive) and a projection
for the global carbon budget for year 2018; (2) the introduction of metrics that evaluate
components of the individual models used to estimate $S_{OCEAN}$ and $S_{LAND}$ using observations, as an
effort to document, encourage and support model improvements through time; (3) the revisions
of the $CO_2$ emissions associated with cement production based on revised clinker ratios; (4) a
projection for fossil fuel emissions for European Union 28 member states based on compiled
energy statistics; and (5) the addition of sub-section 2.7.2 on additional emissions from calcination
not included in the budget. The main methodological differences between annual carbon budgets
are summarised in Table 3.
**2.1    Fossil $CO_2$ emissions ($E_{FF}$)**
**2.1.1    Emissions estimates**
The estimates of global and national fossil $CO_2$ emissions ($E_{FF}$) include the combustion of fossil
fuels through a wide range of activities (e.g. transport, heating and cooling, industry, fossil
industry own use & gas flaring), the production of cement, and other process emissions (e.g. the
production of chemicals & fertilizers). The estimates of $E_{FF}$ rely primarily on energy consumption
data, specifically data on hydrocarbon fuels, collated and archived by several organisations
(Andres et al., 2012). We use four main data sets for historical emissions (1751-2017):

1. Global and national emission estimates for coal, oil, and gas from CDIAC for the time period 1751-2014 (Boden et al., 2017), as it is the only data set that extends back to 1751 by country.

2. Official UNFCCC national inventory reports for 1990-2016 for the 42 Annex I countries in the UNFCCC (UNFCCC, 2018). We assess these to be the most accurate estimates because they are compiled by experts within countries that have access to detailed energy data, and they are periodically reviewed.

3. The BP Statistical Review of World Energy (BP, 2018), as these are the most up-to-date estimates of national energy statistics.

4. Global and national cement emissions updated from Andrew (2018), which include revised emissions factors.

In the following section we provide more details for each data set and describe the additional modifications that are required to make the data set consistent and usable.

*CDIAC*: The CDIAC estimates have been updated annually to the year 2014, derived primarily from energy statistics published by the United Nations (UN, 2017b). Fuel masses and volumes are converted to fuel energy content using country-level coefficients provided by the UN, and then converted to $CO_2$ emissions using conversion factors that take into account the relationship between carbon content and energy (heat) content of the different fuel types (coal, oil, gas, gas flaring) and the combustion efficiency (Marland and Rotty, 1984).

*UNFCCC*: Estimates from the UNFCCC national inventory reports follow the IPCC guidelines (IPCC, 2006), but have a slightly larger system boundary than CDIAC by including emissions coming from carbonates other than in cement manufacture. We reallocate the detailed UNFCCC estimates to the CDIAC definitions of coal, oil, gas, cement, and other to allow consistent comparisons over time and between countries.

*BP*: For the most recent period when the UNFCCC (2018) and CDIAC (2015-2017) estimates are not available, we generate preliminary estimates using the BP Statistical Review of World Energy (Andres et al., 2014;Myhre et al., 2009;BP, 2018). We apply the BP growth rates by fuel type (coal, oil, gas) to estimate 2017 emissions based on 2016 estimates (UNFCCC), and to estimate 2015-2017 emissions based on 2014 estimates (CDIAC). BP's data set explicitly covers about 70 countries (96% of global emissions), and for the remaining countries we use growth rates from the sub-region the country belongs to. For the most recent years, flaring is assumed constant from the

most recent available year of data (2016 for countries that report to the UNFCCC, 2014 for the
remainder).
*Cement*: Estimates of emissions from cement production are taken directly from Andrew (2018).
Additional calcination and carbonation processes are not included explicitly here, except in
national inventories provided UNFCCC, but are discussed in Section 2.7.2.
*Country mappings*: The published CDIAC data set includes 256 countries and regions. This list
includes countries that no longer exist, such as the USSR and Yugoslavia. We reduce the list to 213
countries by reallocating emissions to the currently defined territories, using mass-preserving
aggregation or disaggregation. Examples of aggregation include merging East and West Germany
to the currently defined Germany. Examples of disaggregation include reallocating the emissions
from former USSR to the resulting independent countries. For disaggregation, we use the emission
shares when the current territories first appeared, and thus historical estimates of disaggregated
countries should be treated with extreme care. In addition, we aggregate some overseas
territories (e.g. Réunion, Guadeloupe) into their governing nations (e.g. France) to align with
UNFCCC reporting.
*Global total*: Our global estimate is based on CDIAC for fossil fuel combustion plus Andrew (2018)
for cement emissions. This is greater than the sum of emissions from all countries. This is largely
attributable to emissions that occur in international territory, in particular, the combustion of
fuels used in international shipping and aviation (bunker fuels). The emissions from international
bunker fuels are calculated based on where the fuels were loaded, but we do not include them in
the national emissions estimates. Other differences occur 1) because the sum of imports in all
countries is not equal to the sum of exports, and 2) because of inconsistent national reporting,
differing treatment of oxidation of non-fuel uses of hydrocarbons (e.g. as solvents, lubricants,
feedstocks, etc.), and 3) changes in fuel stored (Andres et al., 2012).
**2.1.2   Uncertainty assessment for $E_{FF}$**
We estimate the uncertainty of the global fossil $CO_2$ emissions at ±5% (scaled down from the
published ±10 % at ±2σ to the use of ±1σ bounds reported here; Andres et al., 2012). This is
consistent with a more detailed recent analysis of uncertainty of ±8.4% at ±2σ (Andres et al.,
2014) and at the high-end of the range of ±5-10% at ±2σ reported by Ballantyne et al. (2015). This
includes an assessment of uncertainties in the amounts of fuel consumed, the carbon and heat

contents of fuels, and the combustion efficiency. While we consider a fixed uncertainty of ±5% for all years, the uncertainty as a percentage of the emissions is growing with time because of the larger share of global emissions from emerging economies and developing countries (Marland et al., 2009). Generally, emissions from mature economies with good statistical processes have an uncertainty of only a few per cent (Marland, 2008), while emissions from developing countries such as China have uncertainties of around ±10% (for ±1σ; Gregg et al., 2008). Uncertainties of emissions are likely to be mainly systematic errors related to underlying biases of energy statistics and to the accounting method used by each country.

We assign a medium confidence to the results presented here because they are based on indirect estimates of emissions using energy data (Durant et al., 2011). There is only limited and indirect evidence for emissions, although there is high agreement among the available estimates within the given uncertainty (Andres et al., 2012;Andres et al., 2014), and emission estimates are consistent with a range of other observations (Ciais et al., 2013), even though their regional and national partitioning is more uncertain (Francey et al., 2013).

### 2.1.3   Emissions embodied in goods and services

CDIAC, UNFCCC, and BP national emission statistics 'include greenhouse gas emissions and removals taking place within national territory and offshore areas over which the country has jurisdiction' (Rypdal et al., 2006), and are called territorial emission inventories. Consumption-based emission inventories allocate emissions to products that are consumed within a country, and are conceptually calculated as the territorial emissions minus the 'embodied' territorial emissions to produce exported products plus the emissions in other countries to produce imported products (Consumption = Territorial – Exports + Imports). Consumption-based emission attribution results (e.g. Davis and Caldeira, 2010) provide additional information to territorial-based emissions that can be used to understand emission drivers (Hertwich and Peters, 2009) and quantify emission transfers by the trade of products between countries (Peters et al., 2011b). The consumption-based emissions have the same global total, but reflect the trade-driven movement of emissions across the Earth's surface in response to human activities.

We estimate consumption-based emissions from 1990-2016 by enumerating the global supply chain using a global model of the economic relationships between economic sectors within and between every country (Andrew and Peters, 2013;Peters et al., 2011a). Our analysis is based on

the economic and trade data from the Global Trade and Analysis Project (GTAP; Narayanan et al., 2015), and we make detailed estimates for the years 1997 (GTAP version 5), 2001 (GTAP6), and 2004, 2007, and 2011 (GTAP9.2), covering 57 sectors and 141 countries and regions. The detailed results are then extended into an annual time-series from 1990 to the latest year of the Gross Domestic Product (GDP) data (2016 in this budget), using GDP data by expenditure in current exchange rate of US dollars (USD; from the UN National Accounts main Aggregrates database; UN, 2017a) and time series of trade data from GTAP (based on the methodology in Peters et al., 2011b ). We estimate the sector-level $CO_2$ emissions using the GTAP data and methodology, include flaring and cement emissions from CDIAC, and then scale the national totals (excluding bunker fuels) to match the emission estimates from the carbon budget. We do not provide a separate uncertainty estimate for the consumption-based emissions, but based on model comparisons and sensitivity analysis, they are unlikely to be significantly different than for the territorial emission estimates (Peters et al., 2012a).

**2.1.4   Growth rate in emissions**

We report the annual growth rate in emissions for adjacent years (in percent per year) by calculating the difference between the two years and then normalising to the emissions in the first year: $(E_{FF}(t_{0+1})-E_{FF}(t_0))/E_{FF}(t_0)\times100\%$. We apply a leap-year adjustment where relevant to ensure valid interpretations of annual growth rates. This affects the growth rate by about 0.3% yr$^-$ $^1$ (1/365) and causes growth rates to go up approximately 0.3% if the first year is a leap year and down 0.3% if the second year is a leap year.

The relative growth rate of $E_{FF}$ over time periods of greater than one year can be re-written using its logarithm equivalent as follows:

$$\frac{1}{E_{FF}} \frac{dE_{FF}}{dt} = \frac{d(lnE_{FF})}{dt} \qquad (2)$$

Here we calculate relative growth rates in emissions for multi-year periods (e.g. a decade) by fitting a linear trend to $ln(E_{FF})$ in Eq. (2), reported in percent per year.

**2.1.5   Emissions projections**

To gain insight on emission trends for the current year (2018), we provide an assessment of global fossil $CO_2$ emissions, $E_{FF}$, by combining individual assessments of emissions for China, USA, the EU, and India (the four countries/regions with the largest emissions), and the rest of the world.

Our 2018 estimate for China uses: (1) the sum of domestic production (NBS, 2018b) and net
imports (General Administration of Customs of the People's Republic of China, 2018) for coal, oil
and natural gas and production of cement (NBS, 2018b) from preliminary statistics for January
through September of 2018; and (2) historical relationships between January-September
production and import statistics and full-year consumption figures from final official statistics for
2000-2016 (NBS, 2015, 2017) and preliminary full-year data for 2017 (NBS, 2018a). See also Liu et
al. (subm.) and Jackson et al. (2018b) for details. The uncertainty is based on the variance of the
difference between the January-September and full-year data from historical data, as well as
typical variance in the preliminary full-year data used for 2017 and typical changes in the energy
content of coal for the period 2013-2016  (NBS, 2017, 2015). We note that developments for the
final three months this year may be atypical due to the ongoing trade disputes between China and
the U.S., and this additional uncertainty has not been quantified. Results and uncertainties are
discussed further in Sect. 3.4.1.
For the USA, we use the forecast of the U.S. Energy Information Administration (EIA) for emissions
from fossil fuels (EIA, 2018). This is based on an energy forecasting model which is updated
monthly (last update to October), and takes into account heating-degree days, household
expenditures by fuel type, energy markets, policies, and other effects. We combine this with our
estimate of emissions from cement production using the monthly U.S. cement data from USGS for
January-August, assuming changes in cement production over the first part of the year apply
throughout the year. While the EIA's forecasts for current full-year emissions have on average
been revised downwards, only ten such forecasts are available, so we conservatively use the full
range of adjustments following revision, and additionally assume symmetrical uncertainty to give
±2.5% around the central forecast.
For India, we use (1) monthly coal production and sales data from the Ministry of Mines (2018),
Coal India Limited (CIL, 2018) and Singareni Collieries Company Limited (SCCL, 2018), combined
with import data from the Ministry of Commerce and Industry (MCI, 2018) and power station
stocks data from the Central Electricity Authority (CEA, 2018); (2) monthly oil production and
consumption data from the Ministry of Petroleum and Natural Gas (PPAC, 2018a); (3) monthly
natural gas production and import data from the Ministry of Petroleum and Natural Gas (PPAC,
2018b); and (4) monthly cement production data from the Office of the Economic Advisor (OEA,
2018). All data were available for January to September or October. We use Holt-Winters
exponential smoothing with multiplicative seasonality (Chatfield, 1978) on each of these four
emissions series to project to the end of the current year. This iterative method produces
estimates of both trend and seasonality at the end of the observation period that are a function of
all prior observations, weighted most strongly to more recent data, while maintaining some
smoothing effect. The main source of uncertainty in the projection of India's emissions is the
assumption of continued trends and typical seasonality.
For the EU, we use (1) monthly coal supply data from Eurostat for the first 6-9 months of the year
(Eurostat, 2018) cross-checked with more recent data on coal-generated electricity from ENTSO-E
for January through October (ENTSO-E, 2018); (2) monthly oil and gas demand data for January
through August from the Joint Organisations Data Initiative (JODI, 2018); and (3) cement
production is assumed stable. For oil and gas emissions we apply the Holt-Winters method
separately to each country and energy carrier to project to the end of the current year, while for
coal — which is much less strongly seasonal because of strong weather variations – we assume
the remaining months of the year are the same as the previous year in each country.
For the rest of the world, we use the close relationship between the growth in GDP and the
growth in emissions (Raupach et al., 2007) to project emissions for the current year. This is based
on a simplified Kaya Identity, whereby $E_{FF}$ (GtC yr$^{-1}$) is decomposed by the product of GDP (USD yr$^{-1}$)
$^{-1}$) and the fossil fuel carbon intensity of the economy ($I_{FF}$; GtC USD$^{-1}$) as follows:

$$E_{FF} = \ GDP \ \times \ I_{FF} \tag{3}$$

Taking a time derivative of Equation (3) and rearranging gives:

$$\frac{1}{E_{FF}}\frac{dE_{FF}}{dt} = \ \frac{1}{GDP}\frac{dGDP}{dt} + \frac{1}{I_{FF}}\frac{dI_{FF}}{dt} \tag{4}$$

where the left-hand term is the relative growth rate of $E_{FF}$, and the right-hand terms are the
relative growth rates of GDP and $I_{FF}$, respectively, which can simply be added linearly to give the
overall growth rate.
The growth rates are reported in percent by multiplying each term by 100. As preliminary
estimates of annual change in GDP are made well before the end of a calendar year, making
assumptions on the growth rate of $I_{FF}$ allows us to make projections of the annual change in $CO_2$
emissions well before the end of a calendar year. The $I_{FF}$ is based on GDP in constant PPP
(purchasing power parity) from the International Energy Agency (IEA) up to 2016 (IEA/OECD,
2017) and extended using the International Monetary Fund (IMF) growth rates for 2016 and 2017
(IMF, 2018). Interannual variability in $I_{FF}$ is the largest source of uncertainty in the GDP-based
emissions projections. We thus use the standard deviation of the annual $I_{FF}$ for the period 2007-
2017 as a measure of uncertainty, reflecting a ±1σ as in the rest of the carbon budget. This is
±1.0% $yr^{-1}$ for the rest of the world (global emissions minus China, USA, EU and India).
The 2018 projection for the world is made of the sum of the projections for China, USA, EU, India,
and the rest of the world. The uncertainty is added in quadrature among the five regions. The
uncertainty here reflects the best of our expert opinion.
**2.2    $CO_2$ emissions from land use, land-use change and forestry ($E_{LUC}$)**
The net $CO_2$ flux from land use, land-use change and forestry ($E_{LUC}$, called land-use change
emissions in the rest of the text) include $CO_2$ fluxes from deforestation, afforestation, logging and
forest degradation (including harvest activity), shifting cultivation (cycle of cutting forest for
agriculture, then abandoning), and regrowth of forests following wood harvest or abandonment
of agriculture. Only some land management activities are included in our land-use change
emissions estimates (Table A1). Some of these activities lead to emissions of $CO_2$ to the
atmosphere, while others lead to $CO_2$ sinks. $E_{LUC}$ is the net sum of emissions and removals due to
all anthropogenic activities considered. Our annual estimate for 1959-2017 is provided as the
average of results from two bookkeeping models (Sect. 2.2.1): the estimate published by
Houghton and Nassikas (2017; hereafter H&N2017) extended here to 2017, and an estimate using
the BLUE model (Bookkeeping of Land Use Emissions; Hansis et al., 2015). In addition, we use
results from Dynamic Global Vegetation Models (DGVMs; see Sect. 2.2.3 and Table 4), to help
quantify the uncertainty in $E_{LUC}$, and thus better characterise our understanding. The three
methods are described below, and differences are discussed in Sect. 3.2.
**2.2.1    Bookkeeping models**
Land-use change $CO_2$ emissions and uptake fluxes are calculated by two bookkeeping models.
Both are based on the original bookkeeping approach of Houghton (2003) that keeps track of the
carbon stored in vegetation and soils before and after a land-use change (transitions between
various natural vegetation types, croplands and pastures). Literature-based response curves
describe decay of vegetation and soil carbon, including transfer to product pools of different
lifetimes, as well as carbon uptake due to regrowth. In addition, the bookkeeping models
represent long-term degradation of primary forest as lowered standing vegetation and soil carbon
stocks in secondary forests, and also include forest management practices such as wood harvests.
The bookkeeping models do not include land ecosystems' transient response to changes in
climate, atmospheric $CO_2$ and other environmental factors, and the carbon densities are based on
contemporary data reflecting stable environmental conditions at that time. Since carbon densities
remain fixed over time in bookkeeping models, the additional sink capacity that ecosystems
provide in response to $CO_2$-fertilization and some other environmental changes is not captured by
these models (Pongratz et al., 2014; see Section 2.7.3).
The H&N2017 and BLUE models differ in (1) computational units (country-level vs spatially explicit
treatment of land-use change), (2) processes represented (see Table A1), and (3) carbon densities
assigned to vegetation and soil of each vegetation type. A notable change of H&N2017 over the
original approach by Houghton et al. (2003) used in earlier budget estimates is that no shifting
cultivation or other back- and forth-transitions at a level below country are included. Only a
decline in forest area in a country as indicated by the Forest Resource Assessment of the FAO that
exceeds the expansion of agricultural area as indicated by FAO is assumed to represent a
concurrent expansion and abandonment of cropland. In contrast, the BLUE model includes sub-
grid-scale transitions at the grid level between all vegetation types as indicated by the harmonized
land-use change data (LUH2) data set (Hurtt et al., in prep.). Furthermore, H&N2017 assume
conversion of natural grasslands to pasture, while BLUE allocates pasture proportionally on all
natural vegetation that exist in a gridcell. This is one reason for generally higher emissions in
BLUE. H&N2017 add carbon emissions from peat burning based on the Global Fire Emission
Database (GFED4s; van der Werf et al. (2017)), and peat drainage, based on estimates by Hooijer
et al. (2010) to the output of their bookkeeping model for the countries of Indonesia and
Malaysia. Peat burning and emissions from the organic layers of drained peat soils, which are not
captured by bookkeeping methods directly, need to be included to represent the substantially
larger emissions and interannual variability due to synergies of land-use and climate variability in
Southeast Asia, in particular during El-Niño events. Similarly to H&N2017, peat burning and
drainage-related emissions are also added to the BLUE estimate.
The two bookkeeping estimates used in this study also differ with respect to the land use change
data used to drive the models. H&N2017 base their estimates directly on the Forest Resource
Assessment of the FAO which provides statistics on forest-area change and management at
intervals of five years currently updated until 2015 (FAO, 2015). The data is based on country
reporting to FAO, and may include remote-sensing information in more recent assessments.
Changes in land use other than forests are based on annual, national changes in cropland and
pasture areas reported by FAO (FAOSTAT, 2015). BLUE uses the harmonized land-use change data
LUH2 (Hurtt et al., in prep.), which describes land use change, also based on the FAO data, but
downscaled at a quarter-degree spatial resolution, considering sub-grid-scale transitions between
primary forest, secondary forest, cropland, pasture and rangeland. The LUH2 data provides a new
distinction between rangelands and pasture. To constrain the models' interpretation on whether
rangeland implies the original natural vegetation to be transformed to grassland or not (e.g.,
browsing on shrubland), a new forest mask was provided with LUH2; forest is assumed to be
transformed, while all other natural vegetation remains. This is implemented in BLUE.
The estimate of H&N2017 was extended here by two years (to 2017) by adding the anomaly of
total tropical emissions (peat drainage from Hooijer et al. (2010), peat burning as well as tropical
deforestation and degradation fires from GFED4s) over the previous decade (2006-2015) to the
decadal average of the bookkeeping result.
**2.2.2   Dynamic Global Vegetation Models (DGVMs)**
Land-use change $CO_2$ emissions have also been estimated using an ensemble of 16 DGVM
simulations. The DGVMs account for deforestation and regrowth, the most important
components of $E_{LUC}$, but they do not represent all processes resulting directly from human
activities on land (Table A1). All DGVMs represent processes of vegetation growth and mortality,
as well as decomposition of dead organic matter associated with natural cycles, and include the
vegetation and soil carbon response to increasing atmospheric $CO_2$ levels and to climate variability
and change. Some models explicitly simulate the coupling of carbon and nitrogen cycles and
account for atmospheric N deposition (Table A1). The DGVMs are independent from the other
budget terms except for their use of atmospheric $CO_2$ concentration to calculate the fertilization
effect of $CO_2$ on plant photosynthesis.
The DGVMs used the HYDE land-use change data set (Klein Goldewijk et al., 2017a;Klein Goldewijk
et al., 2017b), which provides annual, half-degree, fractional data on cropland and pasture. These
data are based on annual FAO statistics of change in agricultural land area available to 2012. The
FAOSTAT land use database is updated annually, currently covering the period 1961-2016 (but
used here to 2015 because of the timing of data availability).  HYDE applied annual changes in FAO
data to the year 2012 data from the previous release to derive new 2013-2015 data. After the
year 2015 HYDE extrapolates cropland, pasture, and urban land use data until the year 2018.
Some models also use an update of the more comprehensive harmonised land-use data set (Hurtt
et al., 2011), that further includes fractional data on primary and secondary forest vegetation, as
well as all underlying transitions between land-use states (Hurtt et al., in prep.; Table A1). This
new data set is of quarter degree fractional areas of land use states and all transitions between
those states, including a new wood harvest reconstruction, new representation of shifting
cultivation, crop rotations, management information including irrigation and fertilizer application.
The land-use states now include five different crop types in addition to the pasture-rangeland split
discussed before. Wood harvest patterns are constrained with Landsat tree cover loss data.
DGVMs implement land-use change differently (e.g. an increased cropland fraction in a grid cell
can either be at the expense of grassland or shrubs, or forest, the latter resulting in deforestation;
land cover fractions of the non-agricultural land differ between models). Similarly, model-specific
assumptions are applied to convert deforested biomass or deforested area, and other forest
product pools into carbon, and different choices are made regarding the allocation of rangelands
as natural vegetation or pastures.
The DGVM model runs were forced by either the merged monthly CRU and 6 hourly JRA-55
dataset or by the monthly CRU dataset, both providing observation based temperature,
precipitation, and incoming surface radiation on a 0.5°x0.5° grid and updated to 2017 (Harris et
al., 2014). The combination of CRU monthly data with 6 hourly forcing is updated this year from
NCEP to JRA-55 (Kobayashi et al., 2015), adapting the methodology used in previous years (Viovy,
2016) to the specifics of the JRA-55 data. The forcing data also include global atmospheric $CO_2$,
which changes over time (Dlugokencky and Tans, 2018), and gridded, time dependent N
deposition (as used in some models; Table A1).
Two sets of simulations were performed with the DGVMs. Both applied historical changes in
climate, atmospheric $CO_2$ concentration, and N deposition. The two sets of simulations differ,
however, with respect to land use: one set applies historical changes in land use, the other a time-
invariant preindustrial land cover distribution and preindustrial wood harvest rates. By difference
of the two simulations, the dynamic evolution of vegetation biomass and soil carbon pools in
response to land use change can be quantified in each model ($E_{LUC}$). We only retain model outputs
with positive $E_{LUC}$, i.e. a positive flux to the atmosphere, during the 1990s (Table A1). Using the
difference between these two DGVM simulations to diagnose $E_{LUC}$ means the DGVMs account for
the loss of additional sink capacity (around 0.3 GtC yr$^{-1}$; see Section 2.7.3), while the bookkeeping
models do not.

### 2.2.3   Uncertainty assessment for $E_{LUC}$

Differences between the bookkeeping models and DGVM models originate from three main
sources:  the different methodologies; the underlying land use/land cover data set, and the
different processes represented (Table A1). We examine the results from the DGVM models and
of the bookkeeping method, and use the resulting variations as a way to characterise the
uncertainty in $E_{LUC}$.
The $E_{LUC}$ estimate from the DGVMs multi-model mean is consistent with the average of the
emissions from the bookkeeping models (Table 5). However there are large differences among
individual DGVMs (standard deviation at around 0.6-0.7 GtC yr$^{-1}$; Table 5), between the two
bookkeeping models (average of 0.7 GtC yr$^{-1}$), and between the current estimate of H&N2017 and
its previous model version (Houghton et al., 2012). The uncertainty in $E_{LUC}$ of ±0.7 GtC yr$^{-1}$ reflects
our best value judgment that there is at least 68% chance (±1σ) that the true land-use change
emission lies within the given range, for the range of processes considered here. Prior to the year
1959, the uncertainty in $E_{LUC}$ was taken from the standard deviation of the DGVMs. We assign low
confidence to the annual estimates of $E_{LUC}$ because of the inconsistencies among estimates and of
the difficulties to quantify some of the processes in DGVMs.

### 2.2.4   Emissions projections

We project emissions for both H&N2017 and BLUE for 2018 using the same approach as for the
extrapolation of H&N2017 for 2016-2017. Peat burning as well as tropical deforestation and
degradation are estimated using active fire data (MCD14ML; Giglio et al. (2016)), which scales
almost linearly with GFED (van der Werf et al., 2017), and thus allows for tracking fire emissions in
deforestation and tropical peat zones in near-real time. During most years, emissions during
January-October cover most of the fire season in the Amazon and Southeast Asia, where a large
part of the global deforestation takes place.

**2.3    Growth rate in atmospheric CO$_2$ concentration (G$_{ATM}$)**

**2.3.1    Global growth rate in atmospheric CO$_2$ concentration**

The rate of growth of the atmospheric CO$_2$ concentration is provided by the US National Oceanic and Atmospheric Administration Earth System Research Laboratory (NOAA/ESRL; Dlugokencky and Tans, 2018), which is updated from Ballantyne et al. (2012). For the 1959-1979 period, the global growth rate is based on measurements of atmospheric CO$_2$ concentration averaged from the Mauna Loa and South Pole stations, as observed by the CO$_2$ Program at Scripps Institution of Oceanography (Keeling et al., 1976). For the 1980-2017 time period, the global growth rate is based on the average of multiple stations selected from the marine boundary layer sites with well-mixed background air (Ballantyne et al., 2012), after fitting each station with a smoothed curve as a function of time, and averaging by latitude band (Masarie and Tans, 1995). The annual growth rate is estimated by Dlugokencky and Tans (2018) from atmospheric CO$_2$ concentration by taking the average of the most recent December-January months corrected for the average seasonal cycle and subtracting this same average one year earlier. The growth rate in units of ppm yr$^{-1}$ is converted to units of GtC yr$^{-1}$ by multiplying by a factor of 2.124 GtC per ppm (Ballantyne et al., 2012).

The uncertainty around the atmospheric growth rate is due to three main factors. First, the long-term reproducibility of reference gas standards (around 0.03 ppm for 1σ from the 1980s). Second, small unexplained systematic analytical errors that may have a duration of several months to two years come and go. They have been simulated by randomizing both the duration and the magnitude (determined from the existing evidence) in a Monte Carlo procedure. Third, the network composition of the marine boundary layer with some sites coming or going, gaps in the time series at each site, etc (Dlugokencky and Tans, 2018). The latter uncertainty was estimated by NOAA/ESRL with a Monte Carlo method by constructing 100 "alternative" networks (NOAA/ESRL 2017; Masarie, and Tans, 1995). The second and third uncertainties, summed in quadrature, add up to 0.085 ppm on average (Dlugokencky and Tans, 2018). Fourth, the uncertainty associated with using the average CO$_2$ concentration from a surface network to approximate the true atmospheric average CO$_2$ concentration (mass-weighted, in 3 dimensions) as needed to assess the total atmospheric CO$_2$ burden. In reality, CO$_2$ variations measured at the stations will not exactly track changes in total atmospheric burden, with offsets in magnitude and phasing due to vertical and horizontal mixing. This effect must be very small on decadal and

longer time scales, when the atmosphere can be considered well mixed. Preliminary estimates
suggest this effect would increase the annual uncertainty, but a full analysis is not yet available.
We therefore maintain an uncertainty around the annual growth rate based on the multiple
stations data set ranges between 0.11 and 0.72 GtC yr$^{-1}$, with a mean of 0.61 GtC yr$^{-1}$ for 1959-
1979 and 0.18 GtC yr$^{-1}$ for 1980-2017, when a larger set of stations were available as provided by
Dlugokencky and Tans (2018), but recognise further exploration of this uncertainty is required. At
this time, we estimate the uncertainty of the decadal averaged growth rate after 1980 at 0.02 GtC
yr$^{-1}$ based on the calibration and the annual growth rate uncertainty, but stretched over a 10-year
interval. For years prior to 1980, we estimate the decadal averaged uncertainty to be 0.07 GtC yr$^{-1}$
based on a factor proportional to the annual uncertainty prior and after 1980 (0.61/0.18*0.02 GtC
yr$^{-1}$).
We assign a high confidence to the annual estimates of $G_{ATM}$ because they are based on direct
measurements from multiple and consistent instruments and stations distributed around the
world (Ballantyne et al., 2012).
In order to estimate the total carbon accumulated in the atmosphere since 1750 or 1870, we use
an atmospheric $CO_2$ concentration of 277 ± 3 ppm or 288 ± 3 ppm, respectively, based on a cubic
spline fit to ice core data (Joos and Spahni, 2008). The uncertainty of ±3 ppm (converted to ±1σ) is
taken directly from the IPCC's assessment (Ciais et al., 2013). Typical uncertainties in the growth
rate in atmospheric $CO_2$ concentration from ice core data are equivalent to ±0.1-0.15 GtC yr$^{-1}$ as
evaluated from the Law Dome data (Etheridge et al., 1996) for individual 20-year intervals over
the period from 1870 to 1960 (Bruno and Joos, 1997).
**2.3.2   Atmospheric growth rate projection**
We provide an assessment of $G_{ATM}$ for 2018 based on the observed increase in atmospheric $CO_2$
concentration at the Mauna Loa station for January to October, and a mean growth rate over the
past 5 years for the months November to December. Growth at Mauna Loa is closely correlated
with the global growth (r=0.95) and is used here as a proxy for global growth, but the regression is
not 1-to-1. We also adjust the projected global growth rate to take this into account. The
assessment method used this year differs from the forecast method used in Le Quéré et al. (2018)
based on the relationship between annual $CO_2$ growth rate and sea surface temperatures (SSTs) in
the Niño3.4 region of Betts et al. (2016). A change was introduced because although the observed
growth rate for 2017 of 2.2 ppm was within the projection range of 2.5 ± 0.5 ppm of last year (Le
Quéré et al. 2018), the forecast values for 2018 for January to October are too high by
approximately 0.4 ppm above observed values on average. The reasons for the difference are
being investigated. The use of observed growth at MLO for the first half of the year is thought to
be more robust because of its high correlation with the global growth rate. Furthermore,
additional analysis suggests that the first half of the year shows more interannual variability than
the second half of the year, so that the exact projection method applied to November-December
has only a small impact (<0.1 ppm) on the projection of the full year.  Uncertainty is estimated
from past variability using the standard deviation of the last 5 years' monthly growth rates.
**2.4    Ocean $CO_2$ sink**
Estimates of the global ocean $CO_2$ sink $S_{OCEAN}$ are from an ensemble of global ocean
biogeochemistry models (GOBMs) that meet observational constraints over the 1990s (see
below). We use observation-based estimates of $S_{OCEAN}$ to provide a qualitative assessment of
confidence in the reported results, and to estimate the cumulative accumulation of $S_{OCEAN}$ over
the preindustrial period.
**2.4.1    Observation-based estimates**
We use the observational constraints assessed by IPCC of a mean ocean $CO_2$ sink of 2.2 ± 0.4 GtC
$yr^{-1}$ for the 1990s (Denman et al., 2007) to verify that the GOBMs provide a realistic assessment of
$S_{OCEAN}$. This is based on indirect observations with seven different methodologies and their
uncertainties, using the methods that are deemed most reliable for the assessment of this
quantity (Denman et al., 2007). The IPCC confirmed this assessment in 2013 (Ciais et al., 2013).
The observational-based estimates use the ocean/land $CO_2$ sink partitioning from observed
atmospheric $O_2/N_2$ concentration trends (Manning and Keeling, 2006; updated in Keeling and
Manning 2014), an oceanic inversion method constrained by ocean biogeochemistry data
(Mikaloff Fletcher et al., 2006), and a method based on penetration time scale for CFCs (McNeil et
al., 2003). The IPCC estimate of 2.2 GtC $yr^{-1}$ for the 1990s is consistent with a range of methods
(Wanninkhof et al., 2013).
We also use two estimates of the ocean $CO_2$ sink and its variability based on interpolations of
measurements of surface ocean fugacity of $CO_2$ (pCO2 corrected for the non-ideal behaviour of
the gas; Pfeil et al., 2013). We refer to these as pCO2-based flux estimates. The measurements are

from the Surface Ocean $CO_2$ Atlas version 6, which is an update of version 3 (Bakker et al., 2016) and contains quality-controlled data to 2017 (see data attribution Table A4). The SOCAT v6 data were mapped using a data-driven diagnostic method (Rödenbeck et al., 2013) and a combined self-organising map and feed-forward neural network (Landschützer et al., 2014). The global $pCO_2$-based flux estimates were adjusted to remove the preindustrial ocean source of $CO_2$ to the atmosphere of 0.78 GtC $yr^{-1}$ from river input to the ocean (Resplandy et al., 2018), per our definition of $S_{OCEAN}$. Several other ocean sink products based on observations are also available but they continue to show large unresolved discrepancies with observed variability. Here we used the two $pCO_2$-based flux products that had the best fit to observations for their representation of tropical and global variability (Rödenbeck et al., 2015).

We further use results from two diagnostic ocean models of Khatiwala et al. (2013) and DeVries (2014) to estimate the anthropogenic carbon accumulated in the ocean prior to 1959. The two approaches assume constant ocean circulation and biological fluxes, with $S_{OCEAN}$ estimated as a response in the change in atmospheric $CO_2$ concentration calibrated to observations. The uncertainty in cumulative uptake of ±20 GtC (converted to ±1σ) is taken directly from the IPCC's review of the literature (Rhein et al., 2013), or about ±30% for the annual values (Khatiwala et al., 2009).

**2.4.2   Global Ocean Biogeochemistry Models (GOBMs)**

The ocean $CO_2$ sink for 1959-2017 is estimated using seven GOBMs (Table A2). The GOBMs represent the physical, chemical and biological processes that influence the surface ocean concentration of $CO_2$ and thus the air-sea $CO_2$ flux. The GOBMs are forced by meteorological reanalysis and atmospheric $CO_2$ concentration data available for the entire time period. They mostly differ in the source of the atmospheric forcing data (meteorological reanalysis), spin up strategies, and in their horizontal and vertical resolutions  (Table A2). GOBMs do not include the effects of anthropogenic changes in nutrient supply, which could lead to an increase of the ocean sink of up to about 0.3 GtC $yr^{-1}$ over the industrial period (Duce et al., 2008). They also do not include the perturbation associated with changes in riverine organic carbon (see Sect. 2.7.3).

### 2.4.3  GOBM evaluation and uncertainty assessment for $S_{OCEAN}$

The mean ocean $CO_2$ sink for all GOBMs fall within 90% confidence of the observed range, or 1.6 to 2.8 GtC $yr^{-1}$ for the 1990s. Here we have adjusted the confidence interval to the IPCC confidence interval of 90% to avoid rejecting models that may be outliers but are still plausible.

The GOBMs and flux products have been further evaluated using $fCO_2$ from the SOCAT v6 database. We focused this initial evaluation on the interannual mismatch metric proposed by Rödenbeck et al. (2015) for the comparison of flux products. The metric provides a measure of the mismatch between observations and models or flux products on the x-axis as well as a measure of the amplitude of the interannual variability on the y-axis. A smaller number on the x-axis indicates a better fit with observations. The amplitude of the interannual variability of $S_{OCEAN}$ (y-axis) is calculated as the temporal standard deviation of the $CO_2$ flux time-series.

The calculation for the x-axis is done as follows: 1) the mismatch between the observed and the modelled $fCO_2$ is calculated for the period 1985 to 2017 (except for IPSL model which uses 1985 to 2015 due to data availability), but only for grid points where actual observations exist. 2) The interannual variability of this mismatch is calculated as the temporal standard deviation of the mismatch. 3) To put numbers into perspective, the interannual variability of the mismatch is reported relative to the interannual variability of the mismatch between a benchmark $fCO_2$ field and the observations. The benchmark $fCO_2$ field is designed to have no interannual variability, i.e. it is calculated as the mean seasonal cycle at each grid point over the full period plus the deseasonalized atmospheric $fCO_2$ increase over time. By definition, the interannual variability of the misfit between benchmark and observations is large as the benchmark field does not contain any interannual variability from the ocean. A smaller relative interannual variability mismatch indicates a better fit between observed and modelled $fCO_2$. This metric is chosen because it is the most direct measure of the year-to-year variability of $S_{OCEAN}$ in ocean biogeochemistry models. We apply the metric globally and by latitude bands (Fig. B1). Results are shown in Fig. B1 and discussed in Section 3.1.3.

The uncertainty around the mean ocean sink of anthropogenic $CO_2$ was quantified by Denman et al. (2007) for the 1990s (see Sect. 2.4.1). To quantify the uncertainty around annual values, we examine the standard deviation of the GOBM ensemble, which averages between 0.2 and 0.3 GtC $yr^{-1}$ during 1959-2017. We estimate that the uncertainty in the annual ocean $CO_2$ sink is about ± 0.5 GtC $yr^{-1}$ from the combined uncertainty of the mean flux based on observations of ± 0.4 GtC

yr$^{-1}$ and the standard deviation across GOBMs of up to ± 0.3 GtC yr$^{-1}$, reflecting both the
uncertainty in the mean sink from observations during the 1990's (Denman et al., 2007; Section
2.4.1) and in the interannual variability as assessed by GOBMs.
We examine the consistency between the variability of the model-based and the $pCO_2$-based flux
products to assess confidence in $S_{OCEAN}$. The interannual variability of the ocean fluxes (quantified
as the standard deviation) of the two $pCO_2$-based flux products for 1985-2017 (where they
overlap) is ± 0.36 GtC yr$^{-1}$ (Rödenbeck et al., 2014) and ± 0.38 GtC yr$^{-1}$ (Landschützer et al., 2015),
compared to ± 0.29 GtC yr$^{-1}$ for the GOBM ensemble. The standard deviation includes a
component of trend and decadal variability in addition to interannual variability, and their relative
influence differs across estimates. Individual estimates (both GOBM and flux products) generally
produce a higher ocean $CO_2$ sink during strong El Niño events. The annual $pCO_2$-based flux
products correlate with the ocean $CO_2$ sink estimated here with a correlation of r = 0.75 (0.59 to
0.79 for individual GOBMs), and r = 0.80 (0.71 to 0.81) for the $pCO_2$-based flux products of
Rödenbeck et al. (2014) and Landschützer et al. (2015), respectively (simple linear regression),
with their mutual correlation at 0.73. The agreement between models and the flux products
reflects some consistency in their representation of underlying variability since there is little
overlap in their methodology or use of observations. The use of annual data for the correlation
may reduce the strength of the relationship because the dominant source of variability associated
with El Niño events is less than one year. We assess a medium confidence level to the annual
ocean $CO_2$ sink and its uncertainty because it is based on multiple lines of evidence, and the
results are consistent in that the interannual variability in the GOBMs and data-based estimates
are all generally small compared to the variability in the growth rate of atmospheric $CO_2$
concentration.
**2.5    Terrestrial $CO_2$ sink**
**2.5.1    DGVM simulations**
The terrestrial land sink ($S_{LAND}$) is thought to be due to the combined effects of fertilisation by
rising atmospheric $CO_2$ and N deposition on plant growth, as well as the effects of climate change
such as the lengthening of the growing season in northern temperate and boreal areas. $S_{LAND}$ does
not include land sinks directly resulting from land use and land-use change (e.g. regrowth of
vegetation) as these are part of the land use flux ($E_{LUC}$), although system boundaries make it
difficult to attribute exactly $CO_2$ fluxes on land between $S_{LAND}$ and $E_{LUC}$ (Erb et al., 2013).
$S_{LAND}$ is estimated from the multi-model mean of the DGVMs (Table 4). As described in section
2.2.3, DGVM simulations include all climate variability and $CO_2$ effects over land, with some
DGVMs also including the effect of N deposition. The DGVMs do not include the perturbation
associated with changes in river organic carbon, which is discussed section 2.7.
**2.5.2    DGVM evaluation and uncertainty assessment for $S_{LAND}$**
We apply three criteria for minimum DGVM realism by including only those DGVMs with (1)
steady state after spin up, (2) net land fluxes ($S_{LAND}$ – $E_{LUC}$) that is an atmosphere-to-land carbon
flux over the 1990s ranging between -0.3 and 2.3GtC yr$^{-1}$, within 90% confidence of constraints by
global atmospheric and oceanic observations (Keeling and Manning, 2014;Wanninkhof et al.,
2013), and (3) global $E_{LUC}$ that is a carbon source to the atmosphere over the 1990s. All 16 DGVMs
meet the three criteria.
In addition, the DGVM results are now also evaluated using the International Land Model
Benchmarking system (ILAMB; Collier et al., 2018). This evaluation is provided here to document,
encourage and support model improvements through time. ILAMB variables cover key processes
that are relevant for the quantification of $S_{LAND}$ and resulting aggregated outcomes. The selected
variables are vegetation biomass, gross primary productivity, leaf area index, net ecosystem
exchange, ecosystem respiration, evapotranspiration, and runoff (see Fig. B2 for the results and
for the list of observed databases). Results are shown in Fig. B2 and discussed in Section 3.1.3.
For the uncertainty, we use the standard deviation of the annual $CO_2$ sink across the DGVMs,
which averages to ± 0.8 GtC yr$^{-1}$ for the period 1959 to 2017. We attach a medium confidence
level to the annual land $CO_2$ sink and its uncertainty because the estimates from the residual
budget and averaged DGVMs match well within their respective uncertainties (Table 5).
**2.6    The atmospheric perspective**
The world-wide network of atmospheric measurements can be used with atmospheric inversion
methods to constrain the location of the combined total surface $CO_2$ fluxes from all sources,
including fossil and land-use change emissions and land and ocean $CO_2$ fluxes. The inversions
assume $E_{FF}$ to be well known, and they solve for the spatial and temporal distribution of land and
ocean fluxes from the residual gradients of $CO_2$ between stations that are not explained by fossil
fuel emissions.
Four atmospheric inversions (Table A3) used atmospheric $CO_2$ data to the end of 2017 (including
preliminary values in some cases) to infer the spatio-temporal distribution of the $CO_2$ flux
exchanged between the atmosphere and the land or oceans. We focus here on the largest and
most consistent sources of information, namely the total land and ocean $CO_2$ flux and their
partitioning among the mid-high latitude region of the Northern Hemisphere (30°N-90°N), the
tropics (30°S-30°N) and the mid-high latitude region of the Southern Hemisphere (30°S-90°S). We
also break down those estimates for the land and ocean regions separately, to further scrutinise
the constraints from atmospheric observations. We use these estimates to comment on the
consistency across various data streams and process-based estimates.
**Atmospheric inversions**
The four inversion systems used in this release are the CarbonTracker Europe (CTE; van der Laan-
Luijkx et al., 2017), the Jena CarboScope (Rödenbeck, 2005), the Copernicus Atmosphere
Monitoring Service (CAMS; Chevallier et al., 2005), and MIROC (Patra et al., 2018). See Table A3
for version numbers. The inversions are based on the same Bayesian inversion principles that
interpret the same, for the most part, observed time series (or subsets thereof), but use different
methodologies (Table A3). These differences mainly concern the selection of atmospheric $CO_2$
data, the used prior fluxes, spatial breakdown (i.e. grid size), assumed correlation structures, and
mathematical approach. The details of these approaches are documented extensively in the
references provided above. Each system uses a different transport model, which was
demonstrated to be a driving factor behind differences in atmospheric-based flux estimates, and
specifically their distribution across latitudinal bands (e.g., Gaubert et al., 2018).
The inversions use atmospheric $CO_2$ observations from various flask and in situ networks, as
detailed in Table A3. They prescribe global $E_{FF}$, which is scaled to the present study for CAMS and
CTE, while slightly lower $E_{FF}$ values based on alternative emissions compilations were used in
CarboScope and MIROC. Since this is known to result directly in lower total $CO_2$ uptake in
atmospheric inversions (Gaubert et al., 2018;Peylin et al., 2013) we adjusted the land sink of each
inversion estimate (where most of the emissions occur) by its fossil fuel difference to the CAMS
model. These differences amount to as much as 0.7 GtC for certain years (CarboScope inversion
region NH) and are thus an important consideration in an inverse flux comparison.
The land/ocean $CO_2$ fluxes from atmospheric inversions contain anthropogenic perturbation and
natural pre-industrial $CO_2$ fluxes. Natural pre-industrial fluxes are land $CO_2$ sinks corresponding to
carbon transported to ocean by rivers. These land $CO_2$ sinks are compensated over the globe by
ocean $CO_2$ sources corresponding to the outgassing of riverine carbon inputs to the ocean. We
apply the distribution of land $CO_2$ fluxes in three latitude bands using estimates from Resplandy et
al. (2018), which are constrained by ocean heat transport to a total sink of 0.78 GtC $y^{-1}$. The
latitude distribution of river-induced ocean $CO_2$ sources are derived from a simulation of the IPSL
GOBM using as an input the river flux constrained by heat transport of Resplandy et al. (2018). We
adjusted the land/ocean fluxes per latitude band based on these results.
The atmospheric inversions are now evaluated using vertical profiles of atmospheric $CO_2$
concentrations (Fig. B3). More than 50 aircraft programs over the globe, either regular or
occasional, have been used in order to draw a robust picture of the model performance but the
space-time data coverage is irregular, denser around 2009 or in the 0-45°N latitude band.  The
four models are compared to independent $CO_2$ measurements made onboard aircraft over many
places of the world between 1 and 7 km above sea level, between 2008 and 2016. Results are
shown in Fig. B3 and discussed in Section 3.1.3.
**2.7     Processes not included in the global carbon budget**
The contribution of anthropogenic CO and $CH_4$ to the global carbon budget has been partly
neglected in Eq. 1 and is described in Sect. 2.7.1. The contributions of other carbonates to $CO_2$
emissions is described in Sect. 2.7.2. The contribution of anthropogenic changes in river fluxes is
conceptually included in Eq. 1 in $S_{OCEAN}$ and in $S_{LAND}$, but it is not represented in the process
models used to quantify these fluxes. This effect is discussed in Sect. 2.7.3. Similarly, the loss of
additional sink capacity from reduced forest cover is missing in the combination of approaches
used here to estimate both land fluxes ($E_{LUC}$ and $S_{LAND}$) and its potential effect is discussed and
quantified in Sect. 2.7.4.
**2.7.1     Contribution of anthropogenic CO and $CH_4$ to the global carbon budget**
Equation (1) includes only partly the net input of $CO_2$ to the atmosphere from the chemical
oxidation of reactive carbon-containing gases from sources other than the combustion of fossil
fuels, such as: (1) cement process emissions, since these do not come from combustion of fossil
fuels, (2) the oxidation of fossil fuels, (3) the assumption of immediate oxidation of vented
methane in oil production. It omits however any other anthropogenic carbon-containing gases
that are eventually oxidised in the atmosphere, such as anthropogenic emissions of CO and $CH_4$.
An attempt is made in this section to estimate their magnitude, and identify the sources of
uncertainty. Anthropogenic CO emissions are from incomplete fossil fuel and biofuel burning and
deforestation fires. The main anthropogenic emissions of fossil $CH_4$ that matter for the global
carbon budget are the fugitive emissions of coal, oil and gas upstream sectors (see below). These
emissions of CO and $CH_4$ contribute a net addition of fossil carbon to the atmosphere.
In our estimate of $E_{FF}$ we assumed (Sect. 2.1.1) that all the fuel burned is emitted as $CO_2$, thus CO
anthropogenic emissions associated with incomplete combustion and their atmospheric oxidation
into $CO_2$ within a few months are already counted implicitly in $E_{FF}$ and should not be counted
twice (same for $E_{LUC}$ and anthropogenic CO emissions by deforestation fires). Anthropogenic
emissions of fossil $CH_4$ are not included in $E_{FF}$, because these fugitive emissions are not included in
the fuel inventories. Yet they contribute to the annual $CO_2$ growth rate after $CH_4$ gets oxidized into
$CO_2$. Anthropogenic emissions of fossil $CH_4$ represent 15% of total $CH_4$ emissions (Kirschke et al.,
2013) that is 0.061 GtC $yr^{-1}$ for the past decade. Assuming steady state, these emissions are all
converted to $CO_2$ by OH oxidation, and thus explain 0.06 GtC $yr^{-1}$ of the global $CO_2$ growth rate in
the past decade, or 0.07-0.1 GtC $yr^{-1}$ using the higher $CH_4$ emissions reported recently (Schwietzke
et al., 2016).
Other anthropogenic changes in the sources of CO and $CH_4$ from wildfires, vegetation biomass,
wetlands, ruminants or permafrost changes are similarly assumed to have a small effect on the
$CO_2$ growth rate. The $CH_4$ emissions and sinks are published and analysed separately in the Global
Methane Budget publication that follows a similar approach as presented here (Saunois et al.,

23   2016).

**2.7.2   Contribution of other carbonates to $CO_2$ emissions**
The contribution of fossil carbonates other than cement production is not systematically included
in estimates of $E_{FF}$, except at the national level where they are accounted in the UNFCCC national
inventories. The missing processes include $CO_2$ emissions associated with the calcination of lime
and limestone outside cement production, and the reabsorption of $CO_2$ by the rocks and concrete
from carbonation through their life time (Xi et al., 2016). Carbonates are used in various
industries, including in iron and steel manufacture and in agriculture. They are found naturally in

some coals. Carbonation from cement life-cycle, including demolition and crushing, was estimated by one study to be around 0.25 GtC $yr^{-1}$ for year 2013 (Xi et al., 2016). Carbonation emissions from cement life-cycle would offset calcination emissions from lime and limestone production. The balance of these two processes is not clear.

### 2.7.3 Anthropogenic carbon fluxes in the land to ocean aquatic continuum

The approach used to determine the global carbon budget refers to the mean, variations, and trends in the perturbation of $CO_2$ in the atmosphere, referenced to the preindustrial era. Carbon is continuously displaced from the land to the ocean through the land-ocean aquatic continuum (LOAC) comprising freshwaters, estuaries and coastal areas (Bauer et al., 2013;Regnier et al., 2013). A significant fraction of this lateral carbon flux is entirely 'natural' and is thus a steady state component of the preindustrial carbon cycle. We account for this preindustrial flux where appropriate in our study. However, changes in environmental conditions and land use change have caused an increase in the lateral transport of carbon into the LOAC – a perturbation that is relevant for the global carbon budget presented here.

The results of the analysis of Regnier et al. (2013) can be summarized in two points of relevance for the anthropogenic $CO_2$ budget. First, the anthropogenic perturbation has increased the organic carbon export from terrestrial ecosystems to the hydrosphere at a rate of 1.0 ± 0.5 GtC $yr^{-1}$, mainly owing to enhanced carbon export from soils. Second, this exported anthropogenic carbon is partly respired through the LOAC, partly sequestered in sediments along the LOAC and to a lesser extent, transferred to the open ocean where it may accumulate. The increase in storage of land-derived organic carbon in the LOAC and open ocean combined is estimated by Regnier et al. (2013) at 0.65 ± 0.35GtC $yr^{-1}$. We do not attempt to incorporate the changes in LOAC in our study.

The inclusion of freshwater fluxes of anthropogenic $CO_2$ affects the estimates of, and partitioning between, $S_{LAND}$ and $S_{OCEAN}$ in Eq. (1), but does not affect the other terms. This effect is not included in the GOBMs and DGVMs used in our global carbon budget analysis presented here.

### 2.7.4 Loss of additional sink capacity

Historical land-cover change was dominated by transitions from vegetation types that can provide a large sink per area unit (typically, forests) to others less efficient in removing $CO_2$ from the atmosphere (typically, croplands). The resultant decrease in land sink, called the 'loss of sink

capacity', is calculated as the difference between the actual land sink under changing land-cover
and the counter-factual land sink under preindustrial land-cover. An efficient protocol has yet to
be designed to estimate the magnitude of the loss of additional sink capacity in DGVMs. Here, we
provide a quantitative estimate of this term to be used in the discussion. Our estimate uses the
compact Earth system model OSCAR whose land carbon cycle component is designed to emulate
the behaviour of DGMVs (Gasser et al., 2017). We use OSCAR v2.2.1 (an update of v2.2 with minor
changes) in a probabilistic setup identical to the one of Arneth et al. (2017) but with a Monte Carlo
ensemble of 2000 simulations. For each, we calculate separately $S_{LAND}$ and the loss of additional
sink capacity. We then constrain the ensemble by weighting each member to obtain a distribution
of cumulative $S_{LAND}$ over 1850-2005 close to the DGVMs used here. From this ensemble, we
estimate a loss of additional sink capacity of 0.4 ± 0.3 GtC yr$^{-1}$ on average over 2005-2014, and 20
± 15 GtC accumulated between 1870 and 2017 (using a linear extrapolation of the trend to
estimate the last few years).
**3    Results**
**3.1    Global carbon budget mean and variability for 1959 – 2017**
The global carbon budget averaged over the last half-century is shown in Fig. 3. For this time
period, 82% of the total emissions ($E_{FF}$ + $E_{LUC}$) were caused by fossil $CO_2$ emissions, and 18% by
land-use change. The total emissions were partitioned among the atmosphere (45%), ocean (24%)
and land (30%). All components except land-use change emissions have grown since 1959, with
important interannual variability in the growth rate in atmospheric $CO_2$ concentration and in the
land $CO_2$ sink (Fig. 4), and some decadal variability in all terms (Table 6). Differences with previous
budget releases is documented in Fig. B4.
**3.1.1    $CO_2$ emissions**
Global fossil $CO_2$ emissions have increased every decade from an average of 3.1 ± 0.2 GtC yr$^{-1}$ in
the 1960s to an average of 9.4 ± 0.5 GtC yr$^{-1}$ during 2008-2017 (Table 6, Fig. 2 and Fig. 5). The
growth rate in these emissions decreased between the 1960s and the 1990s, from 4.5% yr$^{-1}$ in the
1960s (1960-1969), 2.8% yr$^{-1}$ in the 1970s (1970-1979), 1.9% yr$^{-1}$ in the 1980s (1980-1989), and to
1.0% yr$^{-1}$ in the 1990s (1990-1999). After this period, the growth rate began increasing again in the
2000s at an average growth rate of 3.2% yr$^{-1}$, decreasing to 1.5% yr$^{-1}$ for the last decade (2008-
2017), with a 3-year period of no or low growth during 2014-2016 (Fig. 5).
In contrast, $CO_2$ emissions from land use, land-use change and forestry have remained relatively
constant, at around $1.3 \pm 0.7$ GtC $yr^{-1}$ over the past half-century but with large spread across
estimates (Fig. 6). These emissions are also relatively constant in the DGVM ensemble of models,
except during the last decade when they increase to $1.9 \pm 0.7$ GtC $yr^{-1}$. However, there is no
agreement on this recent increase between the two bookkeeping models, each suggesting an
opposite trend (Fig. 6).
**3.1.2   Partitioning among the atmosphere, ocean and land**
The growth rate in atmospheric $CO_2$ level increased from $1.7 \pm 0.07$ GtC $yr^{-1}$ in the 1960s to $4.7 \pm$
$0.02$ GtC $yr^{-1}$ during 2008-2017 with important decadal variations (Table 6 and Fig. 2). Both ocean
and land $CO_2$ sinks increased roughly in line with the atmospheric increase, but with significant
decadal variability on land (Table 6), and possibly in the ocean (Fig. 7).
The ocean $CO_2$ sink increased from $1.0 \pm 0.5$ GtC $yr^{-1}$ in the 1960s to $2.4 \pm 0.5$ GtC $yr^{-1}$ during 2008-
2017, with interannual variations of the order of a few tenths of GtC $yr^{-1}$ generally showing an
increased ocean sink during large El Niño events (i.e. 1997-1998) (Fig. 7; Rödenbeck et al., 2014).
Although there is some coherence among the GOBMs and $pCO_2$-based flux products regarding the
mean, there is poor agreement for interannual variability and the ocean models underestimate
decadal variability (Sect. 2.4.3 and Fig. 7; DeVries et al. (2017)).
The terrestrial $CO_2$ sink increased from $1.2 \pm 0.5$ GtC $yr^{-1}$ in the 1960s to $3.2 \pm 0.7$ GtC $yr^{-1}$ during
2008-2017, with important interannual variations of up to 2 GtC $yr^{-1}$ generally showing a
decreased land sink during El Niño events (Fig. 6), responsible for the corresponding enhanced
growth rate in atmospheric $CO_2$ concentration. The larger land $CO_2$ sink during 2008-2017
compared to the 1960s is reproduced by all the DGVMs in response to the combined atmospheric
$CO_2$ increase and changes in climate, and consistent with constraints from the other budget terms
(Table 5).
Estimates of total atmosphere-to-land fluxes ($S_{LAND} - E_{LUC}$) from the DGVMs are consistent with
the budget constraints (Table 5), except during 2008-2017, where the DGVM ensemble estimates
a total atmosphere-to-land flux of $1.3 \pm 0.5$ GtC $yr^{-1}$, likely below the budget constraints of $2.1 \pm$
$0.7$ GtC $yr^{-1}$ and outside the range of the inversions (Table 5). This comparison suggests that the
DGVMs could overestimate $E_{LUC}$ emissions and/or underestimate the terrestrial sink $S_{LAND}$ during
the last decade.

### 3.1.3 Model evaluation

The evaluation of ocean estimates (Fig. B1) shows a relative interannual mismatch of 15% and 17% for the two $pCO_2$-based flux products over the globe, relative to the $pCO_2$ observations from the SOCAT v6 database for the period 1985-2017. A 0% mismatch would indicate a perfect model, and a field with no interannual variability would result in a 100% mismatch. A larger than 100% mismatch is possible when the method produces a larger mismatch than the benchmark field with no interannual variability (see section 2.4.3). This mismatch by the $pCO_2$-based flux products is improved compared with earlier published versions of these two flux products of around 20-25% for the 1992-2009 time period (Rödenbeck et al. 2015), likely because of the larger data availability after 2009. The GOBMs show a global relative interannual mismatch between 50% and 60%, with one model at 94% and one at 193%. The GOBM mismatch is of the same order as the mismatch calculated in an ensemble of 14 flux products, but larger than the two flux products used in this report (Fig. 5 in Rödenbeck et al. 2015). The mismatch is generally larger at high latitudes compared to the tropics, for both the flux products and the GOBMs. The two flux products have similar mismatch of around 10-15% in the tropics, around 25% in the north, and 30-55% in the south. The GOBM mismatch is more spread across regions, ranging from 29% to 178% in the tropics, 70% to 192% in the North, and 108% to 304% in the South. The higher mismatch occurs in regions with stronger climate variability, such as the northern and southern high-latitudes (poleward of the subtropical gyres) and the equatorial Pacific. The latter is also apparent in the model mismatch, but is hidden in Figure B1 due to the averaging over 30°S to 30°N (see also section 4).

The evaluation of the DGVMs (Fig. B2) shows generally high skill scores across models for runoff, and to a lesser extent for vegetation biomass, GPP, and ecosystem respiration (Fig. B2, left panel). Skill score was lowest for leaf area index and net ecosystem exchange, with a widest disparity among models for soil carbon. Further analysis of the results will be provided separately, focusing the strengths and weaknesses in the DGVM ensemble and its validity for use in the global carbon budget.

The evaluation of the atmospheric inversions (Fig. B3) shows long-term mean biases in the free troposphere better than 0.8 ppm in absolute values for each product. CAMS and CTE biases show some dependency on latitude (a trend of -0.0018 ± 0.0005 and 0.0043 ± 0.0004 ppm per degree for CAMS and CTE, respectively). These latitude-dependent biases may reveal biases in the surface

fluxes (e.g., Houweling et al., 2015) but the link is not straight-forward and will be analysed
separately. The biases for MIROC and CarboScope behave similarly together in relative values, but
they are less regular than the two other products, which hampers the interpretation. Lesser
model performance for specific aircraft programs, like for the four-year Discover-AQ campaign in
continental US (https://discover-aq.larc.nasa.gov/), contributes to this variability.
**3.1.4  Budget imbalance**
The carbon budget imbalance ($B_{IM}$; Eq. 1) quantifies the mismatch between the estimated total
emissions and the estimated changes in the atmosphere, land and ocean reservoirs. The mean
budget imbalance from 1959 to 2017 is small (0.14 GtC $yr^{-1}$) and shows no trend over the full time
series. The process models (GOBMs and DGVMs) have been selected to match observational
constraints in the 1990s but no further constraints have been applied to their representation of
trend and variability. Therefore, the near-zero mean and trend in the budget imbalance is an
indirect evidence of a coherent community understanding of the emissions and their partitioning
on those time scales (Fig. 4). However, the budget imbalance shows substantial variability of the
order of ± 1 GtC $yr^{-1}$, particularly over semi-decadal time scales, although most of the variability is
within the uncertainty of the estimates. The positive carbon imbalance during the 1960s, early
1990s, and in the last decade, suggest that either the emissions were overestimated or the sinks
were underestimated during these periods. The reverse is true for the 1970s and around 1995-
2000 (Fig. 4).
We cannot attribute the cause of the variability in the budget imbalance with our analysis, only to
note that the budget imbalance is unlikely to be explained by errors or biases in the emissions
alone because of its large semi-decadal variability component, a variability that is untypical of
emissions and has not changed in the past 50 years in spite of a nearly tripling in emissions (Fig.
4). Errors in $S_{LAND}$ and $S_{OCEAN}$ are more likely to be the main cause for the budget imbalance. For
example, underestimation of the $S_{LAND}$ by DGVMs has been reported following the eruption of
Mount Pinatubo in 1991 possibly due to missing responses to changes in diffuse radiation
(Mercado et al., 2009) or other yet unknown factor, and DGVMs are suspected to overestimate
the land sink in response to the wet decade of the 1970s (Sitch et al., 2008). Decadal and semi-
decadal variability in the ocean sink has been also reported recently (DeVries et al.,
2017;Landschützer et al., 2015), with the $pCO_2$-based ocean flux products suggesting a smaller
than expected ocean $CO_2$ sink in the 1990s and a larger than expected sink in the 2000s (Fig. 7),
possibly caused by changes in ocean circulation (DeVries et al., 2017) not captured in coarse
resolution GOBMs used here (Dufour et al., 2013). The absence of internal variability could also be
at fault. Internal variability is not captured by single realizations of coarse resolution model
simulations (Li and Ilyina, 2017), and is thought to be largest in regions with strong seasonal and
interannual climate variability, i.e. the high latitude ocean regions (poleward of the subtropical
gyres) and the equatorial Pacific (McKinley et al., 2016). Some of these errors could be driven by
errors in the climatic forcing data, particularly precipitation (for $S_{LAND}$) and wind (for $S_{OCEAN}$) rather
than in the models.
**3.2    Global carbon budget for the last decade (2008 – 2017)**
The global carbon budget averaged over the last decade (2008-2017) is shown in Fig. 2 and Fig. 9.
For this time period, 87% of the total emissions ($E_{FF}$ + $E_{LUC}$) were from fossil $CO_2$ emissions ($E_{FF}$),
and 13% from land-use change ($E_{LUC}$). The total emissions were partitioned among the
atmosphere (44%), ocean (22%) and land (29%), with a remaining unattributed budget imbalance

14  (5%).

**3.2.1    $CO_2$ emissions**
Global fossil $CO_2$ emissions grew at a rate of 1.5% yr$^{-1}$ for the last decade (2008-2017). China's
emissions increased by +3.0% yr$^{-1}$ on average (increasing by +0.64 GtC yr$^{-1}$ during the 10-year
period) dominating the global trends, followed by India's emissions increase by +5.2% yr$^{-1}$
(increasing by +0.25 GtC yr$^{-1}$), while emissions decreased in EU28 by −1.8% yr$^{-1}$ (decreasing by −
0.17 GtC yr$^{-1}$), and in the USA by 0.9% yr$^{-1}$ (decreasing by −0.18 GtC yr$^{-1}$). In the past decade, fossil
$CO_2$ emissions decreased significantly (at the 95% level) in 25 countries: Aruba, Barbados, Croatia,
Czech Republic, North Korea, Denmark, France, Greece, Greenland, Iceland, Ireland, Malta,
Netherlands, Romania, Slovakia, Slovenia, Sweden, Switzerland, Syria, Trinidad and Tobago,
Ukraine, United Kingdom, USA, Uzbekistan and Venezuela. Notable was Germany, whose
emissions did not decrease significantly.
In contrast, there is no apparent trend in $CO_2$ emissions from land-use change (Fig. 6), though the
data are very uncertain, with the two bookkeeping estimates showing opposite trends over the
last decade. Larger emissions are expected increasingly over time for DGVM-based estimates as
they include the loss of additional sink capacity, while the bookkeeping estimates don't. The LUH2
dataset also features large dynamics in land use in particular in the tropics in recent years, causing
higher emissions in DGVMs and BLUE than in H&N.

### 3.2.2    Partitioning among the atmosphere, ocean and land

The growth rate in atmospheric $CO_2$ concentration increased during 2008-2017, in contrast to
more constant levels the previous decade and reflecting a similar decrease in the land sink
compared to an increase in the previous decade, albeit with large interannual variability (Fig. 4).
During the same period, the ocean $CO_2$ sink appears to have intensified, an effect which is
particularly apparent in the $pCO_2$-based flux products (Fig. 7) and is thought to originate at least in
part in the Southern Ocean (Landschützer et al., 2015).
The budget imbalance (Table 6) and the residual sink from global budget (Table 5) include an error
term due to the inconsistency that arises from using $E_{LUC}$ from bookkeeping models, but $S_{LAND}$
from DGVMs. This error term includes the fundamental differences between bookkeeping models
and DGVMs, most notably the loss of additional sink capacity. Other differences include: an
incomplete accounting of LUC practices and processes in DGVMs, while they are all accounted for
in bookkeeping models by using observed carbon densities, and bookkeeping error of keeping
present-day carbon densities fixed in the past. That the budget imbalance shows no clear trend
towards larger values over time is an indication that the loss of additional sink capacity plays a
minor role compared to other errors in $S_{LAND}$ or $S_{OCEAN}$ (discussed in 3.1.4).

### 3.2.3    Regional distribution

Fig. 8 shows the partitioning of the total atmosphere-to-surface fluxes excluding fossil $CO_2$
emissions ($S_{LAND} + S_{OCEAN} - E_{LUC}$) according to the multi-model average of the process models in the
ocean and on land (GOBMs and DGVMs), and to the atmospheric inversions. Fig. 8 provides
information on the regional distribution of those fluxes by latitude bands. The global mean total
atmosphere-to-surface $CO_2$ fluxes from process models for 2008-2017 is 3.7 ± 1.2 GtC yr$^{-1}$. This is
below but still within the uncertainty range of a global mean atmosphere-to-surface flux of 4.6 ±
0.5 GtC yr$^{-1}$ inferred from the carbon budget ($E_{FF} - G_{ATM}$ in Equation 1; Table 6). The total
atmosphere-to-surface $CO_2$ fluxes from the four inversions are very similar, ranging from 4.7 to
5.0 GtC yr$^{-1}$, consistent with the carbon budget as expected from the constraints on the inversions
and the adjustments to the same $E_{FF}$ distribution (See Section 2.6).
In the south (south of 30°S), the atmospheric inversions suggest an atmosphere-to-surface flux for
2008-2017 around 1.6-1.7 GtC yr$^{-1}$, close to the process models' estimate of 1.4 ± 0.7 GtC yr$^{-1}$ (Fig.
8). The interannual variability in the south is low because of the dominance of ocean area with
low variability compared to land areas. The split between land ($S_{LAND}$-$E_{LUC}$) and ocean ($S_{OCEAN}$)
shows a small contribution to variability in the south coming from the land, with no consistency
between the DGVMs and the inversions or among inversions. This is expected due to the difficulty
of separating exactly the land and oceanic fluxes when viewed from atmospheric observations
alone. The oceanic variability in the south is estimated to be significant in the two flux products
and in at least one of the inversions, with decadal variability of around 0.5 GtC yr$^{-1}$. The GOBMs do
not reproduce this variability.
In the tropics (30°S-30°N), both the atmospheric inversions and process models suggest the total
carbon balance in this region is close to neutral on average over the past decade, with
atmosphere-to-surface fluxes for the 2008-2017 average ranging between –0.4 and +0.4 GtC yr$^{-1}$.
The agreement between inversions and models is significantly better for the last decade than for
any previous decade, although the reasons for this better agreement are still unclear. Both the
process models and the inversions consistently allocate more year-to-year variability of $CO_2$ fluxes
to the tropics compared to the north (north of 30°N; Fig. 8). The split between the land and ocean
indicates the land is the origin of most of the tropical variability, consistently among models (both
for the land and for the ocean) and inversions. The oceanic variability in the tropics is similar
among models and with the two ocean flux products, reflected in their lower observational
mismatch (Section 3.1.3). While the inversions indicate that atmosphere-to-land $CO_2$ fluxes are
more variable than atmosphere-to-ocean $CO_2$ fluxes in the tropics, the correspondence between
the inversions and the ocean flux products or GOBMs is much poorer.
In the north (north of 30°N), the inversions and process models show less agreement on the
magnitude of the atmosphere-to-land flux, with the ensemble mean of the process models
suggesting a total Northern Hemisphere sink for 2008-2017 of 2.2 ± 0.6 GtC yr$^{-1}$, likely below the
estimates from the inversions ranging from 2.6 to 3.6 GtC yr$^{-1}$ (Fig. 8). The discrepancy in the
north-tropics distribution of $CO_2$ fluxes between the inversions and models arises from the
differences in mean fluxes over the northern land. This discrepancy is also evidenced over the
previous decade and highlights not only persistent issues with the quantification of the drivers of
the net land $CO_2$ flux (Arneth et al., 2017;Huntzinger et al., 2017) but also the distribution of
atmosphere-to-land fluxes between the tropics and higher latitudes that is particularly marked in
previous decades, as highlighted previously (Stephens et al., 2007;Baccini et al., 2017;Schimel et
al., 2015).
Differences between inversions may be related for example to differences in their
interhemispheric transport, and other inversion settings (Table A3). Separate analysis has shown
that the influence of the chosen prior land and ocean fluxes is minor compared to other aspects of
each inversion. In comparison to the previous global carbon budget publication, the fossil fuel
inputs were adjusted to match that of $E_{FF}$ used in this analysis (see Section 2.6), therefore
removing differences due to fossil emissions prior. Differences between inversions and the
ensemble of process models in the north cannot be simply explained. They could either reflect a
bias in the inversions or missing processes or biases in the process models, such as the lack of
adequate parameterizations for forest management in the north and for forest degradation
emissions in the tropics for the DGVMs. The estimated contribution of the north and its
uncertainty from process models is sensitive both to the ensemble of process models used and to
the specifics of each inversion.
Resolving the differences in the Northern Hemisphere land sink will require the consideration and
inclusion of larger volumes of semi-continuous observations from tall towers close to the surface
$CO_2$ exchange. Some of this data is becoming available, but not used in the current inverse models
sometimes due to the short records, and sometimes because the coarse transport models cannot
adequately represent these time series. Improvements in model resolution and atmospheric
transport realism together with expansion of the observational record (also in the data sparse
Boreal Eurasian area) will help anchor the mid-latitude fluxes per continent. In addition, new
metrics could potentially differentiate between the more- and less realistic realisations of the
Northern Hemisphere land sink shown in Fig. 8.
**3.2.4   Budget imbalance**
The budget imbalance was +0.5 GtC $yr^{-1}$ on average over 2008-2017. Although the uncertainties
are large in each term, the sustained imbalance over this last decade suggests an overestimation
of the emissions and/or an underestimation of the sinks. An origin in the land and/or ocean sink
may be more likely, given the large variability of the land sink and the suspected underestimation
of decadal variability in the ocean sink. An underestimate of $S_{LAND}$ would also reconcile model
results with inversions estimates for fluxes in the total land during the past decade (Fig. 8; Table
5). However, we cannot exclude that the budget imbalance over the last decade could partly be
due to an overestimation of $CO_2$ emissions from land-use change, given their large uncertainty, as
has been suggested elsewhere (Piao et al., 2018). More integrated use of observations in the
Global Carbon Budget, either on their own or for further constraining model results, should help
resolve some of the budget imbalance (Peters et al. 2017; Section 4).
**3.3    Global carbon budget for year 2017**
**3.3.1    $CO_2$ emissions**
Preliminary estimates of global fossil $CO_2$ emissions based on BP energy statistics are for emissions
growing by 1.6% between 2016 and 2017 to 9.9 ± 0.5 GtC in 2017 (Fig. 5), distributed among coal
(40%), oil (35%), gas (20%), cement (4%) and gas flaring (0.7%). Compared to the previous year,
emissions from coal increased by 1.6%, while emissions from oil, gas, and cement increased by
1.7%, 3.0%, and 1.2%, respectively. All growth rates presented are adjusted for the leap year,
unless stated otherwise.
The growth in emissions of 1.6% in 2017 is within the range of the projected growth of 2.0%
(range of 0.8 to 3.0%) published in Le Quéré et al. (2018) based on national emissions projections
for China, the USA, and India and projections of gross domestic product corrected for $I_{FF}$ trends for
the rest of the world. The growth in emissions in 2017 for China, UEA, and the rest of the world is
also within their previously projected range, while the growth in India was slightly above the
projection (Table 7).
In 2017, the largest absolute contributions to global $CO_2$ emissions were from China (27%), the
USA (15%), the EU (28-member states; 10%), India (7%), while the rest of the world contributed
42%. The percentages are the fraction of the global emissions including bunker fuels (3.1%). These
four regions account for 59% of global $CO_2$ emissions. Growth rates for these countries from 2016
to 2017 were +1.5% (China), –0.5% (USA), +1.2% (EU28), and +3.9% (India), with +1.9% for the rest
of the world. The per-capita $CO_2$ emissions in 2017 were 1.1 tC person$^{-1}$ yr$^{-1}$ for the globe, and
were 4.4 (USA), 2.0 (China), 1.9 (EU28) and 0.5 (India) tC person$^{-1}$ yr$^{-1}$ for the four highest emitting
countries (Fig. 5).
In 2016 (the last year available), the largest absolute contributions to global $CO_2$ emissions from a
consumption perspective were China (25%), USA (16%), the EU (12%), and India (6%). The

difference between territorial and consumption emissions (the net emission transfer via international trade) has generally increased from 1990 to around 2005 and remained relatively stable afterwards until the last year available (2016; Fig. 5).

The global $CO_2$ emissions from land-use change are estimated as 1.4 ± 0.7 GtC in 2017, close to the previous decade but with low confidence in the annual change. This brings the total $CO_2$ emissions from fossil plus land-use change ($E_{FF}+E_{LUC}$) to 11.3 ± 0.9 GtC (41.2 ± 3 $GtCO_2$).

### 3.3.2 Partitioning among the atmosphere, ocean and land

The growth rate in atmospheric $CO_2$ concentration was 4.6 ± 0.2 GtC in 2017 (2.16 ± 0.09 ppm; Fig. 4; Dlugokencky and Tans, 2018). This is near the 2008-2017 average of 4.7 ± 0.1 GtC $yr^{-1}$ and reflects the return to normal conditions after the El Niño of 2015-2016.

The estimated ocean $CO_2$ sink was 2.5 ± 0.5 GtC in 2017. All models and data products estimate a small reduction or no change in the sink (average of 0.1, ranging from +0.02 to -0.4 GtC), consistent with the return to normal conditions after the El Niño which caused an enhanced sink in previous years (Fig. 7).

The terrestrial $CO_2$ sink from the model ensemble was 3.8 ± 0.8 GtC in 2017, above the decadal average (Fig. 4) and consistent with constraints from the rest of the budget (Table 5).

The budget imbalance was +0.3 GtC in 2017, indicating, as for the last decade, a small overestimation of the emissions and/or underestimation of the sinks for that year. This imbalance is indicative only, given the large uncertainties in the estimation of the $B_{IM}$.

### 3.4 Global carbon budget projection for year 2018

### 3.4.1 $CO_2$ emissions

Based on available data as of 7 November 2018 (see Sect. 2.1.5), fossil $CO_2$ emissions ($E_{FF}$) for 2018 are projected to increase by +2.7% (range of 1.8% to +3.7%; Table 7). Our method contains several assumptions that could influence the estimate beyond the given range, and as such, it has an indicative value only. Within the given assumptions, global emissions would be 10.1 ± 0.5 GtC (37.1 ± 1.8 $GtCO_2$) in 2018. The interpretation of the 2018 emissions projection is provided elsewhere (Figueres et al., 2018;Jackson et al., 2018a).

For China, the expected change is for an increase in emissions of +4.7% (range of +2.0% to +7.4%)
in 2018 compared to 2017. This is based on estimated growth in coal (+4.5%; the main fuel source
in China), oil (+3.6%), natural gas (+17.7%) consumption, and cement production (+1.0%). The
uncertainty range considers the variations in the difference between preliminary January-
September data and final full-year data, the uncertainty in the preliminary data used for the 2017
base, and uncertainty in the evolution of energy density and carbon content of coal. See also Liu
et al. (2018) for further analysis of China's projected emissions.
For the USA, the EIA emissions projection for 2018 combined with cement data from USGS gives
an increase of 2.5 % (range of +0.5 to +4.5 %) compared to 2017.
For the European Union, our projection for 2018 is for a decrease of –0.7% (range of –2.6% to
+1.3%) over 2017. This is based on estimates for coal of –1.2%, oil of +1.2%, gas of –2.9%, and
stable cement emissions.
For India, our projection for 2018 is for an increase of +6.3% (range of 4.3% to +8.3%) over 2017.
This is based on separate projections for coal (+7.1%), oil (+2.9%), gas (+6.0%) and cement

15  (+13.4%).

For the rest of the world, the expected growth for 2018 is +1.8% (range of +0.5% to +3.0%). This is
computed using the GDP projection for the world excluding China, USA, EU, and India, of 2.8%
made by the IMF (IMF, 2018) and a decrease in $I_{FF}$ of –1.0% $yr^{-1}$ which is the average from 2008-
2017. The uncertainty range is based on the standard deviation of the interannual variability in $I_{FF}$
during 2008-2017 of ±0.7% $yr^{-1}$ and our estimate of uncertainty in the IMF's GDP forecast of

21  ±0.5%.

Preliminary estimate of fire emissions in deforestation zones indicate that emissions from land-
use change ($E_{LUC}$) for 2018 were below average until October, and are expected to range between
0.1 and 0.2 lower than the 2008-2017 average. We therefore expect $E_{LUC}$ emissions of around 1.2
GtC in 2018, for a total $CO_2$ emissions of 11.3 ± 0.9 GtC (41.5 ± 3 $GtCO_2$).
**3.4.2   Partitioning among the atmosphere, ocean and land**
The 2018 growth in atmospheric $CO_2$ concentration ($G_{ATM}$) is projected to be 4.9 ± 0.7 GtC (2.3 ±
0.3 ppm) based on MLO observations until the end of August 2018, bringing the atmospheric $CO_2$
concentration to an expected level of 407 ppm averaged over the year. Combining projected $E_{FF}$,

$E_{LUC}$ and $G_{ATM}$ suggests a combined land and ocean sink ($S_{LAND}$ + $S_{OCEAN}$) of about 6.5 GtC for 2018. Although each term has large uncertainty, the oceanic sink $S_{OCEAN}$ has generally low interannual variability and is likely to remain close to its 2017 value of around 2.5 GtC, leaving a rough estimated land sink $S_{LAND}$ of around 4.0 GtC. If realised, it would be among the largest $S_{LAND}$ over the historical period. However, the possible onset of an El Niño at the end of 2018 could reduce $S_{LAND}$, with $G_{ATM}$ returning to high growth rate towards the end of the year.

### 3.5   Cumulative sources and sinks

Cumulative historical sources and sinks are estimated as in Eq. (1) with semi-independent estimates for each term and a global carbon budget imbalance. Cumulative fossil $CO_2$ emissions for 1870-2017 were 425 ± 20 GtC for $E_{FF}$ and 190± 75 GtC for $E_{LUC}$ (Table 8; Fig. 9), for a total of 615 ± 80 GtC. The cumulative emissions from $E_{LUC}$ are particularly uncertain, with large spread among individual estimates of 135 GtC (Houghton) and 240 GtC (BLUE) for the two bookkeeping models and a similar wide estimate of 180± 75 GtC for the DGVMs. These estimates are consistent with indirect constraints from vegetation biomass observations (Li et al., 2017), but given the large spread a best estimate is difficult to ascertain.

Emissions were partitioned among the atmosphere (250 ± 5 GtC), ocean (150 ± 20 GtC), and the land (190 ± 50 GtC). The use of nearly independent estimates for the individual terms shows a cumulative budget imbalance of 25 GtC during 1870-2017 (Fig. 2), which, if correct, suggests emissions are too high by the same proportion or the land or ocean sinks are underestimated. The bulk of the imbalance is likely to originate largely from the large estimation of $E_{LUC}$ between the mid 1920s and the mid 1960s which is unmatched by a growth in atmospheric $CO_2$ concentration as recorded in ice cores (Fig. 3). The known loss of additional sink capacity of about 20 GtC due to reduced forest cover has not been accounted in our method and would further exacerbate the budget imbalance (Section 2.7.4).

Cumulative emissions through to year 2018 increase to 625 ± 80 GtC (2290 ± 290 GtCO$_2$), with about 70% contribution from $E_{FF}$ and about 30% contribution from $E_{LUC}$. Cumulative emissions and their partitioning for different periods are provided in Table 8.

Given the large and persistent uncertainties in cumulative emissions, we suggest extreme caution is needed if using cumulative emission estimate to determine the "remaining carbon budget" to stay below given temperature limit (Rogelj et al., 2016). We suggest estimating the remaining

carbon budget by integrating scenario data from the current time to some time in the future
(Millar et al., 2017).
**4    Discussion**
Each year when the global carbon budget is published, each flux component is updated for all
previous years to consider corrections that are the result of further scrutiny and verification of the
underlying data in the primary input data sets. Annual estimates may improve with improvements
in data quality and timeliness (e.g. to eliminate need for extrapolation of forcing data such as land
use). Of the various terms in the global budget, only the fossil $CO_2$ emissions and the growth rate
in atmospheric $CO_2$ concentration are based primarily on empirical inputs supporting annual
estimates in this carbon budget. Although it is an imperfect measure, the carbon budget
imbalance provides a strong indication of the limitations in observations, in understanding or full
representation of processes in models, and/or in the integration of the carbon budget
components.
The persistent unexplained variability in the carbon budget imbalance limits our ability to verify
reported emissions (Peters et al., 2017) and suggests we do not yet have a complete
understanding of the underlying carbon cycle processes. Resolving most of this unexplained
variability should be possible through different and complementary approaches. First, as intended
with our annual updates, the imbalance as an error term is reduced by improvements of individual
components of the global carbon budget that follow from improving the underlying data and
statistics and by improving the models through the resolution of some of the key uncertainties
detailed in Table 9. Second, additional clues to the origin and processes responsible for the
current imbalance could be obtained through a closer scrutiny of carbon variability in light of
other Earth system data (e.g. heat balance, water balance), and the use of a wider range of
biogeochemical observations to better understand the land/over partitioning of the carbon
imbalance (e.g. oxygen, carbon isotopes). Finally, additional information could also be obtained
through higher resolution and process knowledge at the regional level, and through the
introduction of inferred fluxes such as those based on satellite $CO_2$ retrievals. The limit of the
resolution of the carbon budget imbalance is yet unclear, but most certainly not yet reached given
the possibilities for improvements that lie ahead.

The assessment of the GOBMs used for $S_{OCEAN}$ with flux products based on observations highlights substantial discrepancy at mid and high latitudes. Given the good data coverage of $pCO_2$ observations in the Northern Hemisphere (Bakker et al., 2016), this discrepancy points at an underestimation of variability in the GOBMs globally and consequently, the variability in $S_{OCEAN}$ appears to be underestimated. The size of this underestimate (order of 0.5 GtC yr$^{-1}$) could account for some of the budget imbalance, but not all. Increasing model resolution or incorporating internal variability (Li and Ilyina, 2017) have been suggested as ways to increase model variability (Section 3.1.4).

The assessment of the net land-atmosphere exchange derived from land sink and net land use change flux with atmospheric inversions also shows substantial discrepancy, particularly for the estimate of the total land flux over the northern extra-tropics in the past decade. This discrepancy highlights the difficulty to quantify complex processes ($CO_2$ fertilisation, nitrogen deposition, climate change and variability, land management, etc.) that collectively determine the net land $CO_2$ flux. Resolving the differences in the Northern Hemisphere land sink will require the consideration and inclusion of larger volumes of observations (Section 3.2.3).

Estimates of $E_{LUC}$ suffer from a range of intertwined issues, including the poor quality of historical land-cover and land-use change maps, the rudimentary representation of management processes in most models, and the confusion in methodologies and boundary conditions used across methods (e.g. Pongratz et al., 2014, Arneth et al. 2017, and Section 2.7.4 on the loss of sink capacity). Uncertainties in current and historical carbon stocks in soils and vegetation also add uncertainty in the LUC flux estimates. Unless a major effort to resolve these issues is made, little progress is expected in the resolution of $E_{LUC}$. This is particularly concerning given the growing important of $E_{LUC}$ for climate mitigation strategies, and the large issues in the quantification of the cumulative emissions over the historical period that arise from large uncertainties in $E_{LUC}$.

To move towards the resolution of the carbon budget imbalance, this year we have introduced metrics for the evaluation of the ocean and land models and atmospheric inversions. These metrics expand the use of observations in the global carbon budget, helping 1) to support improvements in the ocean and land carbon models that produce the sink estimates, and 2) to constrain the representation of key underlying processes in the models and to allocate the regional partitioning of the $CO_2$ fluxes. This is an initial step towards the introduction of a broader

range of observations that we hope will support continued improvements in the annual estimates of the global carbon budget.

We assessed elsewhere (Peters et al. 2017) that a sustained decrease of –1% in global emissions could be detected at the 66% likelihood level after a decade only. Similarly, a change in behaviour of the land and/or ocean carbon sink would take as long to detect, and much longer if it emerges more slowly. Reducing the carbon imbalance, regionalising the carbon budget, and integrating multiple variables are powerful ways to shorten the detection limit and ensure the research community can rapidly identify growing issues of concern in the evolution of the global carbon cycle under the current rapid and unprecedented changing environmental conditions.

**5    Data availability**

The data presented here are made available in the belief that their wide dissemination will lead to greater understanding and new scientific insights of how the carbon cycle works, how humans are altering it, and how we can mitigate the resulting human-driven climate change. The free availability of these data does not constitute permission for publication of the data. For research projects, if the data are essential to the work, or if an important result or conclusion depends on the data, co-authorship may need to be considered. Full contact details and information on how to cite the data included in the GCP (2018) release are given at the top of each page in the accompanying database and summarised in Table 2.

The accompanying database includes two Excel files organised in the following spreadsheets (accessible with the free viewer https://support.microsoft.com/en-gb/help/273711/how-to-obtain-the-latest-excel-viewer):

File Global_Carbon_Budget_2018v1.0.xlsx includes the following:

1. Summary
2. The global carbon budget (1959-2017);
3. Global $CO_2$ emissions from fossil fuels and cement production by fuel type, and the per-capita emissions (1959-2017);
4. $CO_2$ emissions from land-use change from the individual methods and models (1959-2017);
5. Ocean $CO_2$ sink from the individual ocean models and $pCO_2$-based products (1959-2017);
6. Terrestrial $CO_2$ sink from the DGVMs (1959-2017);

7. Additional information on the carbon balance prior to 1959 (1750-2017).

File National_Carbon_Emissions_2018v1.0.xlsx includes the following:

1. Summary

2. Territorial country $CO_2$ emissions from fossil $CO_2$ emissions (1959-2017) from CDIAC, extended to 2016 using BP data;

3. Territorial country $CO_2$ emissions from fossil $CO_2$ emissions (1959-2017) from CDIAC with UNFCCC data overwritten where available, extended to 2017 using BP data;

4. Consumption country $CO_2$ emissions from fossil $CO_2$ emissions and emissions transfer from the international trade of goods and services (1990-2016) using CDIAC/UNFCCC data (worksheet 3 above) as reference;

5. Emissions transfers (Consumption minus territorial emissions; 1990-2016);

6. Country definitions;

7. Details of disaggregated countries;

8. Details of aggregated countries.

National emissions data are also available from the Global Carbon Atlas (globalcarbonatlas.org).

**6 Conclusions**

The estimation of global $CO_2$ emissions and sinks is a major effort by the carbon cycle research community that requires a careful compilation and synthesis of measurements, statistical estimates and model results. The delivery of an annual carbon budget serves two purposes. First, there is a large demand for up-to-date information on the state of the anthropogenic perturbation of the climate system and its underpinning causes. A broad stakeholder community relies on the data sets associated with the annual carbon budget including scientists, policy makers, businesses, journalists, and non-governmental organizations engaged in adapting to and mitigating human-driven climate change. Second, over the last decade we have seen unprecedented changes in the human and biophysical environments (e.g. changes in the growth of fossil fuel emissions, Earth's temperatures, and strength of the carbon sinks), which call for frequent assessments of the state of the planet, a growing understanding, and an improved capacity to anticipate the evolution of the carbon cycle in the future. Building this scientific understanding to meet the extraordinary climate mitigation challenge requires frequent, robust, and transparent data sets and methods

that can be scrutinized and replicated. This paper via 'living data' helps to keep track of new
budget updates.
**Competing interests.** The authors declare that they have no conflict of interest.
**Acknowledgements.** We thank all people and institutions who provided the data used in this
carbon budget; Richard Betts, Erik Buitenhuis, Jinfeng Chang, S. Shu and Naomi Smith for their
involvement in the development, use and analysis of the models and data-products used here;
and Fortunat Joos, Samar Khatiwala and Timothy DeVries for providing historical data. We thanks
Zhu Liu and Bo Zheng for their insights on the China emissions projections and its uncertainty, and
Rob Jackson and the Global Carbon Project members for their input throughout the development
of this update. We thank Ed Dlugokencky for providing atmospheric $CO_2$ measurements; Camilla
Stegen Landa, Christophe Bernard, and Steve Jones of the Bjerknes Climate Data Centre and the
ICOS Ocean Thematic Centre data management at the University of Bergen, who helped with
gathering information from the SOCAT community; and V. Kitidis, P.M.S. Monteiro, L. Gregor, M.
Gonzáles-Dávila, M. Santana Casino, R. Negri and X. A. Padin who contributed to the provision of
ocean $pCO_2$ observations (see Table A1). This is NOAA-PMEL contribution number 4847. We thank
the institutions and funding agencies responsible for the collection and quality control of the data
included in SOCAT, and the support of the International Ocean Carbon Coordination Project
(IOCCP), the Surface Ocean Lower Atmosphere Study (SOLAS), and the Integrated Marine
Biogeochemistry, Ecosystem Research (IMBER) programme. We thank FAO and its member
countries for the collection and free dissemination of data relevant to this work. We thank data
providers to ObsPack GLOBALVIEWplus v1.0 and NRT v3.0 for atmospheric $CO_2$ observations used
in CTE2016-FT, and the following people for sharing their aircraft data used in Fig. B3: T. Machida,
G. Chen, S. Wofsy, K. Davis, J. DiGangi, J. Peischl, R. B. Ryerson, B. Stephens, C. Sweeney, K.
McKain, and L.G. Gatti; University of Colorado/CIRES for the NOAA WP-3D aircraft vertical profile
data; and Japan Meteorological Agency. We thank the individuals and institutions that provided
the databases used for the model evaluations introduced here, and Nigel Hawtin for producing
Figure 2.

1    Finally, we thank all funders who have supported the individual and joint contributions to this

2    work (see Table A5), as well as the reviewers of this manuscript and previous versions, and the

3    many researchers who have provided feedback.

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

**Tables**
**Table 1.** Factors used to convert carbon in various units (by convention, Unit 1 = Unit 2
conversion).

| Unit 1 | Unit 2 | Conversion | Source |
|---|---|---|---|
| GtC (gigatonnes of carbon) | ppm (parts per million)[a] | 2.124[b] | Ballantyne et al. (2012) |
| GtC (gigatonnes of carbon) | PgC (petagrams of carbon) | 1 | SI unit conversion |
| $GtCO_2$ (gigatonnes of carbon dioxide) | GtC (gigatonnes of carbon) | 3.664 | 44.01/12.011 in mass equivalent |
| GtC (gigatonnes of carbon) | MtC (megatonnes of carbon) | 1000 | SI unit conversion |

[a] Measurements of atmospheric $CO_2$ concentration have units of dry-air mole fraction. 'ppm' is an
abbreviation for micromole/mol, dry air.
[b]The use of a factor of 2.124 assumes that all the atmosphere is well mixed within one year. In
reality, only the troposphere is well mixed and the growth rate of $CO_2$ concentration in the less
well-mixed stratosphere is not measured by sites from the NOAA network. Using a factor of 2.124
makes the approximation that the growth rate of $CO_2$ concentration in the stratosphere equals
that of the troposphere on a yearly basis.

1 **Table 2.** How to cite the individual components of the global carbon budget presented here.

| Component | Primary reference |
|---|---|
| Global fossil $CO_2$ emissions ($E_{FF}$), total and by fuel type | Boden et al., (2017) |
| National territorial fossil $CO_2$ emissions ($E_{FF}$) | CDIAC source: Boden et al., (2017) <br><br> UNFCCC (2018) |
| National consumption-based fossil $CO_2$ emissions ($E_{FF}$) by country (consumption) | Peters et al. (2011b) updated as described in this paper |
| Land-use change emissions ($E_{LUC}$) | Average from Houghton and Nassikas (2017) and Hansis et al., (2015), both updated as described in this paper |
| Growth rate in atmospheric $CO_2$ concentration ($G_{ATM}$) | Dlugokencky and Tans (2018) |
| Ocean and land $CO_2$ sinks ($S_{OCEAN}$ and $S_{LAND}$) | This paper for $S_{OCEAN}$ and $S_{LAND}$ and references in Table 4 for individual models. |

1 **Table 3.** Main methodological changes in the global carbon budget since first publication. Methodological changes introduced in one year are
2 kept for the following years unless noted. Empty cells mean there were no methodological changes introduced that year.

| Publication year[a] | Fossil fuel emissions | | | LUC emissions | Reservoirs | | | Uncertainty & other changes |
|---|---|---|---|---|---|---|---|---|
| | Global | Country (territorial) | Country (consumption) | | Atmosphere | Ocean | Land | |
| 2006 Raupach et al. (2007) | | Split in regions | | | | | | |
| 2007 Canadell et al. (2007) | | | | $E_{LUC}$ based on FAO-FRA 2005; constant $E_{LUC}$ for 2006 | 1959-1979 data from Mauna Loa; data after 1980 from global average | Based on one ocean model tuned to reproduced observed 1990s sink | | ±1σ provided for all components |
| 2008 (online) 2009 Le Quéré et al. (2009) | | Split between Annex B and non-Annex B | Results from an independent study discussed | Constant $E_{LUC}$ for 2007 Fire-based emission anomalies used for 2006-2008 | | Based on four ocean models normalised to observations with constant delta | First use of five DGVMs to compare with budget residual | |
| 2010 Friedlingstein et al. (2010) | Projection for current year based on GDP | Emissions for top emitters | | $E_{LUC}$ updated with FAO-FRA 2010 | | | | |
| 2011 Peters et al. (2012b) | | | Split between Annex B and non-Annex B | | | | | |
| 2012 Le Quéré et al. (2013) Peters et al. (2013) | | 129 countries from 1959 | 129 countries and regions from 1990-2010 based on GTAP8.0 | $E_{LUC}$ for 1997-2011 includes interannual anomalies from fire-based emissions | All years from global average | Based on 5 ocean models normalised to observations with ratio | Ten DGVMs available for $S_{LAND}$; First use of four models to compare with $E_{LUC}$ | |
| 2013 Le Quéré et al. (2014) | | 250 countries[b] | 134 countries and regions 1990-2011 based on GTAP8.1, with detailed estimates for years 1997, 2001, 2004, and 2007 | $E_{LUC}$ for 2012 estimated from 2001-2010 average | | Based on six models compared with two data-products to year 2011 | Coordinated DGVM experiments for $S_{LAND}$ and $E_{LUC}$ | Confidence levels; cumulative emissions; budget from 1750 |
| 2014 Le Quéré et al. (2015b) | Three years of BP data | Three years of BP data | Extended to 2012 with updated GDP data | $E_{LUC}$ for 1997-2013 includes interannual anomalies from fire-based emissions | | Based on seven models | Based on ten models | Inclusion of breakdown of the sinks in three latitude bands and comparison with three atmospheric inversions |
| 2015 Le Quéré et al. (2015a) Jackson et al. (2016) | Projection for current year based Jan-Aug data | National emissions from UNFCCC extended to 2014 also provided | Detailed estimates introduced for 2011 based on GTAP9 | | | Based on eight models | Based on ten models with assessment of minimum realism | The decadal uncertainty for the DGVM ensemble mean now uses ±1σ of the decadal spread across models |
| 2016 Le Quéré et al. (2016) | Two years of BP data | Added three small countries; CHN emissions from 1990 from BP data (this release only) | | Preliminary $E_{LUC}$ using FRA-2015 shown for comparison; use of five DGVMs | | Based on seven models | Based on fourteen models | Discussion of projection for full budget for current year |
| 2017 Le Quéré et al. (2018) | Projection includes India-specific data | | | Average of two bookkeeping models; use of twelve DGVMs | | Based on eight models that match the observed sink for the 1990s; no longer normalised | Based on fifteen models that meet observation-based criteria (see Sect. 2.5) | Land multi-model average now used in main carbon budget, with the carbon imbalance presented separately; new table of key uncertainties |
| 2018 (this study) | Revision in cement | Aggregation of overseas territories | | Use of sixteen DGVMs[c] | Use of four atmospheric | Based on seven models | Based on sixteen models; revised atmospheric | Introduction of metrics for evaluation of individual |

| | emissions; Projection includes EU-specific data | into governing nations for total of 213 countries[b] | inversions | forcing from CRUNCEP to CRU-JRA-55 | models using observations |
|---|---|---|---|---|---|

[a]The naming convention of the budgets has changed. Up to and including 2010, the budget year (Carbon Budget 2010) represented the latest year of the data. From 2012,
the budget year (Carbon Budget 2012) refers to the initial publication year.
[b]The CDIAC database has about 250 countries, but we show data for 213 countries since we aggregate and disaggregate some countries to be consistent with current
country definitions (see Sect. 2.1.1 for more details)
[c]$E_{LUC}$ is still estimated based on bookkeeping models as in 2017, but the number of DGVMs used to characterise the uncertainty has changed.

1 **Table 4.** References for the process models, $pCO_2$-based ocean flux products, and atmospheric
2 inversions included in Figs. 6-8. All models and products are updated with new data to end of year
3 2017, and the atmospheric forcing for the DGVMs has been updated as described in Section 2.2.2.

| Model/data name | Reference | Change from Le Quéré et al. (2018) |
|---|---|---|
| *Bookkeeping models for land-use change emissions* | | |
| BLUE | Hansis et al. (2015) | LUH2 rangelands were treated differently, using the static LUH2 information on forest/non-forest grid-cells to determine clearing for rangelands. Additionally effects on degradation of primary to secondary lands due to rangelands on natural (uncleared) vegetation were added to BLUE. |
| H&N2017 | Houghton and Nassikas (2017) | No change. |
| *Dynamic global vegetation models[a]* | | |
| CABLE-POP | Haverd et al. (2018) | Simple crop harvest and grazing implemented. Small adjustments to photosynthesis parameters to compensate for effect of new climate forcing on GPP. |
| CLASS-CTEM | Melton and Arora (2016) | 20 soil layers used. Soil depth is prescribed following Pelletier et al. (2016). |
| CLM5.0 | Oleson et al. (2013) | No change. |
| DLEM | Tian et al. (2015) | Using observed irrigation data instead of a potential irrigation map. |
| ISAM | Meiyappan et al. (2015) | Crop harvest and N fertilizer application as described in Song et al (2016). |
| JSBACH | Mauritsen et al. (In review) | New version of JSBACH (JSBACH 3.2), as used for CMIP6 simulations. Changes include a new fire algorithm, as well as new processes (land nitrogen cycle, carbon storage of wood products). Furthermore, LUH2 rangelands were treated differently, using the static LUH2 information on forest/non-forest grid-cells to determine clearing for rangelands. |
| JULES | Clarke et al. (2011) | No Change. |
| LPJ-GUESS | Smith et al. (2014)[b] | No Change. |
| LPJ | Poulter et al. (2011)[c] | Uses monthly litter update (previously annual), 3 product pools for deforestation flux, shifting cultivation, wood harvest, and inclusion of boreal needleleaf deciduous PFT. |
| LPX-Bern | Lienert and Joos (2018) | Minor refinement of parameterization. Changed from 1x1 degree to 0.5x0.5 degree resolution. Nitrogen deposition and fertilization from NMIP. |
| OCN | Zaehle and Friend (2010) | No change (uses r294). |
| ORCHIDEE-Trunk | Krinner et al. (2005)[d] | Updated soil water stress and albedo scheme; overall C-cycle optimisation (gross fluxes). |
| ORCHIDEE-CNP | Goll et al. (2017) | First time contribution (ORCHIDEE with nitrogen and phosphorus dynamics). |
| SDGVM | Walker et al. (2017) | No change. |

| | | |
|---|---|---|
| SURFEXv8 | Joetzjer et al. (2015) | Not applicable (not used in 2017). |
| VISIT | Kato et al. (2013) | Updated spinup protocol. |

*Global ocean biogeochemistry models*

| | | |
|---|---|---|
| CCSM-BEC | Doney et al. (2009) | No change. |
| MICOM-HAMOCC (NorESM-OC) | Schwinger et al. (2016) | No drift correction. |
| MITgcm-REcoM2 | Hauck et al. (2016) | No change. |
| MPIOM-HAMOCC | Mauritsen et al. (In review) | Change of atmospheric forcing; cmip6 model version including modifications and bug-fixes in HAMOCC and MPIOM. |
| NEMO-PISCES (CNRM) | Berthet et al. (Submitted) | New model version with update to NEMOv3.6 and improved gas exchange. |
| NEMO-PISCES (IPSL) | Aumont and Bopp (2006) | No change. |
| NEMO-PlankTOM5 | Buitenhuis et al. (2010)[e] | No change. |

*pCO$_2$-based flux ocean products*

| | | |
|---|---|---|
| Landschützer | Landschützer et al. (2016) | No change. |
| Jena CarboScope | Rödenbeck et al. (2014) | No change. |

*Atmospheric inversions*

| | | |
|---|---|---|
| CAMS | Chevallier et al. (2005) | No change. |
| CarbonTracker Europe (CTE) | van der Laan-Luijkx et al. (2017) | Minor changes in the inversion set up. |
| Jena CarboScope | Rödenbeck et al. (2003) | No change. |
| MIROC | Saeki and Patra (2017) | Not applicable (not used in 2017). |

[a] The forcing for all DGVMs has been updated from CRUNCEP to CRUJRA.
[b] To account for the differences between the derivation of shortwave radiation (SWRAD) from CRU cloudiness and SWRAD from CRU-JRA-55, the photosythesis scaling parameter $\alpha_a$ was modified (-15%) to yield similar results.
[c] Compared to published version, decreased LPJ wood harvest efficiency so that 50% of biomass was removed off-site compared to 85% used in the 2012 budget. Residue management of managed grasslands increased so that 100% of harvested grass enters the litter pool.
[d] Compared to published version, new hydrology and snow scheme; revised parameter values for photosynthetic capacity for all ecosystem (following assimilation of FLUXNET data), updated parameters values for stem allocation, maintenance respiration and biomass export for tropical forests (based on literature) and, $CO_2$ down-regulation process added to photosynthesis. Version used for CMIP6.
[e] With no nutrient restoring below the mixed layer depth.

**Table 5.** Comparison of results from the bookkeeping method and budget residuals with results from the DGVMs and inverse estimates for
different periods, last decade, and last year available. All values are in GtC yr$^{-1}$. The DGVM uncertainties represent ±1σ of the decadal or annual
(for 2017 only) estimates from the individual DGVMs: for the inverse models all three results are given where available.

| | Mean (GtC yr$^{-1}$) | | | | | | |
|---|---|---|---|---|---|---|---|
| | 1960-1969 | 1970-1979 | 1980-1989 | 1990-1999 | 2000-2009 | 2008-2017 | 2017 |
| *Land-use change emissions ($E_{LUC}$)* | | | | | | | |
| Bookkeeping methods | 1.5 ± 0.7 | 1.2 ± 0.7 | 1.2 ± 0.7 | 1.4 ± 0.7 | 1.3 ± 0.7 | 1.5 ± 0.7 | 1.4 ± 0.7 |
| DGVMs | 1.5 ± 0.7 | 1.4 ± 0.7 | 1.5 ± 0.7 | 1.3 ± 0.6 | 1.4 ± 0.6 | 1.9 ± 0.6 | 2.0 ± 0.7 |
| *Terrestrial sink ($S_{LAND}$)* | | | | | | | |
| Residual sink from global budget ($E_{FF}+E_{LUC}-G_{ATM}-S_{OCEAN}$) | 1.8 ± 0.9 | 1.8 ± 0.9 | 1.5 ± 0.9 | 2.6 ± 0.9 | 2.9 ± 0.9 | 3.5 ± 1.0 | 4.1 ± 1.0 |
| DGVMs | 1.2 ± 0.5 | 2.1 ± 0.4 | 1.8 ± 0.6 | 2.4 ± 0.5 | 2.7 ± 0.7 | 3.2 ± 0.7 | 3.8 ± 0.8 |
| *Total land fluxes ($S_{LAND} - E_{LUC}$)* | | | | | | | |
| Budget constraint ($E_{FF}-G_{ATM}-S_{OCEAN}$) | 0.3 ± 0.5 | 0.6 ± 0.6 | 0.4 ± 0.6 | 1.2 ± 0.6 | 1.6 ± 0.6 | 2.1 ± 0.7 | 2.7 ± 0.7 |
| DGVMs | -0.3 ± 0.6 | 0.7 ± 0.5 | 0.3 ± 0.6 | 1.1 ± 0.5 | 1.3 ± 0.5 | 1.3 ± 0.5 | 1.8 ± 0.5 |
| Inversions* | —/—/— | —/—/— | -0.2–0.1 | 0.5–1.1 | 0.8–1.5 | 1.4–2.4 | 1.2–3.1 |

*Estimates are corrected for the pre-industrial influence of river fluxes and adjusted to common $E_{FF}$ (Sect. 2.7.2). Two inversions are available for the 1980s and 1990s. Two
additional inversions are available from 2001 and used from the decade of the 2000 (Tables A3).
**Table 6.** Decadal mean in the five components of the anthropogenic $CO_2$ budget for different periods, and last year available. All values are in
GtC yr$^{-1}$, and uncertainties are reported as ±1σ. The table also shows the budget imbalance ($B_{IM}$), which provides a measure of the
discrepancies among the nearly independent estimates and has an uncertainty exceeding ± 1 GtC yr$^{-1}$. A positive imbalance means the
emissions are overestimated and/or the sinks are too small.

| | Mean (GtC yr$^{-1}$) | | | | | | |
|---|---|---|---|---|---|---|---|
| | 1960-1969 | 1970-1979 | 1980-1989 | 1990-1999 | 2000-2009 | 2008-2017 | 2017 |
| *Total emissions ($E_{FF}+E_{LUC}$)* | | | | | | | |
| Fossil $CO_2$ emissions ($E_{FF}$) | 3.1 ± 0.2 | 4.7 ± 0.2 | 5.4 ± 0.3 | 6.3 ± 0.3 | 7.8 ± 0.4 | 9.4 ± 0.5 | 9.9 ± 0.5 |
| Land-use change emissions ($E_{LUC}$) | 1.5 ± 0.7 | 1.2 ± 0.7 | 1.2 ± 0.7 | 1.4 ± 0.7 | 1.3 ± 0.7 | 1.5 ± 0.7 | 1.4 ± 0.7 |
| Total emissions | 4.7 ± 0.7 | 5.8 ± 0.7 | 6.6 ± 0.8 | 7.6 ± 0.8 | 9.0 ± 0.8 | 10.8 ± 0.8 | 11.3 ± 0.9 |
| *Partitioning* | | | | | | | |
| Growth rate in atmospheric $CO_2$ concentration ($G_{ATM}$) | 1.7 ± 0.07 | 2.8 ± 0.07 | 3.4 ± 0.02 | 3.1 ± 0.02 | 4.0 ± 0.02 | 4.7 ± 0.02 | 4.6 ± 0.2 |
| Ocean sink ($S_{OCEAN}$) | 1.0 ± 0.5 | 1.3 ± 0.5 | 1.7 ± 0.5 | 2.0 ± 0.5 | 2.1 ± 0.5 | 2.4 ± 0.5 | 2.5 ± 0.5 |
| Terrestrial sink ($S_{LAND}$) | 1.2 ± 0.5 | 2.1 ± 0.4 | 1.8 ± 0.6 | 2.4 ± 0.5 | 2.7 ± 0.7 | 3.2 ± 0.7 | 3.8 ± 0.8 |
| *Budget imbalance* | | | | | | | |
| $B_{IM} = E_{FF}+E_{LUC} - (G_{ATM}+S_{OCEAN}+S_{LAND})$ | (0.6) | (−0.3) | (−0.3) | (0.2) | (0.2) | (0.5) | (0.3) |

**Table 7.** Comparison of the projection with realised fossil $CO_2$ emissions ($E_{FF}$). The 'Actual' values are first estimate available using actual data, and the 'Projected' values refers to estimate made before the end of the year for each publication. Projections based on a different method from that described here during 2008-2014 are available in Le Quéré et al., (2016). All values are adjusted for leap years.

| | World | | China | | USA | | EU28 | | India | | Rest of World | |
|---|---|---|---|---|---|---|---|---|---|---|---|---|
| | Projected | Actual | Projected | Actual | Projected | Actual | Projected | Actual | Projected | Actual | Projected | Actual |
| 2015[a] | −0.6% (−1.6 to 0.5) | 0.06% | −3.9% (−4.6 to −1.1) | −0.7% | −1.5% (−5.5 to 0.3) | −2.5% | – | – | – | – | 1.2% (−0.2 to 2.6) | +1.2% |
| 2016[b] | −0.2% (−1.0 to +1.8) | 0.2% | −0.5% (−3.8 to +1.3) | −0.3% | −1.7% (−4.0 to +0.6) | −2.1% | – | – | – | – | +1.0% (−0.4 to +2.5) | +1.3% |
| 2017[c] | +2.0% (+0.8 to +3.0) | +1.6% | +3.5 (+0.7 to +5.4) | +1.5% | −0.4% (−2.7 to +1.0) | −0.5% | – | – | +2.0% (+0.2 to +3.8) | +3.9% | +1.6% (0.0 to +3.2) | +1.9% |
| 2018[d] | +2.7% (+1.8 to +3.7) | – | +4.7 (+2.0 to +7.4) | – | +2.5% (+0.5 to +4.5) | – | -0.7% (-2.6 to +1.3) | – | +6.3% (+4.3 to +8.3) | – | +1.8% (+0.5 to +3.0) | – |

[a]Jackson et al. (2016) and Le Quéré et al. (2015a). [b]Le Quéré et al. (2016). [c]Le Quéré et al. (2018). [d]This study.

**Table 8.** Cumulative $CO_2$ for different time periods in gigatonnes of carbon (GtC). All uncertainties are reported as $\pm 1\sigma$. $E_{LUC}$ and $S_{OCEAN}$ have been revised to incorporate multiple estimates (Section 3.5), and the terrestrial sink ($S_{LAND}$) is now estimated independently, from the mean of the DGVM. Therefore the table also shows the budget imbalance, which provides a measure of the discrepancies among the nearly independent estimates. Its uncertainty exceeds $\pm$ 60 GtC. The method used here does not capture the loss of additional sink capacity from reduced forest cover, which is about 20 GtC and would exacerbate the budget imbalance (see Section 2.7.3). All values are rounded to the nearest 5 GtC and therefore columns do not necessarily add to zero.

| Units of GtC | 1750-2017 | 1850-2005 | 1850-2014 | 1959-2017 | 1870-2017 | 1870-2018[a] |
|---|---|---|---|---|---|---|
| *Emissions* | | | | | | |
| Fossil $CO_2$ emissions ($E_{FF}$) | 430 ± 20 | 320 ± 15 | 400 ± 20 | 350 ± 20 | 425 ± 20 | 435 ± 20 |
| Land-use change $CO_2$ emissions ($E_{LUC}$) | 235 ± 95 | 185 ± 70 | 195 ± 75 | 80 ± 40 | 190 ± 75 | 190 ± 75 |
| Total emissions | 660 ± 95 | 500 ± 75 | 595 ± 80 | 430 ± 45 | 615 ± 80 | 625 ± 80 |
| *Partitioning* | | | | | | |
| Growth rate in atmospheric $CO_2$ concentration ($G_{ATM}$) | 275 ± 5 | 200 ± 5 | 235 ± 5 | 190 ± 5 | 250 ± 5 | 255 ± 5 |
| Ocean sink ($S_{OCEAN}$) | 165 ± 20 | 125 ± 20[b] | 150 ± 20 | 100 ± 20 | 150 ± 20 | 155 ± 20 |
| Terrestrial sink ($S_{LAND}$) | 215 ± 50 | 160 ± 45 | 185 ± 50 | 130 ± 30 | 190 ± 50 | 195 ± 50 |
| *Budget imbalance* | | | | | | |
| $B_{IM} = E_{FF} + E_{LUC} - (G_{ATM} + S_{OCEAN} + S_{LAND})$ | (5) | (20) | (25) | (10) | (25) | (25) |

[a]Using projections for year 2018 (Sect. 3.3).

[b]This value was incorrectly reported as 145 in Le Quéré et al. (2018).

**Table 9.** Major known sources of uncertainties in each component of the Global Carbon Budget,
defined as input data or processes that have a demonstrated effect of at least ±0.3 GtC yr[-1].

| Source of uncertainty | Time scale (years) | Location | Status | Evidence |
|---|---|---|---|---|
| Fossil $CO_2$ emissions ($E_{FF}$; Section 2.1) | | | | |
| energy statistics | annual to decadal | mainly China | see Sect. 2.1 | (Korsbakken et al., 2016) |
| carbon content of coal | decadal | mainly China | see Sect. 2.1 | (Liu et al., 2015) |
| Emissions from land-use change ($E_{LUC}$; section 2.2) | | | | |
| land-cover and land-use change statistics | continuous | global; in particular tropics | see Sect. 2.2 | (Houghton et al., 2012) |
| sub-grid-scale transitions | annual to decadal | global | see Table A1 | (Wilkenskjeld et al., 2014) |
| vegetation biomass | annual to decadal | global; in particular tropics | see Table A1 | (Houghton et al., 2012) |
| wood and crop harvest | annual to decadal | global; SE Asia | see Table A1 | (Arneth et al., 2017) |
| peat burning[a] | multi-decadal trend | global | see Table A1 | (van der Werf et al., 2010) |
| loss of additional sink capacity | multi-decadal trend | global | not included; Section 2.7.3 | (Gitz and Ciais, 2003) |
| Atmospheric growth rate ($G_{ATM}$) → no demonstrated uncertainties larger than ±0.3 GtC yr[-1, b] | | | | |
| Ocean sink ($S_{OCEAN}$) | | | | |
| variability in oceanic circulation[c] | semi-decadal to decadal | global; in particular Southern Ocean | see Sect. 2.4.2 | (DeVries et al., 2017) |
| Internal variability | annual to decadal | high latitudes; Equatorial Pacific | no ensembles/ coarse resolution | (McKinley et al., 2016) |
| anthropogenic changes in nutrient supply | multi-decadal trend | global | not included | (Duce et al., 2008) |
| Land sink ($S_{LAND}$) | | | | |
| strength of $CO_2$ fertilisation | multi-decadal trend | global | see Sect. 2.5 | (Wenzel et al., 2016) |
| response to variability in temperature and rainfall | annual to decadal | global; in particular tropics | see Sect. 2.5 | (Cox et al., 2013) |
| nutrient limitation and supply | multi-decadal trend | global | see Sect. 2.5 | (Zaehle et al., 2011) |
| response to diffuse radiation | annual | global | see Sect. 2.5 | (Mercado et al., 2009) |

[a]As result of interactions between land-use and climate
[b]The uncertainties in $G_{ATM}$ have been estimated as ±0.2 GtC yr[-1], although the conversion of the growth rate into a
global annual flux assuming instantaneous mixing throughout the atmosphere introduces additional errors that have
not yet been quantified.
[c]Could in part be due to uncertainties in atmospheric forcing (Swart et al., 2014)

**Appendix A.** Supplementary tables.

**Table A1.** Comparison of the processes included (Y) or not (N) in the bookkeeping and Dynamic Global Vegetation Models for their estimates of $E_{LUC}$ and $S_{LAND}$. See Table 4 for model references. All models include deforestation and forest regrowth after abandonment of agriculture (or from afforestation activities on agricultural land).

| | bookkeeping models | | DGVMs | | | | | | | | | | | | | | | |
|---|---|---|---|---|---|---|---|---|---|---|---|---|---|---|---|---|---|---|
| | H&N2017 | BLUE | CABLE-POP | CLASS-CTEM | CLM5.0 | DLEM | ISAM | JSBACH[j] | JULES | LPJ-GUESS[j] | LPJ | LPX-Bern | OCN | Orchidee-CNP | Orchidee-Trunk | SDGVM | SURFEX | VISIT[i] |
| **Processes relevant for $E_{LUC}$** | | | | | | | | | | | | | | | | | | |
| Wood harvest and forest degradation[a] | Y | Y | Y | N | Y | Y | Y | Y | N | Y | Y | N[d] | Y | N | Y | N | N | Y |
| Shifting cultivation / subgrid scale transitions | N[b] | Y | Y | N | Y | N | N | Y | N | Y | Y | N[d] | N | N | N | N | N | Y |
| Cropland harvest (removed, r, or added to litter, l) | Y(r)[h] | Y(r)[h] | Y(r) | Y(l) | Y(r) | Y | Y | Y(r,l) | N | Y(r) | Y(l) | Y(r) | Y(r,l) | Y(r) | Y(r) | Y(r) | N | Y(r) |
| Peat fires | Y | Y | N | N | Y | N | N | N | N | N | N | N | N | N | N | N | N | N |
| Fire as a management tool | Y[h] | Y[h] | N | N | N | N | N | N | N | N | N | N | N | N | N | N | N | N |
| N fertilization | Y[h] | Y[h] | N | N | Y | Y | Y | N | N | Y | N | Y | Y | Y | N | N | N | N |
| Tillage | Y[h] | Y[h] | Y | Y[e] | N | N | N | N | N | Y | N | N | N | Y[g] | N | N | N | N |
| Irrigation | Y[h] | Y[h] | N | N | Y | Y | Y | N | N | Y | N | N | N | N | N | N | Y[g] | N |
| Wetland drainage | Y[h] | Y[h] | N | N | N | N | N | N | N | N | N | N | N | N | N | N | N | N |
| Erosion | Y[h] | Y[h] | N | N | N | N | N | N | N | N | N | N | N | N | N | N | N | Y |
| Southeaast Asia peat drainage | Y | Y | N | N | N | N | N | N | N | N | N | N | N | N | N | N | N | N |
| Grazing and mowing harvest (removed, r, or added to litter, l) | Y(r)[h] | Y(r)[h] | Y(r) | N | N | N | Y(l) | Y(l) | N | Y(r) | Y(l) | N | Y(r,l) | N | N | N | N | N |
| **Processes relevant also for $S_{LAND}$** | | | | | | | | | | | | | | | | | | |
| Fire simulation | US only | N | N | Y | Y | Y | N | Y | N | Y | Y | Y | N | N | N | Y | Y | Y |
| Climate and variability | N | N | Y | Y | Y | Y | Y | Y | Y | Y | Y | Y | Y | Y | Y | Y | Y | Y |
| CO$_2$ fertilisation | N[f] | N[f] | Y | Y | Y | Y | Y | Y | Y | Y | Y | Y | Y | Y | Y[e] | Y | Y | Y |
| Carbon-nitrogen interactions, including N deposition | N[h] | N[h] | Y | N[d] | Y | Y | Y | Y | N | Y | N | Y | Y | Y | N | Y[c] | N[i] | N |

[a] Refers to the routine harvest of established managed forests rather than pools of harvested products.

[b] No back- and forth-transitions between vegetation types at the country-level, but if forest loss based on FRA exceeded agricultural expansion based on FAO, then this amount of area is interpreted as shifting cultivation.

[c] Limited. Nitrogen uptake is simulated as a function of soil C, and photosynthesis is directly related to canopy N. Does not consider N deposition.

[d] Although C-N cycle interactions are not represented, the model includes a parameterization of down-regulation of photosynthesis as CO$_2$ increases to emulate nutrient constraints (Arora et al., 2009)

[e] Tillage is represented over croplands by increased soil carbon decomposition rate and reduced humification of litter to soil carbon.

[f] Bookkeeping models include effect of CO$_2$-fertilization as captured by observed carbon densities, but not as an effect transient in time.

1    [g] 20% reduction of active soil organic carbon (SOC) pool turnover time for C3 crop and 40% reduction for C4 crops

2    [h] Process captured implicitly by use of observed carbon densities.

3    [i] Simple parameterization of nitrogen limitation based on Yin (2002; assessed on FACE experiments).

1 **Table A2.** Comparison of the processes and model set up for the Global Ocean Biogeochemistry
2 Models for their estimates of $S_{OCEAN}$. See Table 4 for model references.

| | CCSM-BEC | NorESM-OC | MITgcm-REcoM2 | MPIOM-HAMOCC | NEMO3.6-PISCESv2-gas (CNRM) | NEMO-PISCES (IPSL) | NEMO-PlankTOM5 |
|---|---|---|---|---|---|---|---|
| Atmospheric forcing | NCEP | CORE-I (spin up) / NCEP with CORE-II corrections | JRA55 | NCEP / NCEP+ERA-20C (spin-up) | NCEP | NCEP | NCEP |
| Initialisation of carbon chemistry | GLODAP | GLODAP v1 + spin up 1000 years | GLODAP, then spin-up 116 years (2 cycles JRA55) | spin-up with ERA20C | GLODAPv2 + 300 years online | GLODAP from 1948 onwards | GLODAP + spin up 30 years |
| Physical ocean model | POP Version *1.4*.3 | MICOM | MITgcm 65n | MPIOM | NEMOv3.6-GELATOv6-eORCA1L75 | NEMOv3.2-ORCA2L31 | NEMOv2.3-ORCA2 |
| Resolution | 3.6° lon, 0.8 to 1.8° lat | 1° lon, 0.17 to 0.25 lat; 51 isopycnic layers + 2 bulk mixed layer | 2° lon, 0.38-2° lat, 30 levels | 1.5°; 40 levels | 1° lon, 0.3 to 1° lat 75 levels, 1m at surface | 2° lon, 0.3 to 1.5° lat; 31 levels | 2° lon, 0.3 to 1.5° lat; 31 levels |

**Table A3.** Comparison of the inversion set up and input fields for the atmospheric inversions.
Atmospheric inversions see the full $CO_2$ fluxes, including the anthropogenic and pre-industrial
fluxes. Hence they need to be adjusted for the pre-industrial flux of $CO_2$ from the land to the
ocean that is part of the natural carbon cycle before they can be compared with $S_{OCEAN}$ and $S_{LAND}$
from process models. See Table 4 for references.

| | CarbonTracker Europe (CTE) | Jena CarboScope | CAMS | MIROC |
|---|---|---|---|---|
| Version number | CTE2018 | s85oc_v4.2 | v17r1 | tdi84_2018 |
| **Observations** | | | | |
| Atmospheric observations | Hourly resolution (well-mixed conditions) OBSPACK GLOBALVIEWplus v3.2 & NRTv4.2[a] | Flasks and hourly (outliers removed by 2-sigma criterion) | Daily averages of well-mixed conditions - OBSPACK GLOBALVIEWplus v3.2[a] & NRT v4.2, WDCGG, RAMCES and ICOS ATC | Flask and continuous data at remote sites from ObsPack GLOBALVIEWplus v3.2 and v4.0 |
| **Prior fluxes** | | | | |
| Biosphere and fires | SiBCASA-GFED4s[b] | No prior | ORCHIDEE (climatological), GFEDv4 & GFAS | Climatological CASA with 3-hourly downscaling |
| Ocean | Ocean inversion by Jacobson et al. (2007) | pCO$_2$-based ocean flux product oc_v1.6 (update of Rödenbeck et al., 2014) | Landschützer et al. (2015) | Takahashi et al. (2009) |
| Fossil fuels | EDGAR+IER, scaled to CDIAC | CDIAC (extended after 2013 with GCP totals) | EDGAR scaled to CDIAC | EDGARv4.3.2 (2012 map after 2013) |
| **Transport and optimization** | | | | |
| Transport model | TM5 | TM3 | LMDZ v5A | MIROC4-ACTM |
| Weather forcing | ECMWF | NCEP | ECMWF | JRA55 |
| Resolution (degrees) | Global: 3° x 2°, Europe: 1° x 1°, North America: 1° x 1° | Global: 4° x 5° | Global: 3.75° x 1.875° | Global: 2.8° × 2.8° |
| Optimization | Ensemble Kalman filter | Conjugate gradient (re-ortho-normalization)[c] | Variational | Matrix Method, 84 regions |

[a](GLOBALVIEW, 2016;Carbontracker Team, 2017)
[b](van der Velde et al., 2014)
[c]ocean prior not optimised

**Table A4** Attribution of $fCO_2$ measurements for the year 2017 included in SOCAT v6 (Bakker et al., 2016) to inform ocean $pCO_2$-based flux products.

| Platform | Regions | No. of | Principal Investigators | No. of data sets |
|---|---|---|---|---|
| Allure of the Seas | Tropical Atlantic | 127007 | Wanninkhof, R. : Pierrot, D. | 51 |
| Atlantic Cartier | North Atlantic | 33565 | Steinhoff, T. : Koertzinger, A. : Wallace, D. | 7 |
| Aurora Australis | Southern Ocean | 64481 | Tilbrook, B.: Neill, C.: Akl, J. | 3 |
| Benguela Stream | North Atlantic; Tropical Atlantic | 105517 | Schuster, U. : Watson, A.J. | 17 |
| BOBOA_90E_15N | Indian Ocean | 66 | Sutton, A. : O Brien, C. : Hermes, R. | 1 |
| Cap san Lorenzo | North Atlantic; Tropical Atlantic | 33901 | Lefevre, N.: Diverres, D. | 7 |
| Colibri | North Atlantic; Tropical Atlantic | 9334 | Lefevre, N.: Diverres, D. | 2 |
| Discovery | North Atlantic | 2540 | Kitidis, V. | 1 |
| Equinox | Tropical Atlantic | 114369 | Wanninkhof, R. : Pierrot, D. | 42 |
| Finnmaid | North Atlantic | 128793 | Rehder, G. : Glockzin, M. | 11 |
| G.O. Sars | North Atlantic | 99028 | Skjelvan, I. | 7 |
| Gordon Gunter | North Atlantic; Tropical Atlantic | 60213 | Wanninkhof, R. : Pierrot, D. | 12 |
| Henry B. Bigelow | North Atlantic | 40703 | Wanninkhof, R. : Pierrot, D. | 7 |
| Heron Island | Tropical Pacific | 2775 | Tilbrook, B.: van Ooijen, E.: Passmore, A. | 2 |
| Investigator | Southern Ocean; Tropical Pacific | 98081 | Tilbrook, B.: Neill, C.: Akl, J. | 6 |
| Kangaroo Island | Southern Ocean | 1650 | Tilbrook, B.: van Ooijen, E.: Passmore, A. | 1 |
| Laurence M. Gould | Southern Ocean | 41657 | Sweeney, C. : Takahashi, T. : Newberger, T. : Sutherland, S.C. : Munro, D.R. | 7 |
| Maria Island | Southern Ocean | 3023 | Tilbrook, B.: van Ooijen, E.: Passmore, A. | 2 |
| Marion Dufresne | Indian Ocean; Southern Ocean | 6641 | Metzl, N. : Lo Monaco, C. | 1 |
| MSC Marianna | North Atlantic; Tropical Atlantic | 2823 | Gonzalez-Davila, M. : Santana-Casiano, J.M. | 1 |
| New Century 2 | North Atlantic; North Pacific; Tropical Atlantic; Tropical Pacific | 28604 | Nakaoka, S. | 13 |
| Nuka Arctica | North Atlantic | 139842 | Becker, M. : Olsen, A.: Johannessen, T. | 29 |
| Polarstern | Arctic, North Atlantic, Southern Ocean; Tropical Atlantic | 135031 | van Heuven, S. : Hoppema, M. | 6 |
| Ronald H. Brown | Southern Ocean; Tropical Atlantic, Tropical Pacific | 45510 | Wanninkhof, R. : Pierrot, D. | 4 |
| S.A. Agulhas II | Southern Ocean | 8990 | Monteiro, P.M.S. : Gregor, L. | 1 |
| Simon Stevin | North Atlantic | 12189 | Gkritzalis, T. : Theetaert, H. | 3 |
| Soyo Maru | North Pacific | 49613 | Ono, T. | 3 |
| TAO110W_0N | Tropical Pacific | 825 | Sutton, A. | 2 |
| Trans Future 5 | North Pacific, Southern Ocean; Tropical Pacific | 22596 | Nakaoka, S. : Nojiri, Y. | 21 |
| Victor Angelescu | North Atlantic, Southern Ocean, Tropical Atlantic | 4624 | Negri, R.: Padin, X.A. | 1 |
| Wakmatha | Tropical Pacific | 20496 | Tilbrook, B.: Neill, C.: Akl, J. | 6 |

**Table A5.** Funding supporting the production of the various components of the global carbon budget in addition to the authors' supporting institutions (see also acknowledgements).

| Funder and grant number (where relevant) | author initials |
|---|---|
| Australia, Great Barrier Reef Foundation | BT, CN |
| Australia, Integrated Marine Observing System (IMOS) | BT, CN |
| Australian Government National Environment Science Program (NESP) | JGC, VH |
| EC H2020 (AtlantOS: grant no 633211) | AO, US |
| EC H2020 (CRESCENDO: grant no. 641816) | MF, PF, RS, TI |
| EC H2020 European Research Council (ERC) Synergy grant (IMBALANCE-P; grant no. ERC-2013-SyG-610028) | DSG |
| EC H2020 ERC (QUINCY; grant no. 647204). | SZ |
| EC H2020 (RINGO: grant no. 730944; FixO3: grant no. 312463). | US |
| EC H2020 project (VERIFY: grant no. 776810) | CLQ, GPP, IH, JIK, RMA, PP, PC |
| FRA, MOE | TO |
| French Institut National des Sciences de l'Univers (INSU) and Institut Paul Emile Victor (IPEV), Sorbonne Universités (UPMC, Univ Paris 06) | NM |
| German Federal Ministry for Education and Research (BMBF) | GR, MH, TS |
| German Federal Ministry of Transport and Digital Infrastructure (BMVI) | GR, MH, TS |
| German Helmholtz Association in its ATMO programme | AA |
| German Helmholtz Association Innovation and Network Fund (VH-NG-1301) | JH |
| German Research Foundation's Emmy Noether Programme (grant no. PO1751/1-1) | JP |
| Integrated Carbon Observation System (ICOS) RI | GR, MH, NL, TG, TJ, TS, IS, US |
| French Institut de Recherche pour le Développement (IRD) | NL |
| Japan Environment Research and Technology Development Fund of the Ministry of the Environment (grant no. 2-1701) | PKP |
| Japan Fisheries Research and Education Agency (FREA), Ministry of Environment (MOE) | TO |
| Japan National Institute for Environmental Studies (NIES), Ministry of Environment (MOE) | SN |
| Netherlands Organization for Scientific Research (NWO; grant no. SH-312, 16666) | IvdLL |
| Norwegian Research Council (grant no. 229771) | JS |
| Norwegian Research Council (grant no. ICOS 245927) | IS, TJ, BP |
| Norwegian Research Council (grant no. 209701) | RMA, JIK, GPP |
| The Netherlands, Research Foundation – Flanders (FWO contract no. G0H3317N) | TG |
| The Copernicus Atmosphere Monitoring Service, implemented by the European Centre for Medium-Range Weather Forecasts (ECMWF) on behalf of the European Commission | FC |
| Swiss National Science Foundation (grant no. 200020_172476) | SL |
| UK BEIS/Defra Met Office Hadley Centre Climate Programme (grant no. GA01101) | CDJ |
| UK Natural Environment Research Council (SONATA: grant no. NE/P021417/1) | CLQ, US |
| UK NERC (RAGNARoCC: grant no. NE/K002473/1) | US |
| UK Newton Fund, Met Office Climate Science for Service Partnership Brazil (CSSP Brazil) | AW |
| USA Climate Program Office of NOAA (grant no. NA13OAR4310219) | LR |
| USA Department of Agriculture, National Institute of Food and Agriculture (grants no. 2015-67003-23489 and 2015-67003-23485) | DLL |
| USA Department of Commerce, NOAA/OAR's Global Ocean Monitoring & Observing Program | AS, LB, DP |
| USA Department of Commerce, NOAA/OAR's Ocean Acidification Program | AS, DP, LB |

| | |
|---|---|
| USA Department of Energy, Oak Ridge National Laboratory (contract no. DE-AC05-00OR22725) | APW |
| USA Department of Energy, Office of Science and BER prg. (grant no. DE-SC000 0016323) | ATJ |
| USA Department of Energy (grants no. DE-FC03-97ER62402/A010 and DE-SC0012972) | DLL |
| USA NASA Interdisciplinary Research in Earth Science Program. | BP |
| **Computing resources** | |
| Norway UNINETT Sigma2, National Infrastructure for High Performance Computing and Data Storage in Norway (NN2980K/NS2980K) | JS |
| TGCC under allocations 2017-A0030102201 and 2017-A0030106328 made by GENCI | FC, NV |
| Japan National Institute for Environmental Studies computational resources | EK |
| UEA High Performance Computing Cluster, UK | RW, CLQ |
| **Support for aircraft measurements in Obspack** | |
| L. V. Gatti, M. Gloor, J.B. Miller: AMAZONICA consorcium project was funded by NERC (NE/F005806/1), FAPESP (08/58120-3), GEOCARBON project (283080) | |
| Joshua DiGangi, NASA Langley Research Center, principal investigator of the airborne instrument that collected all of the $CO_2$ observations during the Atmospheric Carbon and Transport – America campaigns. | |
| Observations from the The Atmospheric Carbon and Transport (ACT) - America Earth Venture Suborbital mission were funded by NASA's Earth Science Division (Grant NNX15AG76G to Penn State) | |
| Jeff Peischl of the University of Colorado/CIRES for the NOAA WP-3D aircraft vertical profile data | |

**Table A6.** Aircraft measurement programs archived by Cooperative Global Atmospheric Data Integration Project (CGADIP, 2017) that contribute to the evaluation of the atmospheric inversions (Figure B3).

| Measurement program name in Obspack | Specific doi | Data providers |
|---|---|---|
| Airborne Aerosol Observatory, Bondville, Illinois | | Wanninkhof, R. : Pierrot, D. |
| Alaska Coast Guard | | Sweeney, C.; McKain, K.; Karion, A.; Dlugokencky, E.J. |
| Atmospheric Carbon and Transport - America | https://doi.org/10.3334/ORNLDAAC/1556 | Davis, K.J.; Digangi, J.P.; Yang, M. |
| Atmospheric Carbon and Transport - America | | Davis, K.J.; Sweeney, C.; Dlugokencky, E.J.; Yang, M. |
| Alta Floresta | | Gatti, L.V.; Gloor, E.; Miller, J.B.; |
| Aircraft Observation of Atmospheric trace gases by JMA | | ghg_obs@met.kishou.go.jp |
| Aerosol, Radiation, and Cloud Processes affecting Arctic Climate 2008 (air campaign) | | Ryerson, T.B.; Peischl, J.; Aikin, K.C. |
| LARC - NASA Langley Research Center Aircraft Campaign | https://doi.org/10.3334/ORNLDAAC/1556 | Chen, G.; Digangi, J.P. |
| Beaver Crossing, Nebraska | | Sweeney, C.; Dlugokencky, E.J. |
| California Nexus 2010 (air campaign) | | Ryerson, T.B.; Peischl, J.; Aikin, K.C. |
| Briggsdale, Colorado | | Sweeney, C.; Dlugokencky, E.J. |
| Cape May, New Jersey | | Sweeney, C.; Dlugokencky, E.J. |
| CONTRAIL (Comprehensive Observation Network for TRace gases by AIrLiner) | http://dx.doi.org/10.17595/20180208.001 | Machida, T.; Matsueda, H.; Sawa, Y. Niwa, Y. |
| Carbon in Arctic Reservoirs Vulnerability Experiment (CARVE) | | Sweeney, C.; Karion, A.; Miller, J.B.; Miller, C.E.; Dlugokencky, E.J. |
| LARC - NASA Langley Research Center Aircraft Campaign | https://doi.org/10.3334/ORNLDAAC/1556 | Chen, G.; Digangi, J.P.; Beyersdorf, A. |
| LARC - NASA Langley Research Center Aircraft Campaign | https://doi.org/10.3334/ORNLDAAC/1556 | Chen, G.; Digangi, J.P.; Yang, M. |
| Dahlen, North Dakota | | Sweeney, C.; Dlugokencky, E.J. |
| Estevan Point,  British Columbia | | Sweeney, C.; Dlugokencky, E.J. |
| East Trout Lake, Saskatchewan | | Sweeney, C.; Dlugokencky, E.J. |
| Molokai Island, Hawaii | | Sweeney, C.; Dlugokencky, E.J. |
| Homer, Illinois | | Sweeney, C.; Dlugokencky, E.J. |
| HIPPO (HIAPER Pole-to-Pole Observations) | https://doi.org/10.3334/CDIAC/HIPPO_010 | Wofsy, S.C.; Stephens, B.B.; Elkins, J.W.; Hintsa, E.J.; Moore, F. |
| INFLUX (Indianapolis Flux Experiment) | | Sweeney, C.; Dlugokencky, E.J.; Shepson, P.B.; Turnbull, J. |
| Park Falls, Wisconsin | | Sweeney, C.; Dlugokencky, E.J. |
| Mid Continent Intensive | | Sweeney, C.; Dlugokencky, E.J. |
| Marcellus Pennsylvania | | Sweeney, C.; Dlugokencky, E.J. |
| Worcester, Massachusetts | | Sweeney, C.; Dlugokencky, E.J. |
| ORCAS (O2/N2 Ratio and CO2 Airborne Southern Ocean Study) | https://doi.org/10.5065/D6SB445X | Stephens, B.B.; Sweeney, C.; McKain, K.; Kort, E.A. |
| Poker Flat, Alaska | | Sweeney, C.; Dlugokencky, E.J. |
| Rio Branco | | Gatti, L.V.; Gloor, E.; Miller, J.B. |
| Rarotonga | | Sweeney, C.; Dlugokencky, E.J. |
| Montzka | | Sweeney, C.; Dlugokencky, E.J. |

| | | |
|---|---|---|
| Santarem | | Sweeney, C.; Dlugokencky, E.J. |
| Charleston, South Carolina | | Sweeney, C.; Dlugokencky, E.J. |
| LARC - NASA Langley Research Center Aircraft Campaign | https://doi.org/10.3334/ORNLDAAC/1556 | Chen, G.; Digangi, J.P.; Beyersdorf, A. |
| Southeast Nexus 2013 (air campaign) | | Ryerson, T.B.; Peischl, J.; Aikin, K.C. |
| Southern Great Plains, Oklahoma | | Sweeney, C.; Dlugokencky, E.J.; Biraud, S. |
| Shale Oil and Natural Gas Nexus 2015 (air campaign) | | Ryerson, T.B.; Peischl, J.; Aikin, K.C. |
| Harvard University Aircraft Campaign | | Wofsy, S.C. |
| Tabatinga | | Gatti, L.V.; Gloor, E.; Miller, J.B. |
| Sinton, Texas | | Sweeney, C.; Dlugokencky, E.J. |
| Trinidad Head, California | | Sweeney, C.; Dlugokencky, E.J. |
| Atmospheric Tomography Mission (ATom) | | McKain, K.; Sweeney, C. |
| Ulaanbaatar | | Sweeney, C.; Dlugokencky, E.J. |
| West Branch, Iowa | | Sweeney, C.; Dlugokencky, E.J. |

**Figure Captions**

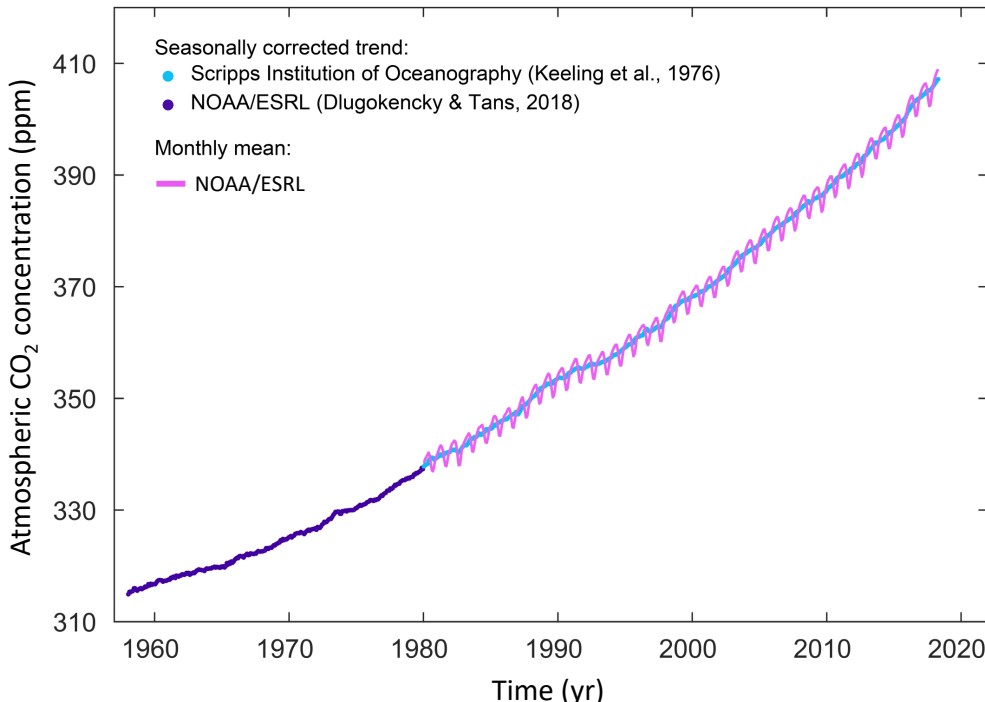

**Figure 1.** Surface average atmospheric $CO_2$ concentration (ppm). The 1980-2018 monthly data are
from NOAA/ESRL (Dlugokencky and Tans, 2018) and are based on an average of direct
atmospheric $CO_2$ measurements from multiple stations in the marine boundary layer (Masarie and
Tans, 1995). The 1958-1979 monthly data are from the Scripps Institution of Oceanography, based
on an average of direct atmospheric $CO_2$ measurements from the Mauna Loa and South Pole
stations (Keeling et al., 1976). To take into account the difference of mean $CO_2$ and seasonality
between the NOAA/ESRL and the Scripps station networks used here, the Scripps surface average
(from two stations) was deseasonalised and harmonised to match the NOAA/ESRL surface average
(from multiple stations) by adding the mean difference of 0.542 ppm, calculated here from
overlapping data during 1980-2012.

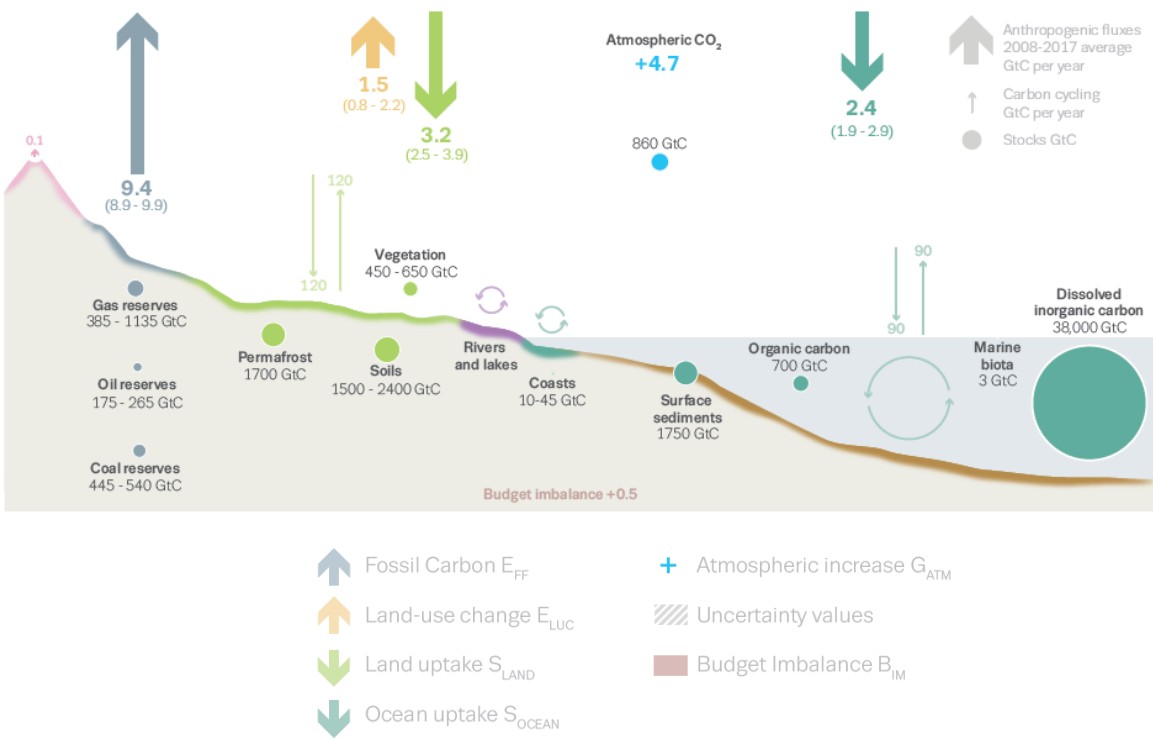

**Figure 2.** Schematic representation of the overall perturbation of the global carbon cycle caused by anthropogenic activities, averaged globally for the decade 2008-2017. See legends for the corresponding arrows and units. The uncertainty in the atmospheric $CO_2$ growth rate is very small (±0.02 Gt C yr$^{-1}$) and is neglected for the figure. The anthropogenic perturbation occurs on top of an active carbon cycle, with fluxes and stocks represented in the background and taken from Ciais et al. (2013) for all numbers, with the ocean fluxes updated to 90 GtC yr$^{-1}$ to account for the increase in atmospheric $CO_2$ since publication, and except for the carbon stocks in coasts which is from a literature review of coastal marine sediments (Price and Warren, 2016).

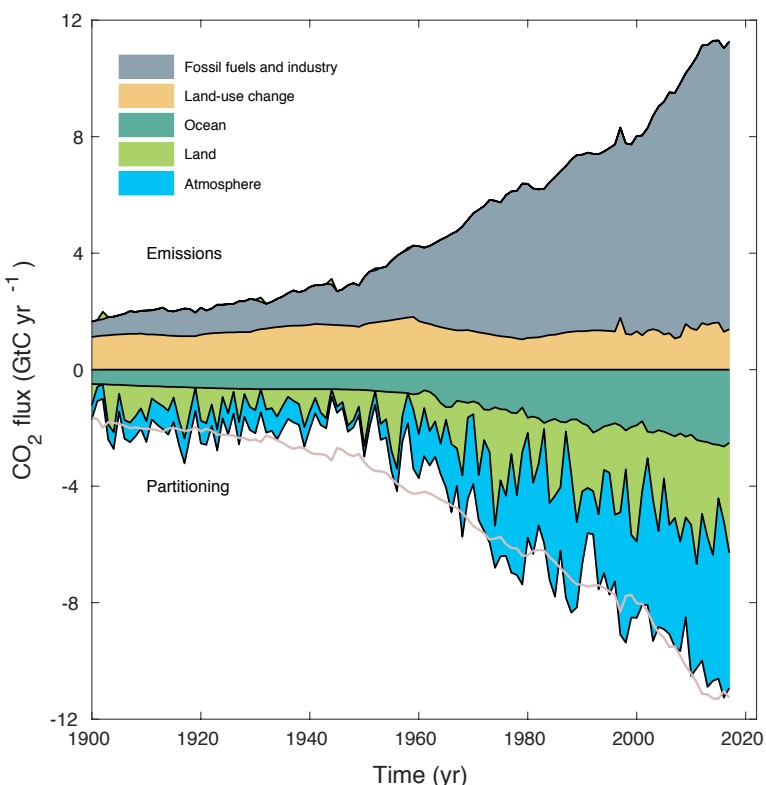

**Figure 3.** Combined components of the global carbon budget illustrated in Fig. 2 as a function of

time, for fossil $CO_2$ emissions ($E_{FF}$; grey) and emissions from land-use change ($E_{LUC}$; brown), as well

as their partitioning among the atmosphere ($G_{ATM}$; purple), ocean ($S_{OCEAN}$; blue), and land ($S_{LAND}$;

green). The partitioning is based on nearly independent estimates from observations (for $G_{ATM}$)

and from process model ensembles constrained by data (for $S_{OCEAN}$ and $S_{LAND}$), and does not

exactly add up to the sum of the emissions, resulting in a budget imbalance which is represented

by the difference between the bottom pink line (reflecting total emissions) and the sum of the

ocean, land and atmosphere. All time series are in GtC yr$^{-1}$. $G_{ATM}$ and $S_{OCEAN}$ prior to 1959 are

based on different methods. $E_{FF}$ are primarily from Boden et al. (2017), with uncertainty of about

±5% (±1σ); $E_{LUC}$ are from two bookkeeping models (Table 2) with uncertainties of about ±50%;

$G_{ATM}$ prior to 1959 is from Joos and Spahni (2008) with uncertainties equivalent to about ±0.1-0.15

GtC yr$^{-1}$, and from Dlugokencky and Tans (2018) from 1959 with uncertainties of about ±0.2 GtC

yr$^{-1}$; $S_{OCEAN}$ prior to 1959 is averaged from Khatiwala et al. (2013) and DeVries (2014) with

uncertainty of about ±30%, and from a multi-model mean (Table 4) from 1959 with uncertainties

of about ±0.5 GtC yr$^{-1}$; $S_{LAND}$ is a multi-model mean (Table 4) with uncertainties of about ±0.9 GtC

yr$^{-1}$. See the text for more details of each component and their uncertainties.

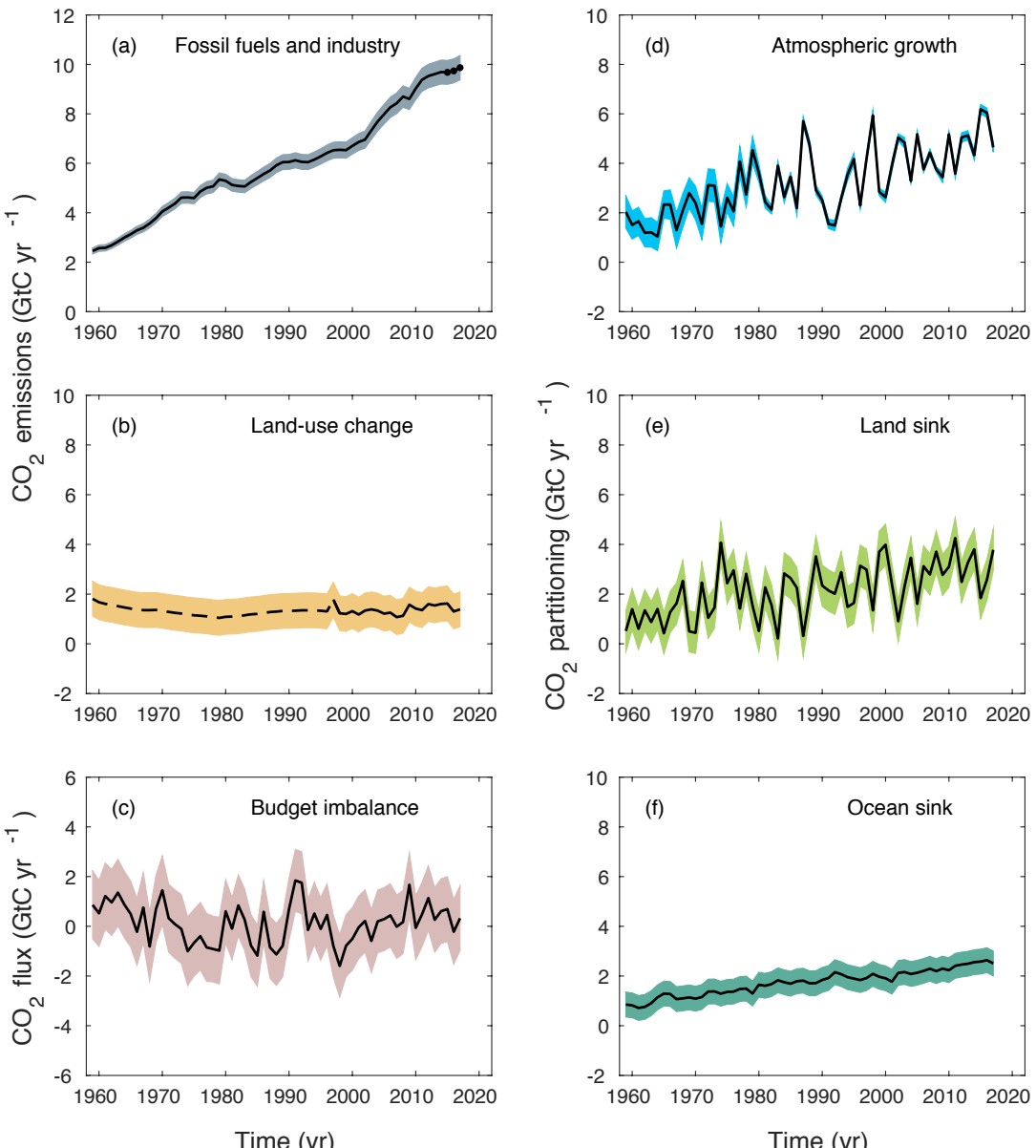

**Figure 4.** Components of the global carbon budget and their uncertainties as a function of time, presented individually for **(a)** fossil $CO_2$ emissions ($E_{FF}$), **(b)** emissions from land-use change ($E_{LUC}$), **(c)** the budget imbalance that is not accounted for by the other terms, **(d)** growth rate in atmospheric $CO_2$ concentration ($G_{ATM}$), and **(e)** the land $CO_2$ sink ($S_{LAND}$, positive indicates a flux from the atmosphere to the land), **(f)** the ocean $CO_2$ sink ($S_{OCEAN}$, positive indicates a flux from the atmosphere to the ocean). All time series are in GtC yr$^{-1}$ with the uncertainty bounds representing $\pm 1\sigma$ in shaded colour. Data sources are as in Fig. 3. The black dots in **(a)** show values for 2015-2017 that originate from a different data set to the remainder of the data (see text). The dashed line in (b) identifies the pre-satellite period before the inclusion of peatland burning.

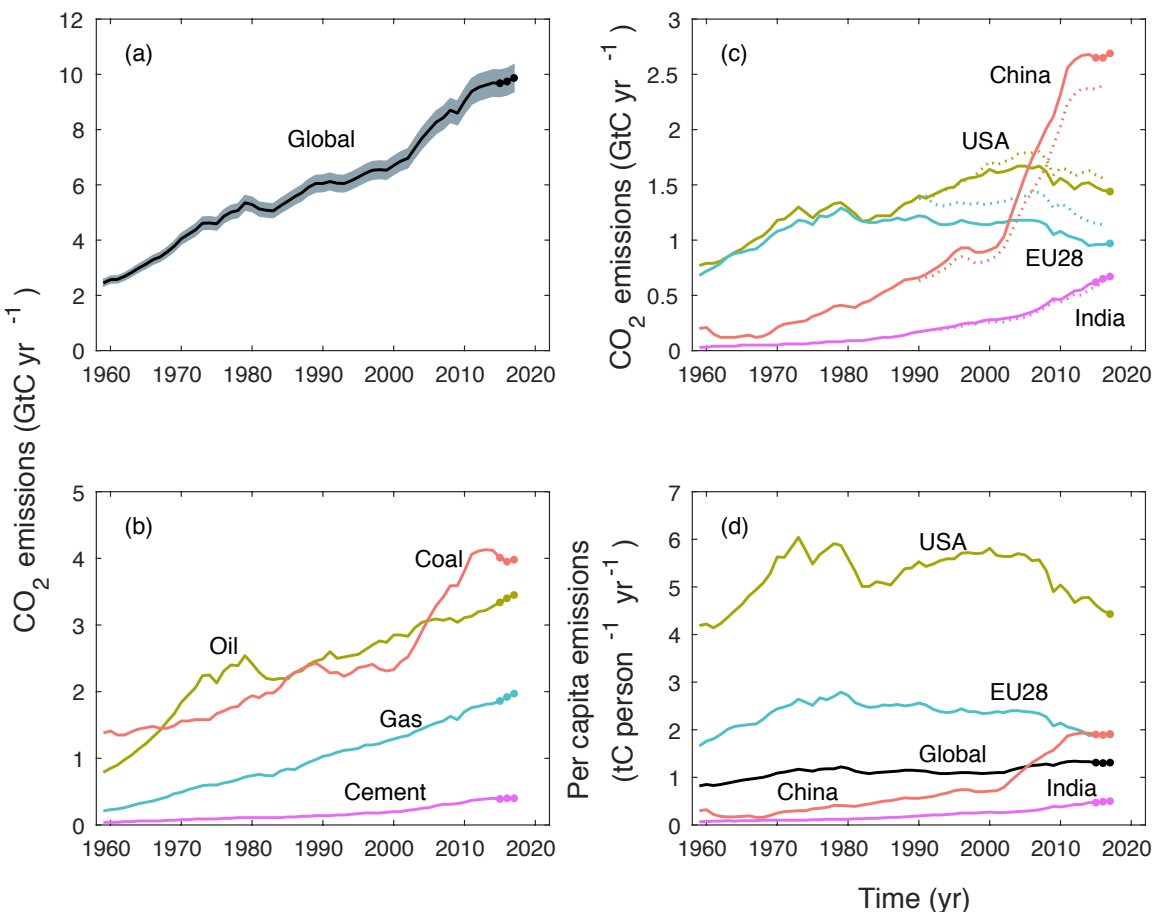

**Figure 5.** Fossil $CO_2$ emissions for **(a)** the globe, including an uncertainty of ± 5% (grey shading),

and the emissions extrapolated using BP energy statistics (black dots), **(b)** global emissions by fuel

type, including coal (salmon), oil (olive), gas (turquoise), and cement (purple), and excluding gas

flaring which is small (0.6% in 2013), **(c)** territorial (solid lines) and consumption (dashed lines)

emissions for the top three country emitters (USA - olive; China - salmon; India - purple) and for

the European Union (EU; turquoise for the 28 member states of the EU as of 2012), and **(d)** per-

capita emissions for the top three country emitters and the EU (all colours as in panel **(c)**) and the

world (black). In **(b-c)**, the dots show the data that were extrapolated from BP energy statistics for

2014-2016. All time series are in GtC yr$^{-1}$ except the per-capita emissions **(d)**, which are in tonnes

of carbon per person per year (tC person$^{-1}$ yr$^{-1}$). Territorial emissions are primarily from Boden et

al. (2017) except national data for the USA and EU28 (the 28 member states of the EU) for 1990-

2016, which are reported by the countries to the UNFCCC as detailed in the text; consumption-

based emissions are updated from Peters et al. (2011a). See Sect. 2.1.1 for details of the

calculations and data sources.

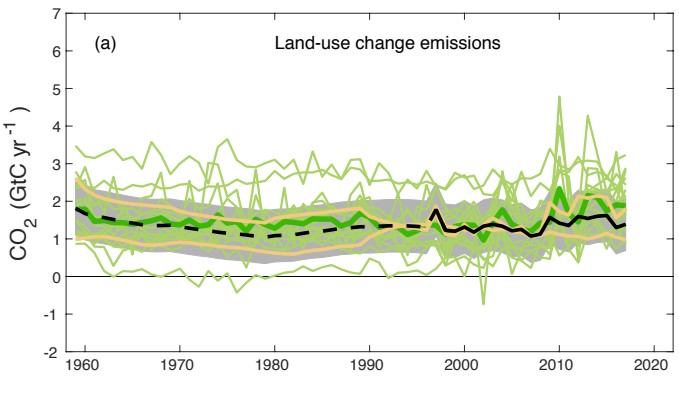

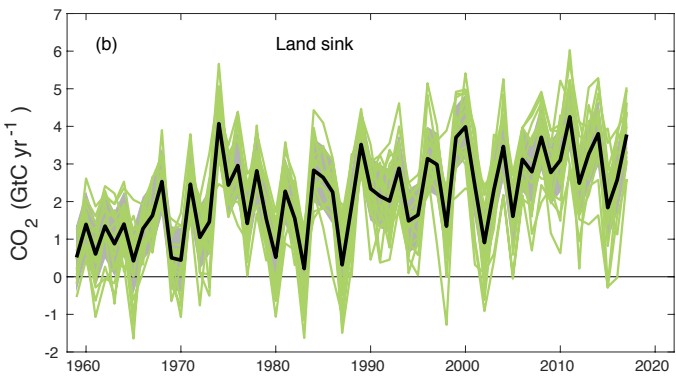

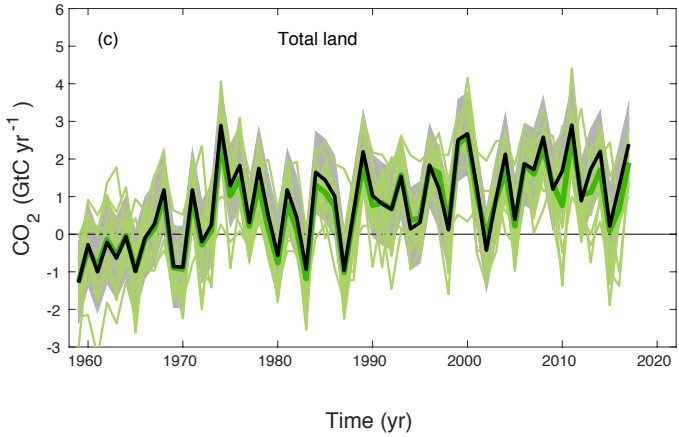

**Figure 6.** $CO_2$ exchanges between the atmosphere and the terrestrial biosphere as used in the

global carbon budget (black with ±1σ uncertainty in grey shading), for **(a)** $CO_2$ emissions from

land-use change ($E_{LUC}$), showing also individually the two bookkeeping models (two brown lines)

and the DGVM model results (green) and their multi-model mean (dark green). The dashed line

identifies the pre-satellite period before the inclusion of peatland burning; **(b)** Land $CO_2$ sink

($S_{LAND}$) with individual DGVMs (green); **(c)** Total land $CO_2$ fluxes (**b minus a**) with individual DGVMs

(green) and their multi-model mean (dark green).

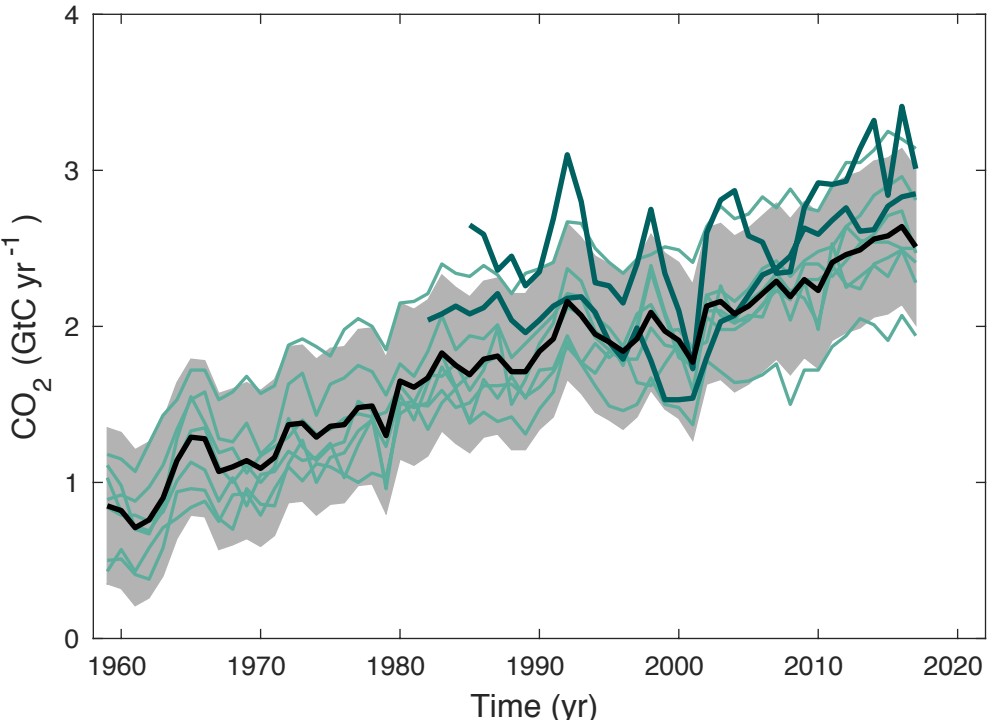

**Figure 7.** Comparison of the anthropogenic atmosphere-ocean $CO_2$ flux showing the budget values
of $S_{OCEAN}$ (black; with ±1σ uncertainty in grey shading), individual ocean models (blue), and the two
ocean $pCO_2$-based flux products (dark blue; see Table 4). Both $pCO_2$-based flux products were
adjusted for the preindustrial ocean source of $CO_2$ from river input to the ocean, which is not
present in the ocean models, by adding a sink of 0.78 GtC $yr^{-1}$ (Resplandy et al., 2018), to make
them comparable to $S_{OCEAN}$. This adjustment does not take into account the anthropogenic
contribution to river fluxes (see Sect. 2.7.3).

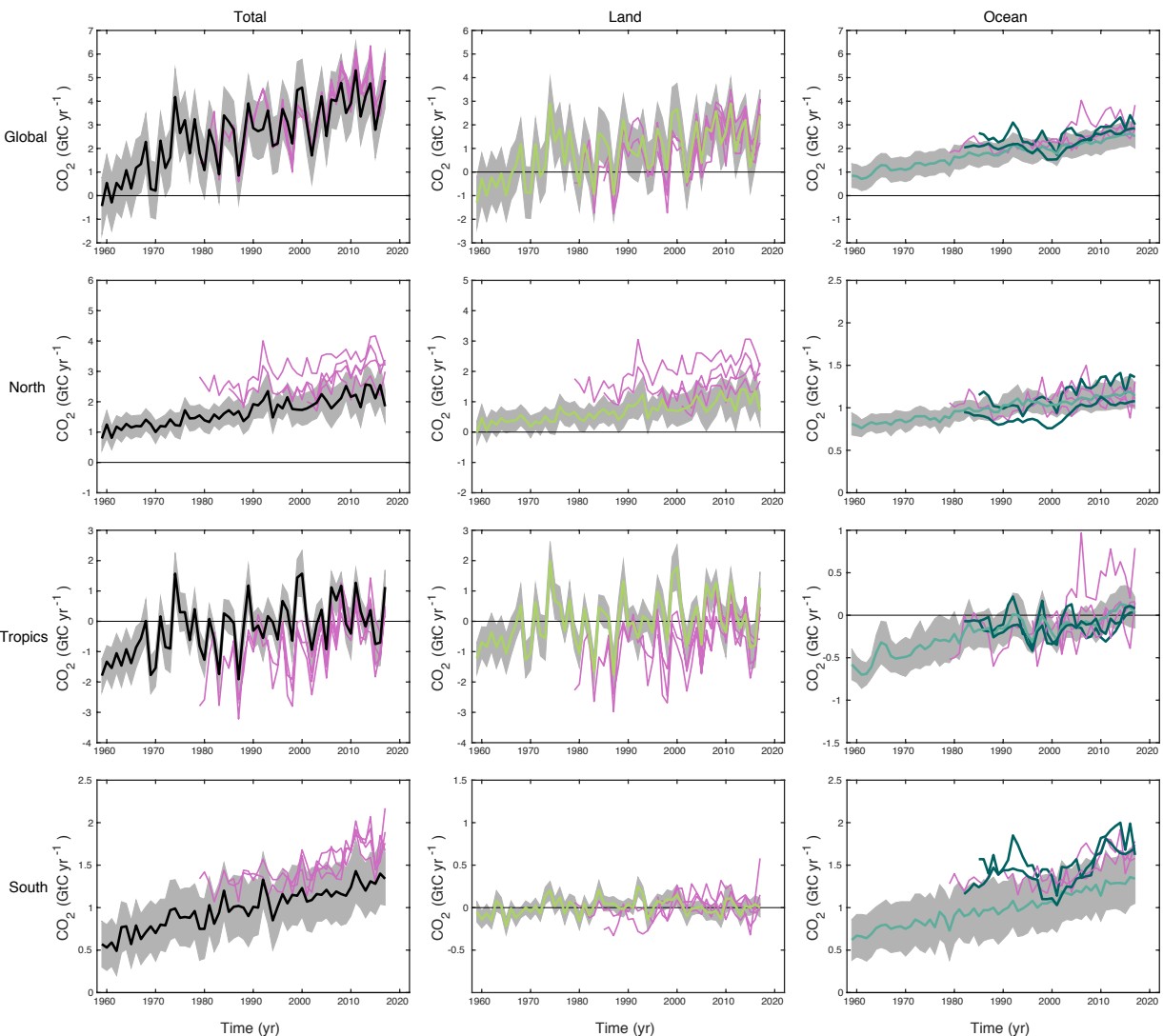

**Figure 8.** $CO_2$ fluxes between the atmosphere and the surface ($S_{OCEAN} + S_{LAND} - E_{LUC}$) by latitude

bands for the (top) globe (2nd row) north (north of 30°N), (3rd row) tropics (30°S-30°N), and

(bottom) south (south of 30°S), and (left) total, (middle) land only ($S_{LAND} - E_{LUC}$) and (right) ocean

only. Positive values indicate a flux from the atmosphere to the land and/or ocean.

Estimates from the combination of the process models for the land and oceans are shown (black

for the total, green for the land, blue for the ocean) with ±1σ of the model ensemble (in grey).

Results from the atmospheric inversions are also shown (pink lines), and from the pCO_2-based flux

products (dark blue lines).

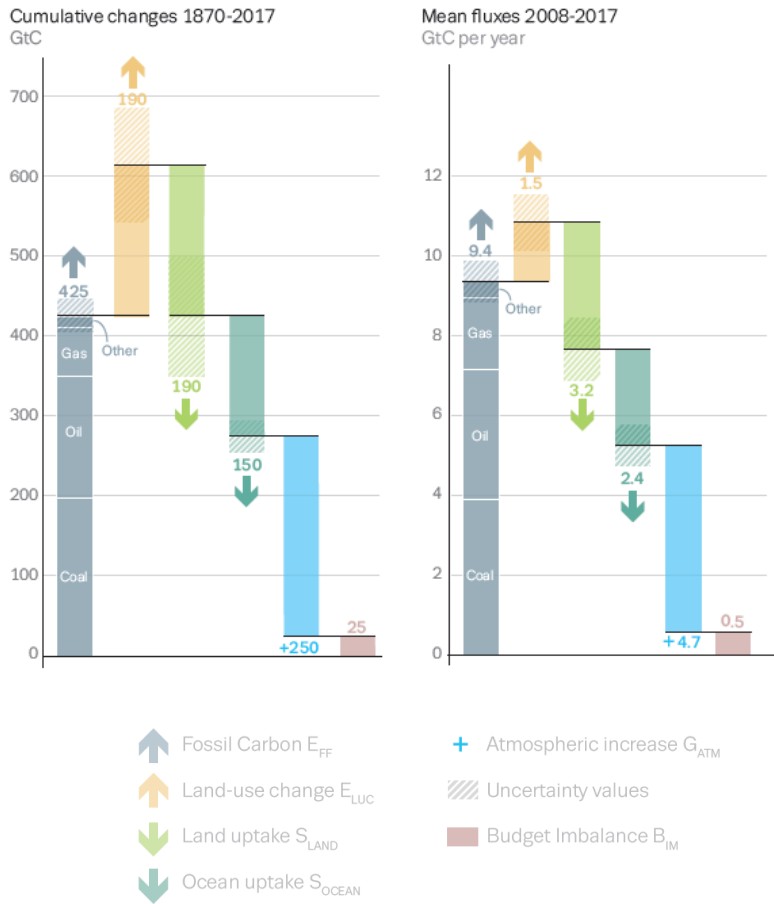

**Figure 9.** Cumulative changes during 1870-2017 and mean fluxes during 2008-2017 for the anthropogenic perturbation as defined in the legend.

**Appendix B.** Supplementary figures.

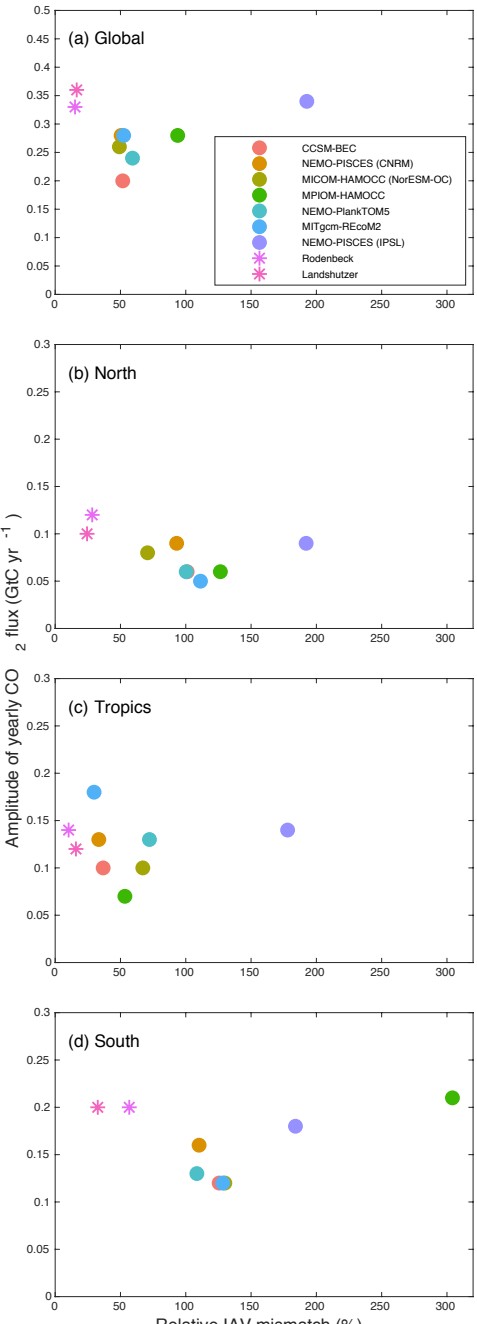

**Figure B1.** Evaluation of the GOBMs and flux products using the interannual mismatch metric for
the period 1985 to 2017, as proposed by Rödenbeck et al. (2015) and the SOCAT v6 database,
versus the amplitude of the annual variability (taken as the annual standard deviation). Results are
presented for the globe, north (>30°N), tropics (30°S-30°N), and south (<30°S) for the GOBMs
(circles) and for the $pCO_2$-based flux products (star symbols). The two $pCO_2$-based flux products
use the SOCAT database and therefore are not fully independent from the data (See section

9    2.4.1).

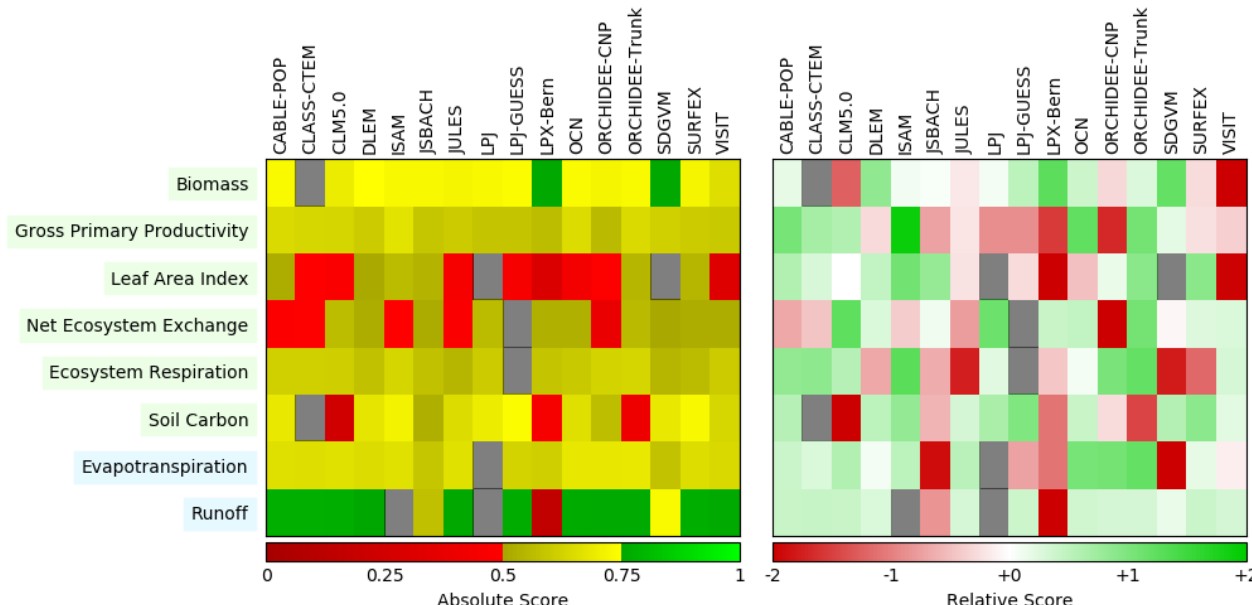

**Figure B2.** Evaluation of the DGVM using the International Land Model Benchmarking system
(ILAMB; Collier et al., 2018). (left) absolute skill scores, (right) skill scores relative to other models.
The benchmarking is done with observations for vegetation biomass (Saatchi et al., 2011; and
GlobalCarbon unpublished data;Avitabile et al., 2016), GPP (Jung et al., 2010;Lasslop et al., 2010),
leaf area index (De Kauwe et al., 2011;Myneni et al., 1997), net ecosystem exchange (Jung et al.,
2010;Lasslop et al., 2010), ecosystem respiration (Jung et al., 2010;Lasslop et al., 2010), soil
carbon (Hugelius et al., 2013;Todd-Brown et al., 2013), evapotranspiration (De Kauwe et al.,
2011), and runoff (Dai and Trenberth, 2002). For each model-observation comparison a series of
error metrics are calculated, scores are then calculated as an exponential function of each error
metric, finally for each variable the multiple scores from different metrics and observational data
sets are combined to give the overall variable scores shown in the left panel. The set of error
metrics vary with data set and can include metrics based on the period mean, bias, root mean
squared error, spatial distribution, interannual variability and seasonal cycle. The relative skill
score shown in the right panel is a Z-score, which indicates in units of standard deviation the
model scores relative to the multi-model mean score for a given variable. Grey boxes represent
missing model data.

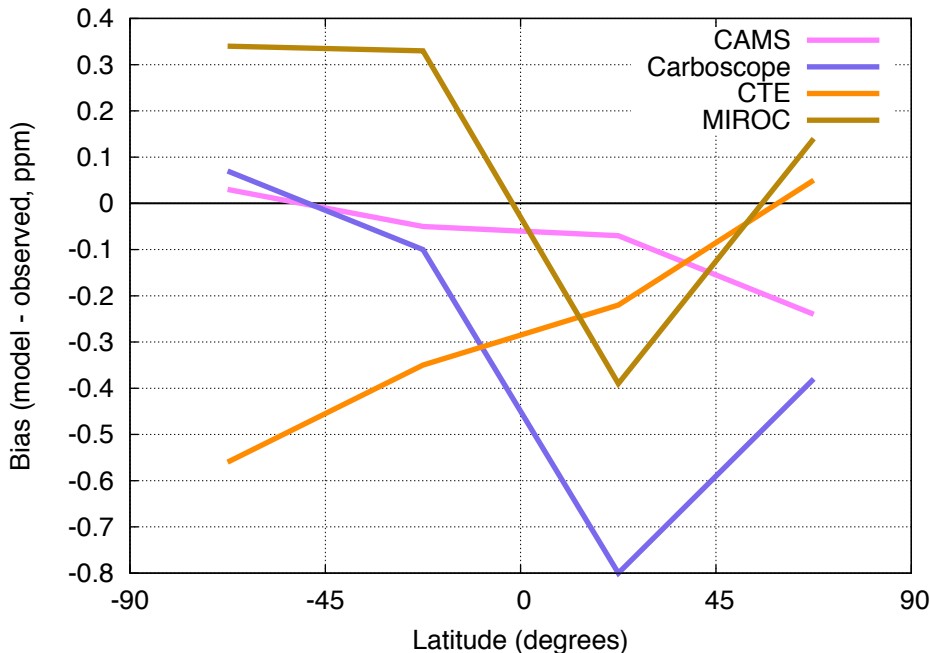

**Figure B3.** Evaluation of the atmospheric inversion products. The mean of the absolute model
minus observed is shown for four latitude bands. The four models are compared to independent
$CO_2$ measurements made onboard aircraft over many places of the world between 1 and 7 km
above sea level.  All data between 2008 and 2016 archived in Cooperative Global Atmospheric
Data Integration Project (CGADIP, 2017) have been used to compute the biases of the differences
in four 45-degree latitude bins. Land of ocean data are used without distinction. The number of
data for each latitude band is: 16,000 (90°S-45°S), 53,000 (45°S-0), 64,000 (0-45°N), 122,000
(45°N-90°N), rounded off to nearest thousand.

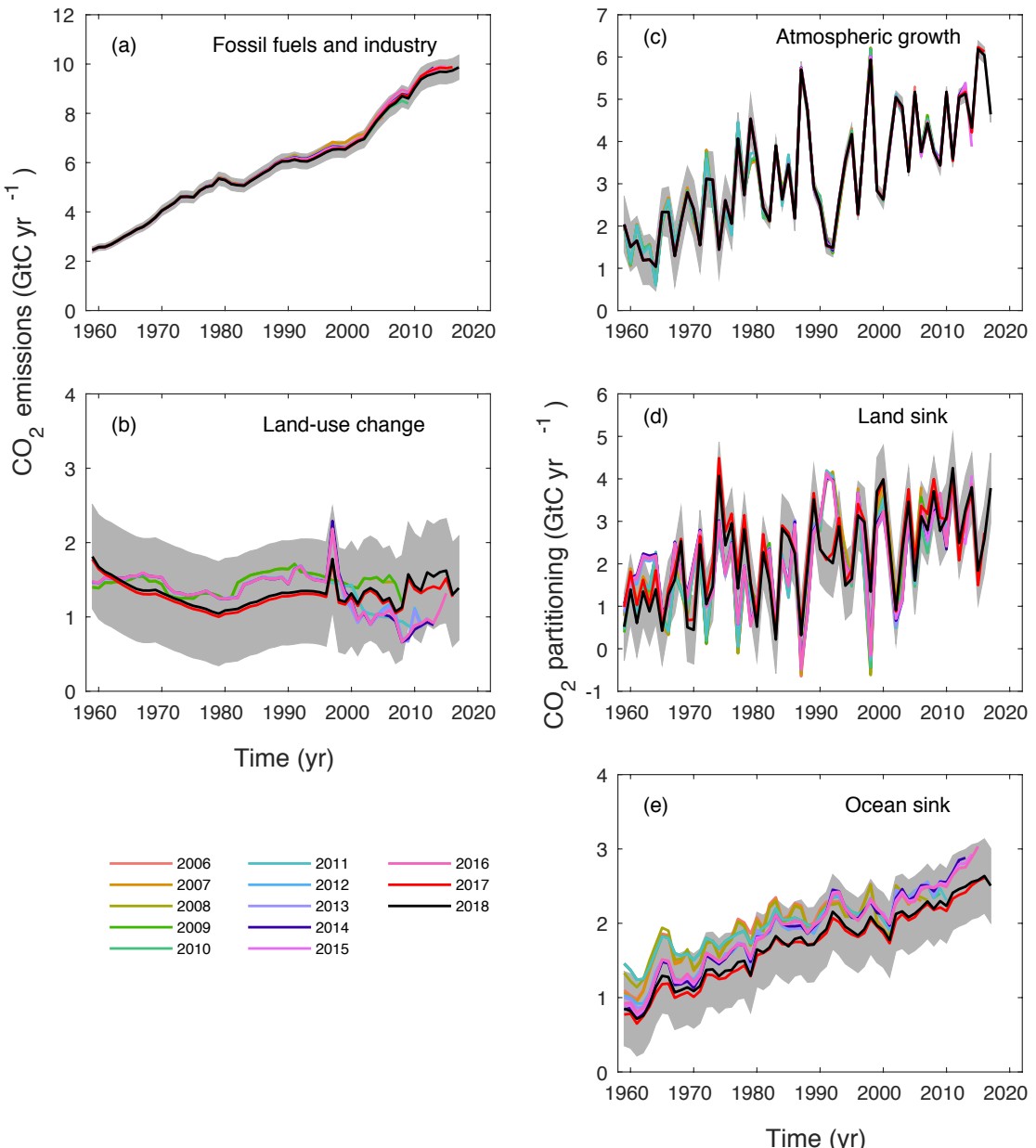

**Figure B4.** Comparison of global carbon budget components released annually by GCP since 2006.
$CO_2$ emissions from **(a)** fossil $CO_2$ emissions ($E_{FF}$), and **(b)** land-use change ($E_{LUC}$), as well as their
partitioning among **(c)** the atmosphere ($G_{ATM}$), **(d)** the land ($S_{LAND}$), and **(e)** the ocean ($S_{OCEAN}$). See
legend for the corresponding years, and Table 3 for references. The budget year corresponds to
the year when the budget was first released. All values are in GtC yr$^{-1}$. Grey shading shows the
uncertainty bounds representing ±1σ of the current global carbon budget.

