# Peer review of "Global Carbon Budget 2018"

_Earth System Science Data, 2018_

## Editor Comment (EC1) · D. J. Carlson (Editor) · 4 Oct 2018

ESSD, in company with all Copernicus journals, promotes and honours open discussion processes. However, for this particular manuscript and dataset - submitted in careful compliance with ESSD's 'Living Data' practice - at this particular time we adopt a closed non-public process during review. We take this unusual step to prevent too-early release, misuse or mis-attribution. We cooperate with partner organisations and journals to achieve simultaneous high-impact announcement. At the time of release, anticipated for early December, all materials - original manuscript, reviews, author responses, revised manuscript, etc. - will become fully open and accessible. I note that the prior version - Global Carbon Budget 2017 (https://doi.org/10.5194/essd-10-405-2018) - published in ESSD in March 2018 underwent rigorous comprehensive review and associated public discussion, with all reviews, comments and revisions openly available then and now to interested readers.

---

## Referee Report (RR1)

Review ESSD-2018-120, carbon budget

Very impressive compilation and synthesis effort!  A very good presentation and a good match to ESSD; one hopes for broad use and strong impact of these annual carbon budgets.

Not surprisingly given the amount of material processed and the number and range of contributors, the manuscript needs a few improvements.  I hope that the authors regard my close reading and consequently rather-long list of comments, questions and proposed corrections as helping to make a good product better.

Check capitalisation!  You use Tropics and tropics (often within the same paragraph!).  You use "Marine Boundary Layer" and "marine boundary layer" on the same page (Page 19).  To make your synthesis more effective you also need to impose consistent spelling and punctation.  Make these changes now to save time at the proofreading stage?  Data set vs datasets - please choose and make consistent.

Reader encounters reference to Supplement several times, but reviewer finds no supplement to evaluate!  Three supplementary figures actually appear in an Appendix, albeit referenced and labelled as S1, S2 and S3.  Meanwhile, Tables A1 to A6 carry an 'Appendix' label but do not appear with Figures S1, S2 and S3 in an Appendix.  I might understand why these particular tables carry an Appendix label and why these three figures should reside in an Appendix: to include technical details not of interest to general readers but relevant to carbon cycle colleagues?  If true (or for whatever other valid reason) we need some revision, reorganisation and re-labelling here!  If you want a proper Appendix, label and put all tables and figure in it as appropriate.

Please adopt and implement an informative flux labelling convention.  For many in our earth systems community, fluxes upward carry positive sign, downward a negative sign.  Not everyone follows this convention (no surprise) and I recognise that you specify fluxes in Gt per year rather than watts per metre squared (unfortunately those who measure $W/m^2 s^{-1}$ generally do not follow the positive up negative down guidelines) but you definitely have fluxes (emissions) going gravitationally upward and fluxes (sinks, land and ocean) going gravitationally downward. Therefore when you specify, e.g. on Page 35 (and in many many other places in the manuscript): "total atmosphere-surface $CO2$ flux from process models for 2008-2017 is 3.7 $\pm$ 1.2 GtC yr$^{-1}$" you either need to carefully specify in every case which direction you mean - in this case you would write 'atmosphere-to-surface' rather than "atmosphere-surface" - or a reader needs to recognise from the sign (positive or negative) which direction the fluxes flow.  Figure 8 highlights this need but you face the problem throughout the manuscript that readers do not know from your text or numbers in which direction the fluxes you cite actually operate.  In the new Figure 2 you show, for example, ocean-to-atmosphere natural cycle flux of 90 up and atmosphere-to-ocean return flux of 90 down (and land with atmosphere likewise, 120 up matched by 120 down) with human impact superimposed but in neutral units.  You should thus describe an 'ocean bidirectional exchange with atmosphere of 90 GtC y$^{-1}$' or you need to show 90 up as positive and 90 down as negative. The challenge becomes acute in Figure  8.  Globally, you show combined land and ocean downward fluxes (sinks!) in positive numbers (e.g. currently roughly +5 GtC y$^{-1}$) but you also show that in the 1960s and in occasional years since net land flux has gone negative (upward, land as a source!), as low as (or as much as) -1.5 to perhaps -2.0 GtC y$^{-1}$.  Globally, ocean fluxes stay downward/positive values/sink, but when you sort by latitude, the tropical ocean (which, disconcertingly, has its own ordinate range different to any of the other 11 panels in this figure) shows a clear ocean source but now designated by negative fluxes!  For purposes of this figure you have in fact adopted a convention - positive downward and negative upward - but you have not declared it and you do not follow it consistently in the text.  I agree that you can extract a lot of useful information from the flux products but you owe readers much more clarity on which direction you mean and when.  You can implement this clarification in text probably more easily than in numbers and equations, but we need a good response on this issue.

You have already done something similar for the budget imbalance, specifying it as positive some years and negative in others, because as the mathematical residual of equation 1 it will necessarily have small positive values some years but small negative values in other years.  Note

also that when you specify rates of change in % (e.g. change in growth rates of CO2 emissions as on Page 34), you carefully and correctly specify either increase or decrease, using language to indicate the direction of change, or - as in line 24 on Page 37 - you carefully use negative percentages.  We need similar careful language for flux descriptions!

After multiple readings, I still find myself confused by the Discussion. Here you have a chance to 'comment' on strengths and weaknesses and to recommend (or at least list) possible improvements.  As I read it, the discussion omits or takes a too-modest approach to positive features and strengths.  This reader finds the discussion almost apologetic in places, when in fact the authors could expand and expound on this statement (from Page 33 line 8): "the near-zero mean and trend in the budget imbalance is an indirect evidence of a coherent community understanding of the emissions and their partitioning on those time scales".  What would make the evidence more direct?  Do the authors feel that longer (paleo?) time scales (e.g. ice core data) have something useful to lend to this accounting?  The discussion also fails to follow-up on specific issues and uncertainties from the text, e.g. the so-called 'loss' of land sink capability in Section 2.7.4, the various factors that go into coming-year projections discussed in Section 3.4, the important outcomes and clear cautions about cumulative emissions accounting in Section 3.5, etc.  You likewise have an opportunity (responsibility?) here to raise highly-relevant but - to this point - not-yet-discussed issues.  E.g., how would we know if either the land or ocean sink changed significantly?  What measurements continued over what time period, and what model confirmation, would we need to quantify and certify such a change?  Such a discussion would add substantial relevance to your results and contentions on variability.  Similarly, how quickly and with what precision could we - using these data and models and this budget approach - detect and confirm an intentional drop in emissions?  How much induced carbon capture sink could we detect, and how soon?  Of course an external record for these topics often exists in scientific literature but readers across a wide spectrum - the broad stakeholders listed in the Conclusion - and particularly a research community interested in but not directly contributing to this budget would very much benefit from explanatory and enlightened discussion on these issues here!  This annual carbon budget effort also constitutes a unique, very necessary, very positive yet challenging mixing of model products with observational products; many readers will not encounter such a breadth of model intercomparisons with observations elsewhere.  This particular version of the budget has attempted a positive step forward by introducing model metrics - twice (unfortunately) in the Discussion text and (not very helpfully) in Figures S1 and (better) in S2 - but inboth cases with a very tentative presentation.  If metrics seem important, and better exposed and evaluated from the point of view of a carbon budget than from some future technical CMIP6 overview (or, god forbid, an IPCC report), shouldn't the authors spend more time in the discussion on what the metrics indicate (not much, evidently) or on what better metrics they might apply or imagine.  In various specific sections below I have made so bold as to suggest alternate language, occasionally even for a full sentence or two.  I looked closely at the Discussion with similar modifications in mind but feel that the actual text of the discussion belongs entirely and exclusively to the authors.  I understand political pressures from various constituencies and also the fact that each annual budget runs right to the edge of release schedules, but I strongly urge the authors to reconsider this Discussion in light of the overall positive effort and some of the clear messages that emerge from that effort.

Good update and changes to Figure 2.

Apparently the word processing software allows line breaks at hyphens, including at the superscript hyphen $yr^{-1}$.  This results in unfortunate line breaks in many locations in the text leaving an orphan superscript 1 at the start of the following line.  Authors and proofreaders will need to watch carefully for these errors in the final product.

I don't know what ESSD or Copernicus recommend or require, but most manuscript reference lists today seem to use doi in place of pages and dates?  Perhaps a change for a future version?

Page and line specific comments:

Page 8 line 2: "estimates of EFF globally and nationally CO2 emissions"  Something awkward here?  Because you have defined global and national CO2 emissions as EFF in the prior sentence, here you only need to say 'global and national EFF'?

Page 8 line 8: "Official UNFCCC national inventory reports for 1990-2016 for the 42 Annex I countries in the UNFCCC (UNFCCC, 2018), as we assess these to be the most accurate estimates because they are compiled by experts within countries that have access to detailed energy data, and they are periodically reviewed." Awkward language here, allows some confusion. I think you mean (roughly) national inventory reports from UNFCC. We assess these national reports to provide the most accurate estimates because they are compiled by experts within countries who have access to detailed national energy data and because they receive periodic review by UNFCCC.

Page 8 line 16: Again awkward language allows subject-predicate confusion. I think you mean 'provide more details for each dataset and describe additional modifications required to make the dataset consistent and usable.'

Page 12, lines 8 to 13: Check punctuation here, I find commas or semicolons missing. Also "and (3)" followed by "and (4)". Authors will know best how they want to separate and punctuate this list.

Page 13 lines 5 and 7: You designate the two (oil and natural gas) PPAC references here in the text as 2018a and 2018b but in the reference list you show them both as PPAC 2018. Change something somewhere?

Page 13 line 18: "which is much less strongly seasonal because of strong weather variations" I had to think about this for a moment. I think you mean that shorter-term cold or warm weather periods dominate coal usage patterns due to their impact on residential and commercial heating or general energy use, as opposed to a smoother winter/high summer/low pattern of use? But, if correct, the same pattern or factors would apply to US and China as well? Already included in CCIA projections for China or EIA projections for US, but not in projections for Europe? If I understand the statement correctly, I find it strange that you only invoke it for Europe. If short-term weather dominates, and varies widely, for Europe then your extrapolation from the prior year for missing winter months of the current year seems unreliable?

Page 14 line 27: "Additionally, they represents permanent degradation of forests by lower vegetation and soil carbon stocks for secondary as compared to the primary forests and forest management such as wood harvest." Awkward. They (meaning bookkeeping models) 'represent long term degradation of primary forest as lowered standing vegetation and soil carbon stocks in secondary forests and also include forest management practices such as wood harvests' - correct?

Page 18 lines 4 to 9, confusing and somewhat redundant. You have a good test here but it gets lost in the language. With the DGVMs you run two scenarios, one of change (described as the "first" runs in line 4) and one with fixed invariant pre-industrial land use cover (described as the "second" in line 6). But in line 7 you say "allowing the models to estimate, by difference with the first simulation". Confusion arises over which models - first, second or all - you refer to at the start of that sentence and therefore which differences you actually calculate. At line 8 you use the word "prescribed" when in fact in both the change and invariant simulations you `prescribe` land use forcing. From the clear statement on line 9 a reader finally understands both the sequence and the purpose. I also don't understand why in line 6 you write "as further described below" in a location and with punctuation that clearly implies that the phrase applies specifically to the second set of invariant simulations when in fact below (where 'below' as I understand it could refer both to Section 2.2.3 on uncertainty in ELUC or more likely to Sections 2.5.1 and 2.5.2 on using DGVMs to diagnose SLAND) you provide additional details about primarily the first active-change DGVM simulations.

I suggest a gentle rewrite along these lines:

`Two sets of simulations were performed with the DGVMs. A first set included historical representations of changing land cover distributions, climate, atmospheric CO2 concentrations, and N deposition. A second set adopted time-invariant preindustrial land cover distribution. Because dynamic evolution of vegetation biomass and soil carbon pools occurs in the first

simulations but not in the second, ELUC is diagnosed in each model as the difference between these two simulations. As discussed in more detail in Section 2.5.2, we only retain model outputs with positive ELUC during the 1990s (Table A1). Using differences between the two DGVM simulations to diagnose ELUC allows the DGVMs to account for the loss of additional sink capacity (around 0.3 GtC yr-1; see Section 2.7.3) while bookkeeping models do not.`

Page 18 line 23:  Eliminating the first two words of this sentence would correct the singular/plural problem and clarify the intent.  E.g., instead of 'We assess an uncertainty of xxx reflects' a reader would see 'An uncertainty of xxx reflects our best value judgement …'.

Page 19 lines 4,5: Confusion here.  Earlier you said you used GFED and GFED 4s for fire occurrence and emissions, correctly referencing van der Werf et al. 2017.  But, as referenced in van der Werf et al. 2017, GFED 4s derives from MODIS MCD64A1 c6 burned area determinations. Here, however, we find a different MODIS product, MCD14ML, accompanied by both the Giglio and van der Werf citations.  Please check!  Different MODIS product or typo?  There is also a right parenthesis ')' missing here somewhere.

Page 19, line 6: "fires season" or fire seasons?  Following in line 7, most burning, at least as detected by satellite, occurs in tropical Africa.  Deforestation and associated emissions do or do not show the same geographic pattern as burned areas?  Why or why not?

Page 19, Section 2.3.1.  The legend to Figure 1 quotes Scripps as 1958 to 1979 followed by NOAA from 1980 to 2018.  Those dates differ slightly from what we read here?  Figure 1 legend also refers to an essential overlap intercalibration period of more than 20 years, not mentioned here?

Page 19 line 29: "Marine Boundary Layer" here but "marine boundary layer" earlier in line 16. Chose one appropriate capitalisation scheme and use it consistently?

Page 20 lines 2,3: Replace existing with: 'The second and third uncertainties, summed in quadrature, add up to 0.085 on average'?

Page 20 lines 6 to 12: Confusing, redundant and disorderly.  Small changes could give a substantially better accounting:

'… assess the total atmospheric CO2 burden. An excess measured at any station will not exactly track changes in total atmospheric burden, with offsets in magnitude and phasing owing to finite rates of vertical mixing and stratosphere-troposphere exchange. For example, excess CO2 from tropical emissions may arrive at distant stations in the network after a delay of months or more and the signals will continue to evolve as the excess mixes throughout the troposphere and stratosphere. This effect must be very small on …'

Page 20 line 20: for years prior to 1980?

Page 21 line 17: 'first halves of years show' or 'first half of the year shows'?

Page 22 line 1: statistical "spread" or geographic "spread" or both?  Following to line 2, do we need a citation to support "… methods that are deemed most reliable for the assessment …"? Following further from line 2 into line 3 "IPCC did not revise its assessment in 2013" - what does this mean?  No further attention to the issue?  I think you mean that IPCC confirmed their confidence in these ocean sink estimates as recently as 2013?

Page 22 line 7 "This estimate"  Which estimate?  The sentence follows an extensive list of observational constraints, but you expect a reader to remember the earlier IPCC estimate?  Give the reader a little more guidance here?

Page 22 lines 18,19: Confusing.  Suggest instead 'Several other ocean sink products based on observations are also available but they continue to show large unresolved discrepancies with observed variability.'

Page 23 line 9: '… in the source of the atmospheric forcing data, spin up strategies, and  resolution of the oceanic physical processes …'?

Page 23 lines 12,13 'changes in river organic carbon (discussed in Section 2.7.3)'?

Page 23 line 26 '… variability in ocean biogeochemistry models.'?

Page 25 line 12: 16 DGVM models (also 16 listed in Table 3 on page 63) but Figure S2 shows only 15?

Page 27 line 6: instead of rivers, suggest 'river's' or 'riverine' or 'fluvial'

Page 27 line 18: need a change in wording. Instead of "is missing in the combination of approached used here to estimate both" suggest 'is missing in the combination of approaches' or more simply 'is missing in the approaches used here …'.

Page 29 line 16: transferred in the open ocean or transferred to the open ocean?

Page 30 line 2: readers will perhaps know CMIP but likely will not know TRENDY for DGVM. Include some acronym definitions or references here?

Page 30 lines 7 to 9. 0.4 GtC per year over decade 2005 to 2014 gives 4 GtC per recent decade. Extrapolated over nearly 15 decades (1870 to 2017) does not give 20 GtC accumulated as stated here. Instead, closer to 60 GtC? The 20 GtC comes not from simple mathematical extrapolation of the most recent rate but from the time history of the OSCAR model ensemble? Including some scaling of the rate in past decades? Please check and explain. As written could allow or encourage errors by this reader and others?

Page 31 line 21: Estimates of total land fluxes (SLAND – ELUC) from the DGVMs is consistent or are consistent?

Page 32 line 3: 193%?

Page 36 line 15: were, not where

Page 39 line 19: total LUC emissions? total fire emissions? Help us out here, we can't tell what you mean.

Page 40 line 17: would further exacerbate or further exacerbates

Page 41 line 9: You have shown repeatedly that mean $B_{IM}$ remains small without a trend; at one point earlier you discussed the generally positive implications of a small imbalance, e.g. that all elements from all sources result in fundamental agreement on basic terms (Section 3.1.4 on Page 33). In the previous sentence you point out occasional higher $B_{IM}$ values and substantial unexplained variability. This represents a valid but still somewhat mixed message. Therefore, in this particular sentence, I suggest perhaps the word 'persistent' to replace the word "large": 'A persistent budget imbalance …' instead of "Such large budget imbalance …".

Page 41 lines 10, 11: This statement seems more negative or cautious than the assessment back in Section 3.1.4.

Page 41 lines 23 to 25: I understand what you mean here but the language tends toward opaque jargon. Could you write instead (shorter, clearer): '… is not new and highlights the difficulty to quantify complex processes (CO2 fertilisation, nitrogen deposition, climate change and variability, land management, etc.) that collectively determine the net land CO2 flux.'?

Page 41 line 27: … uncertainties that have   documented

Page 41 line 31 to Page 42 line 1: Not a proper sentence. Write instead something like: 'When assessing SLAND using DGVMs, uncertainties relate mostly to limitations in understanding and representation of fundamental processes as evidenced by …'?

Page 42 line 2: multiple studies

Page 42 line 10: 'quality of the energy statistics and of the emissions factors remain the largest source of uncertainties for'

Page 42 line 11: again, 'documented' instead of "demonstrated"?

Page 42 line 13: "atmosphere introduces  errors that have not yet been" If they haven't been quantified, we can't know if they represent significant additional errors?

Page 42 lines 17 to 20: awkwardly written paragraph with very mixed messages. For the first half, I think you want to say something like 'changes and improvements to the budget might either increase or decrease accuracy or variability and in any case might represent process that act neither globally nor simultaneously'? I can not agree with the next sentence as written "It is also possible that further yet unknown processes are not taken into account." Surely this statement, written so blandly and open-ended, invites confusion and criticism; it serves to undermine all that you have presented? Thinking in budget terms, we have almost absolute confidence in atmospheric CO2 concentrations - what better examined, reproduced and reproducible, tested, intercalibrated, verified, re-verified dataset do we have about our planet? From the remaining budget terms we can then estimate best and worst cases. Best cases, realised in many past years, show a budget imbalance of essentially zero. Once or twice we might experience the ignorant good luck that emission errors exactly compensate sink errors - thus giving a false budget imbalance close to zero - but as Figure 3 clearly shows for perhaps as many as 50 or 60 or more years of the past 120, we have zero budget imbalance. You say this several times: mean budget imbalance essentially zero with no trends. Either we have confidence in that record or we don't, but we find nothing there to suggest fundamental missing components of the carbon cycle? Taking now the worst case, based on the most recent decade, we could envision a budget imbalance as large as 3 GtC y-1 (working from Table 6 and assuming we err on the high side of all emissions and on the low side of all sinks). Setting aside for the moment that only in the late 1940s before Moana Loa CO2 records start, we might imagine one or two years where budget imbalance approached 3 GtC while in every other year of a nearly 120-year record we never measured an imbalance anywhere close to 3 GtC, what "unknown processes … not taken into account" can we imagine that would force a 30% error in either emissions or sinks? I know the sink side better than the emission side but I can not envision an omission or error that large in emissions or sinks! Some widespread but hidden emission source? Inversions and isotope analyses would already have identified such a source? Processes in soil or ocean, out of sight of satellites and difficult to reach by current sampling methods, that consume Gt of carbon per year? Dozens of researchers over at least four decades will have tried to attack current accounting and propose or discover new sinks, but (apparently) our understanding of basic carbon cycle processes remains valid. Do we have continuing errors - certainly! Does unexplained variability persist, to a degree that clouds our predictive skill - absolutely! But do we miss a fundamental term in the carbon cycle of today's planet? No, and everything GCP and others have published and reported for more than a decade reinforces that point. I suggested at the top several interesting topics for discussion; I contend that this paragraph and particularly this sentence need serious and substantial revision.

Page 42 paragraph beginning at line 21: I applaud this inclusion but this paragraphs repeats almost exactly what you say again on Page 43 in lines 17 to 20, and you say it better (shorter, clearer) there.

Page 43 lines 2 to 4: Awkward, imprecise. Suggest something such as: 'Merging these terms has limited usefulness, however, as it involves mixing direct and indirect processes and bringing in errors from other components; source, sink and uncertainty signals become more difficult to interpret.'

Page 43 line 8: "different boundary limits among models." What does this mean? Geographic boundaries, e.g. at coastlines between land and ocean? Process boundaries, e.g. how various model treat (or do not treat) soil carbon respiration or deforestration/reforestration? Component boundaries among atmosphere, land and ocean? Specificity and clarification necessary.

Page 43, lines 9 and 10: Does this sentence refer to land fluxes only or to carbon fluxes (land and ocean) generally?

Page 43, line 11: Annual estimates, which continue to generate more uncertainty than longer-term means of trends, may improve with improvements in data quality and timeliness.

Page 43 line 17: If you had truly large discrepancies you wouldn't construct long-term much less annual budgets. You should eliminate the word 'large'?

Page 43 lines 17 to 20: Good sentence about why you introduced model metrics. Keep this and eliminate the highly similar paragraph on page 42?

Page 44 line 13: Extraneous characters at the end of that line.

Page 44 line 29: that requires a careful compilation and synthesis of observations, statistical reports and model products.

Page 45 line 8: change of which sink? Ocean, land, both? Also, you should refer to ocean heat content rather than ocean temperatures?

Page 45 line 9: better = improved, more frequent, more precise?

Page 45 line 12: If your budget is correct, the sinks do more than affect, they determine?

Page 45 line 15: add some urgency here? 'Building this scientific understanding to meet urgent social needs depends on more frequent, robust and transparent …'?

Page 45 line 23, 24; here; and F Joos … Likewise line 25: measurements; and V. Kitidis …

Page 46 line 3: profile data; and Japan Met….

Page 46 line 5: evaluations

Page 47 line 22: something wrong or missing with the citation here? Using doi would fix these problems?

Page 48 line 5: "submitted" used twice

Page 49 line 3: spelling and formatting errors in Collier et al. reference

Page 50 line 50: formatting error in the Hooijer reference

Page 51 line 25: Hurtt et al. in prep has an update by now?

Page 51 line 43: spelling and formatting errors in Joetzjer et al. reference¨

Page 52 line 37: capitalisation errors

Page 54 line 38: "in review" used twice

Page 55 line 10: formatting error

Page 55 line 28: "2018" used three times

Page 56 line 23: you do not need "2018" twice here

Page 56 line 27: spelling / formatting error

Page 57 line 3: formatting error

page 57 line 13: formatting / spelling error

Page 57 line 15: formatting / spelling error

Page 58 line 7: formatting / spelling error

Page 58 line 50: formatting / spelling error

Page 59 line 7: van der Werf paper published in 2017, please update the reference

Page 70 line 2: should this read "of at least $\pm$ 0.3 GtC y$^{-1}$"?  (As used for $G_{ATM}$ later in this table)

Page 71 line 8: do we need definition of Vcmax here?

Page 77: Surprised to not see USA National Science Foundation listed in Table A5 because you several times mention NCAR and CCSM (and also HIPPO, flown on an NSF aircraft) in text, other tables or references?

Page 89 Figure S1:  What should we learn from this Figure?  To avoid the IPSL model?  0% IAV variability would indicate perfect fidelity with SOCAT fluxes?  This figure as well as the text in Section 3.1.3 show or describe difference in percentage ranges, e.g. 30 to 50% compared to 50 to 70%.  The authors offer no explanation (e.g. why should tropics differ from higher latitudes), no indication of ranges or uncertainties, and no conclusions.  As a reader nothing useful leaps out at me here?  Will a GOBM owner/operator extract something useful?  Absent some clues from the authors, one wonders.

Page 90 Figure S2: Again, not much explanation from the Figure nor from the text.  Familiar model by parameter matrix, but what should a reader conclude.  Does the figure belong better to the DGVM owner/operators?  Can / should the authors provide some general clues?  The figure legend basically lists the validation data sets and the difference between 'absolute' and 'relative scores' but not much about what we should learn or expect.  That no model offers reasonable skill at LAI or NEE but that most of them can replicate run-off?  The text and figure fail to convince me of relevance to global carbon budgets?  Do DGVM need to score high on all these parameters to better predict carbon sinks?

Page 91 line 7: land or ocean data, rather than "land of ocean"?  Again, what should we conclude from this figure or from the associated text?  That all four inversion techniques vary widely (and wildly) among themselves and in relation to aircraft data over most of the globe except perhaps in the northern half of the northern hemisphere?  Has CGADIP done quality control or error analysis on any of the aircraft data profiles?  Why do we see no error bars?  What combination of profiles and sampling rates allows one to extract more than 200k data points (legend of Figure S3) from 40-some aircraft campaigns (Table A6)?  Trust the authors to know their product and various means of validation but one really stretches to find relevance or utility in this figure and this data?

---

## Author Response (AR2)

We thank all three reviewers for their insights and comments on our manuscript, which have greatly helped to clarify our manuscript. Please find below a point-by-point response.

| Reviewer's comments: | Response: |
|---|---|
| Page 13 line 8. I am not familiar with Holt Winters smoothing, and the text of "exponential smoothing with multiplicative seasonality " does not really brings further understanding. Can the authors say in understandable language why Holt Winters is chosen and what it effectively does? | We added: "This iterative method produces estimates of both trend and seasonality at the end of the observation period that are a function of all prior observations, weighted most strongly to more recent data, while maintaining some smoothing effect." |
| Page 17/18 lines 30-1. Is this an improvement? Has the impact of using this resolution been quantified somewhere? | The statement has been clarified in the text. The climate forcing for the DGVMs has always been provided at the 0.5° resolution. The statement about higher resolution only refers to the 6-hourly forcing (JRA55 vs NCEP) that is combined with the CRU monthly datase to provide the sub-monthly variability needed by most DGVMs. JRA is at a 0.5° resolution as is CRU, while NCEP used in previous years was at the coarser 2.5° resolution. Having now both monthly and sub-monthly forcings at the same 0.5° resolution simplifies the technical merging of datasets, but we expect the impact to be relatively minor. |
| Page 21 Line 19. Strictly what you call here uncertainty is climatic variability. | Indeed we are using climate variability as a proxy for uncertainty in the projection. We clarified this as follows: "Uncertainty is estimated from past variability using the standard deviation of the last 5 years' monthly growth rates." |
| Page 23 lines 15-27. I like this metric very much, but the description of it is rather dense. Maybe have a look, reformulate and make the text a little less dense. | We have added the following text at the end of the first paragraph, and have revised the second paragraph to clarify the findings: "The metric provides a measure of the mismatch between observations and models or flux products on the x-axis as well as a measure of the amplitude of the interannual variability on the y-axis. A smaller number on the x-axis indicates a better fit with observations. The amplitude of the interannual variability of SOCEAN (y-axis) is calculated as the temporal standard deviation of the CO2 flux time-series. " |
| Page 26 line 27-28. Why is only the land sink adjusted and not the ocean sink as well? | The adjustment is made to the land sink because the vast majority of FF emissions occur over land. We have considered making the adjustment proportional to the fraction fossil fuel emissions over land/ocean (92/8%), but we realized this is again a model dependent feature, and also raises the question on how to assign the adjustments spatially (NH, Tro, SH). We decided that this requires some further work from the inverse modeling group, and we would rather first introduce the simple, but most needed, adjustment to the global totals of each model. We clarified the sentence by adding: " (where most of the emissions occur)". |
| Page 32. The skill score on the DGVM's is nice. But it would also be honest to say that the models perform poorly, specifically on important aspects of the carbon cycle. Even though, that is what they are used here for. Now the text is a bit too non-non-committal to my taste. | A comprehensive evaluation paper on the DGVM benchmarking is in preparation (Robertson et al.,) which will elaborate further on the performance of individual DGVM against individual data-sets. Our main aim at inclusion of benchmarking here is to start providing a long-term assessment tool to ensure model improve through time, as invariably individual models add processes etc and are refined each year, rather than to explicitly criticise models for poor performance. Explicit evaluation through ilamb will enhance individual modelling groups ability to assess their own model and critically identify weaknesses. |
| Page 36. Lines 4-10. Yes it is long known that the NH land fluxes are poorly resolved. In fact I would make the point that despite, say 10-15 years of active research, we still have not been able to resolve one of the main uncertainties/unknown in the inversions/continental scale carbon cycle. | We agree that the quantification of the total NH land sink, and its attribution to different continental land-masses remains challenging. But although we have perhaps not yet "resolved" this uncertainty, 10-15 years of research has certainly brought us some progress. This refers to the convergence of DGVM and inverse modeling results to the level stated in this paragraph (< 1 PgC/yr on NH totals, and <0.5 PgC/yr on continental scale), and the now better resolved flux difference between the NH and the Tropics (which is now known to be more likely close to neutral than a large source or sink). Also, attribution of regional carbon cycle anomalies to specific climate anomalies is much improved. Note that much of this progress was done without the actual large-scale investment in new observation sites that would perhaps have helped most. This point is now brought forward in the discussion, rather than in the result section as suggested by the reviewer. Further in this study, the compatibility of inversions total land CO2 fluxes with DGVM anthropogenic land C storage changes has been made more accurate by subtracting from inversions the natural atmosphere-to-land CO2 flux transferred to rivers. Furthermore in this study, the compatibility of inversions total land CO2 fluxes with DGVM anthropogenic land C storage changes has been made more accurate by subtracting from inversions the natural atmosphere-to-land CO2 flux transferred to rivers |
| Page 43. Lines 17-20 can be deleted. Has been stated before. | The discussion has been revised in depth and the overlap deleted. |

| | |
|---|---|
| Page 45. The conclusion section is rather obsolete. I would suggest to close off the paper with the data availability section (5) and add, only if it has not been said before, some of the conclusion statements in the discussion. This would improve to my mind the overall readability of the document, that, it has been said before, is becoming quite long. | The discussion has now been revised entirely to better highlight the key issues raised by this paper. The conclusion has been shortened, but we would like to keep it because although it is very general, it summarises what this paper tries to achieve. We hope the combined revisions to the discussion and conclusion addresses this comment. Note that the order of the information (Discussion, Data availability, and Conclusion) was requested by the Editor in a previous version of this manuscript. |

| | |
|---|---|
| We thank all three reviewers for their insights and comments on our manuscript, which have greatly helped to clarify our manuscript. Please find below a point-by-point response. | |

| Reviewer's comments: | Response: |
|---|---|
| 1) As a note on the use of 1870 as a reference year, the new IPCC 1.5°C Special Report seems to be defining "pre-industrial temperature" as the 1850-1900 average (rather than 1860-1880 as some have previously done) — this suggests that there may be utility in updating the reference year at some point to 1875 to represent the mid-point of the pre-industrial temperature range. | It is slightly more complicated and upon reflection, we think it best to wait for the reference period to be established by IPCC AR6. IPCC AR5 only used 1870 (average of 1860-1880) as reference period for TCRE (SPM figure 10) and the quantification of carbon budgets compatible with 2°C. This was dictated by the CMIP5 climate models simulations that had 1860 as a starting date. AR5 (as IPCC SR1.5) used 1850-1900 as reference period for observed temperature. We note that the CMIP6 protocol starts historical simulations in 1850, so the reference period for carbon budget in AR6 might be slighly different from AR5. It might be 1850-1870 (20 years reference period as usually done for climate models), or 1850-1900 as currently used for observed temperature, we do not know yet. At the moment we would rather keep 1870, as in AR5, and update if needed in agreement with AR6. We note that changing the reference period from 1870 to 1875 as suggested by the reviewer would reduce the carbon budget by about 5GtC, within the error bars of our estimate. |
| 2) The difference between E_LU for DGVMs vs Book-keeping models in the recent decade is obviously striking (and well discussed), though I didn't come away with any explanation for the mechanism behind this difference. Is there some change in LU patterns (used to drive DGVMs) that explains the increase from the previous decades? Or is this a reflection of some response to climate change? Some speculation could help here (even if that is all it is at this point). | We added an explaination at the end of section 3.2.1 as follows: "Larger emissions are expected increasingly over time for DGVM-based estimates as they include the loss of additional sink capacity, while the bookkeeping estimates don't. The LUH2 dataset also features large dynamics in land use in particular in the tropics in recent years, causing higher emissions in DGVMs and BLUE than in H&N." |
| 3) Related to this: | |
| - On page 18 (lines 19-20), note that DGVMs and BK models agree prior to the recent decade (but not for the most recent year / decade) | See response to the previous comment |
| - On page 18, lines 11-12: Is this loss of additional sink capacity the same as the finding that some have shown that simulated LU emissions in models tend to increase as a function of increasing CO2, since CO2 fertilization acts equally in "agricultural" areas, and maintaining constant cropland/pasture areas in a model results in higher emissions when CO2 is higher compared to when it is lower. I think it is kind of the same process at work but I am not completely sure... | Yes this is the same process. No changes required. |
| 4) Section 3.2: It is worth clarifying here that the values chosen to reflect E_LU and S_LAND in the budgets for the recent decade and for 2017 reflect different (and not necessarily consistent) methods. i.e. E_LU is from Book-keeping models (which do not include lost sink capacity), whereas S_LAND is from DGVM (which does include lost since capacity). It would be worth justifying this choice (notably the choice of using the BK estimate of emissions rather than DGVMs) and noting the inconsistency more explicitly to avoid confusion comparing numbers between Tables 5 and 6. | We added the explanation below to the text in Section 3.2. A fully consistent budget based on both DGVMs and bookkeeping models for ELUC (after correcting for the loss of additional sink capacity in the bookkeeping approach) is planned once DGVMs are more comprehensive and similar in their coverage of land use practices. "The budget imbalance (Table 6) and the residual sink from global budget (Table 5) include an error term due to the inconsistency that arises from using ELUC from bookkeeping models, but SLAND from DGVMs. This error term includes the fundamental differences between bookkeeping models and DGVMs, most notably the loss of additional sink capacity. Other differences include: an incomplete accounting of LUC practices and processes in DGVMs, while they are all accounted for in bookkeeping models by using observed C densities, and bookkeeping error of keeping present-day C densities fixed in the past. That the budget imbalance shows no clear trend towards larger values over time is an indication that the loss of additional sink capacity plays a minor role compared to other errors in SLAND or SOCEAN (discussed in 3.1.4)." |
| 5) In the 2018 emissions projections for China, US, EU and India, why not add red dots to Figure 5 (similar to the global value)? Also, it is striking to me that US emissions are anticipated to increase by 2.2%, which is a very large departure from the recent decreasing trend. It would be worth noting this on page 39 (lines 3-4). | The interpretation of the 2018 projection and their uncertainties is discussed in detailed in two commentaries that will appear simultaneously in Nature and in Environmental Research Letters. We keep here the focus on the description of the methodology and the results, aligned with the scope of ESSD. We have removed the projection from Figure 5a to make it consistent with the other panels. |
| 6) Another potentially missing process (that is not mentioned here at all I don't think) is the effect of terrestrial weathering. Of course this is a very small carbon flux, but some recent model studies suggest the possibility of small increases over the historical period (due to warming, increased mid/high latitude runoff, vegetation expansion ...), which might account for some portion of the cumulative carbon imbalance shown here. | Indeed the CO2 consumption due to silicates and carbonates weathering reaction is a very small background flux, estimated to be a sink of atmospheric CO2 of 0.1 -0.44 Pg C y-1 (Hartmann et al 2009). To our knowledge, the only quantification of C sequestration due to changes in weathering during the 20th century estimated a very minor sink of on average 0.005-0.0021 Gt C y-1 due to increased co2 consumption by chemical weathering as well as enhanced biomass production due to increased phosphorus release (Goll et al. 2014). This is neglected here as it is much less other components that are discussed in Section 2.7.

Reference: Climate-driven changes in chemical weathering and associated phosphorus release since 1850: Implications for the land carbon balance. DS Goll, N Moosdorf, J Hartmann, V Brovkin - Geophysical Research Letters, 2014, https://doi.org/10.1002/2014GL059471 |

We thank all three reviewers for their insights and comments on our manuscript, which have greatly helped to clarify our manuscript. Please find below a point-by-point response.

| Reviewer's comments: | Response: |
|---|---|
| Check capitalisation! You use Tropics and tropics (often within the same paragraph!). You use "Marine Boundary Layer" and "marine boundary layer" on the same page (Page 19). To make your synthesis more effective you also need to impose consistent spelling and punctation. Make these changes now to save time at the proofreading stage? Data set vs datasets - please choose and make consistent. | Thank you, we reviewed for consistency. |
| Reader encounters reference to Supplement several times, but reviewer finds no supplement to evaluate! Three supplementary figures actually appear in an Appendix, albeit referenced and labelled as S1, S2 and S3. Meanwhile, Tables A1 to A6 carry an 'Appendix' label but do not appear with Figures S1, S2 and S3 in an Appendix. I might understand why these particular tables carry an Appendix label and why these three figures should reside in an Appendix: to include technical details not of interest to general readers but relevant to carbon cycle colleagues? If true (or for whatever other valid reason) we need some revision, reorganisation and re-labelling here! If you want a proper Appendix, label and put all tables and figure in it as appropriate. | We introduced Appendix A and Appendix B |
| Please adopt and implement an informative flux labelling convention. For many in our earth systems community, fluxes upward carry positive sign, downward a negative sign. Not everyone follows this convention (no surprise) and I recognise that you specify fluxes in Gt per year rather than watts per metre squared (unfortunately those who measure W/m2 s-1 generally do not follow the positive up negative down guidelines) but you definitely have fluxes (emissions) going gravitationally upward and fluxes (sinks, land and ocean) going gravitationally downward. Therefore when you specify, e.g. on Page 35 (and in many many other places in the manuscript): "total atmosphere-surface CO2 flux from process models for 2008-2017 is 3.7 + 1.2 GtC yr-1" you either need to carefully specify in every case which direction you mean - in this case you would write 'atmosphere-to-surface' rather than "atmosphere-surface" - or a reader needs to recognise from the sign (positive or negative) which direction the fluxes flow. Figure 8 highlights this need but you face the problem throughout the manuscript that readers do not know from your text or numbers in which direction the fluxes you cite actually operate. In the new Figure 2 you show, for example, ocean-to-atmosphere natural cycle flux of 90 up and atmosphere-to-ocean return flux of 90 down (and land with atmosphere likewise, 120 up matched by 120 down) with human impact superimposed but in neutral units. You should thus describe an 'ocean bidirectional exchange with atmosphere of 90 GtC y-1' or you need to show 90 up as positive and 90 down as negative. The challenge becomes acute in Figure 8. Globally, you show combined land and ocean downward fluxes (sinks!) in positive numbers (e.g. currently roughly +5 GtC y-1) but you also show that in the 1960s and in occasional years since net land flux has gone negative (upward, land as a source!), as low as (or as much as) -1.5 to perhaps -2.0 GtC y-1. Globally, ocean fluxes stay downward/positive values/sink, but when you sort by latitude, the tropical ocean (which, disconcertingly, has its own ordinate range different to any of the other 11 panels in this figure) shows a clear ocean source but now designated by negative fluxes! For purposes of this figure you have in fact adopted a convention - positive downward and negative upward - but you have not declared it and you do not follow it consistently in the text. I agree that you can extract a lot of useful information from the flux products but you owe readers much more clarity on which direction you mean and when. You can implement this clarification in text probably more easily than in numbers and equations, but we need a good response on this issue.You have already done something similar for the budget imbalance, specifying it as positive some years and negative in others, because as the mathematical residual of equation 1 it will necessarily have small positive values some years but small negative values in other years. Note also that when you specify rates of change in % (e.g. change in growth rates of CO2 emissions as on Page 34), you carefully and correctly specify either increase or decrease, using language to indicate the direction of change, or - as in line 24 on Page 37 - you carefully use negative percentages. We need similar careful language for flux descriptions! | We have adopted the following convention: numbers representing 'emission', 'sinks', and 'atmosphere-to-surface' are positive. Increasing or decreasing growth rates carry the signs + or - for additional clarify. We have revised the paper and tried to apply this systematically and to clarify the text. The arrows in Figure 2 clearly show the direction of the fluxes. We think adding signs would confuse rather than clarify the reader. We will do also further checks on clarity of flux direction at the proof stage. |

| Comment | Response |
|---|---|
| After multiple readings, I still find myself confused by the Discussion. Here you have a chance to 'comment' on strengths and weaknesses and to recommend (or at least list) possible improvements. As I read it, the discussion omits or takes a too-modest approach to positive features and strengths. This reader finds the discussion almost apologetic in places, when in fact the authors could expand and expound on this statement (from Page 33 line 8): "the near-zero mean and trend in the budget imbalance is an indirect evidence of a coherent community understanding of the emissions and their partitioning on those time scales". What would make the evidence more direct? Do the authors feel that longer (paleo?) time scales (e.g. ice core data) have something useful to lend to this accounting? The discussion also fails to follow-up on specific issues and uncertainties from the text, e.g. the so-called 'loss' of land sink capability in Section 2.7.4, the various factors that go into coming-year projections discussed in Section 3.4, the important outcomes and clear cautions about cumulative emissions accounting in Section 3.5, etc. You likewise have an opportunity (responsibility?) here to raise highly-relevant but - to this point - not-yet-discussed issues. E.g., how would we know if either the land or ocean sink changed significantly? What measurements continued over what time period, and what model confirmation, would we need to quantify and certify such a change? Such a discussion would add substantial relevance to your results and contentions on variability. Similarly, how quickly and with what precision could we - using these data and models and this budget approach - detect and confirm an intentional drop in emissions? How much induced carbon capture sink could we detect, and how soon? Of course an external record for these topics often exists in scientific literature but readers across a wide spectrum - the broad stakeholders listed in the Conclusion - and particularly a research community interested in but not directly contributing to this budget would very much benefit from explanatory and enlightened discussion on these issues here! This annual carbon budget effort also constitutes a unique, very necessary, very positive yet challenging mixing of model products with observational products; many readers will not encounter such a breadth of model intercomparisons with observations elsewhere. This particular version of the budget has attempted a positive step forward by introducing model metrics - twice (unfortunately) in the Discussion text and (not very helpfully) in Figures S1 and (better) in S2 - but inboth cases with a very tentative presentation. If metrics seem important, and better exposed and evaluated from the point of view of a carbon budget than from some future technical CMIP6 overview (or, god forbid, an IPCC report), shouldn't the authors spend more time in the discussion on what the metrics indicate (not much, evidently) or on what better metrics they might apply or imagine. In various specific sections below I have made so bold as to suggest alternate language, occasionally even for a full sentence or two. I looked closely at the Discussion with similar modifications in mind but feel that the actual text of the discussion belongs entirely and exclusively to the authors. I understand political pressures from various constituencies and also the fact that each annual budget runs right to the edge of release schedules, but I strongly urge the authors to reconsider this Discussion in light of the overall positive effort and some of the clear messages that emerge from that effort. | We have revised the discussion in depth to address this comment and focus on the new findings. However please note that this journal is a data journal that aims to document data and methodology, rather than really focus on the scientific findings. The research community is encouraged to use the data arising and publish further, more detailed analysis. |
| Good update and changes to Figure 2. | thank you |
| Apparently the word processing software allows line breaks at hyphens, including at the superscript hyphen yr-1. This results in unfortunate line breaks in many locations in the text leaving an orphan superscript 1 at the start of the following line. Authors and proofreaders will need to watch carefully for these errors in the final product. | ok |
| I don't know what ESSD or Copernicus recommend or require, but most manuscript reference lists today seem to use doi in place of pages and dates? Perhaps a change for a future version? | We have reviewed references to follow ESSD guidelines for Style. |
| Page 8 line 2: "estimates of EFF globally and nationally CO2 emissions" Something awkward here? Because you have defined global and national CO2 emissions as EFF in the prior sentence, here you only need to say 'global and national EFF'? | We clarified the sentence. |
| Page 8 line 8: "Official UNFCCC national inventory reports for 1990-2016 for the 42 Annex I countries in the UNFCCC (UNFCCC, 2018), as we assess these to be the most accurate estimates because they are compiled by experts within countries that have access to detailed energy data, and they are periodically reviewed." Awkward language here, allows some confusion. I think you mean (roughly) national inventory reports from UNFCC. We assess these national reports to provide the most accurate estimates because they are compiled by experts within countries who have access to detailed national energy data and because they receive periodic review by UNFCCC. | Changed as suggested. |
| Page 8 line 16: Again awkward language allows subject-predicate confusion. I think you mean 'provide more details for each dataset and describe additional modifications required to make the dataset consistent and usable.' | Changed as suggested. |
| Page 12, lines 8 to 13: Check punctuation here, I find commas or semicolons missing. Also "and (3)" followed by "and (4)". Authors will know best how they want to separate and punctuate this list. | Clarified. |
| Page 13 lines 5 and 7: You designate the two (oil and natural gas) PPAC references here in the text as 2018a and 2018b but in the reference list you show them both as PPAC 2018. Change something somewhere? | The references are correct. The referencing style puts the 2018a and 2018b at the end of the reference. |

| | |
|---|---|
| Page 13 line 18: "which is much less strongly seasonal because of strong weather variations" I had to think about this for a moment. I think you mean that shorter-term cold or warm weather periods dominate coal usage patterns due to their impact on residential and commercial heating or general energy use, as opposed to a smoother winter/high summer/low pattern of use? But, if correct, the same pattern or factors would apply to US and China as well? Already included in CCIA projections for China or EIA projections for US, but not in projections for Europe? If I understand the statement correctly, I find it strange that you only invoke it for Europe. If shortterm weather dominates, and varies widely, for Europe then your extrapolation from the prior year for missing winter months of the current year seems unreliable? | Yes, for USA any typical and expected seasonal patterns are accounted for by EIA. For EU, we find that historical emissions from coal have a much more variable seasonal variation signal than oil or gas, because the response to weather variations (hotter than normal summer, colder than normal winter) are relatively high compared to average seasonal variations (that winter is typically colder than summer). Coal also peaks in winter, but that peak is highly variable, and very dependent on just how cold the winter is (among other things). For China the projection are not from CCIA, but derived from their historical data, and we do not have monthly data, so cannot perform such analysis. |
| Page 14 line 27: "Additionally, they represents permanent degradation of forests by lower vegetation and soil carbon stocks for secondary as compared to the primary forests and forest management such as wood harvest." Awkward. They (meaning bookkeeping models) 'represent long term degradation of primary forest as lowered standing vegetation and soil carbon stocks in secondary forests and also include forest management practices such as wood harvests' - correct? | [p. 15, l. 7] Modified as suggested. |
| Page 18 lines 4 to 9, confusing and somewhat redundant. You have a good test here but it gets lost in the language. With the DGVMs you run two scenarios, one of change (described as the "first" runs in line 4) and one with fixed invariant pre-industrial land use cover (described as the "second" in line 6). But in line 7 you say "allowing the models to estimate, by difference with the first simulation". Confusion arises over which models - first, second or all - you refer to at the start of that sentence and therefore which differences you actually calculate. At line 8 you use the word "prescribed" when in fact in both the change and invariant simulations you `prescribe` land use forcing. From the clear statement on line 9 a reader finally understands both the sequence and the purpose. I also don't understand why in line 6 you write "as further described below" in a location and with punctuation that clearly implies that the phrase applies specifically to the second set of invariant simulations when in fact below (where 'below' as I understand it could refer both to Section 2.2.3 on uncertainty in ELUC or more likely to Sections 2.5.1 and 2.5.2 on using DGVMs to diagnose SLAND) you provide additional details about primarily the first activechange DGVM simulations. | We have clarified the text as follows: "Two sets of simulations were performed with the DGVMs. Both applied historical changes in climate, atmospheric CO2 concentration, and N deposition. The two sets of simulations differ, however, with respect to land use: one set applies historical changes in land use, the other a time-invariant preindustrial land cover distribution and preindustrial wood harvest rates. By difference of the two simulations, the dynamic evolution of vegetation biomass and soil carbon pools in response to land use change can be quantified in each model (ELUC). We only retain model outputs with positive ELUC, i.e. a positive flux to the atmosphere, during the 1990s (Table A1). Using the difference between these two DGVM simulations to diagnose ELUC means the DGVMs account for the loss of additional sink capacity (around 0.3 GtC yr-1; see Section 2.7.3), while the bookkeeping models do not." |
| I suggest a gentle rewrite along these lines: | |
| `Two sets of simulations were performed with the DGVMs. A first set included historical representations of changing land cover distributions, climate, atmospheric CO2 concentrations, and N deposition. A second set adopted time-invariant preindustrial land cover distribution. Because dynamic evolution of vegetation biomass and soil carbon pools occurs in the first simulations but not in the second, ELUC is diagnosed in each model as the difference between these two simulations. As discussed in more detail in Section 2.5.2, we only retain model outputs with positive ELUC during the 1990s (Table A1). Using differences between the two DGVM simulations to diagnose ELUC allows the DGVMs to account for the loss of additional sink capacity (around 0.3 GtC yr-1; see Section 2.7.3) while bookkeeping models do not.` | See reformulation above. |
| Page 18 line 23: Eliminating the first two words of this sentence would correct the singular/plural problem and clarify the intent. E.g., instead of 'We assess an uncertainty of xxx reflects' a reader would see 'An uncertainty of xxx reflects our best value judgement ...'. | Modified as suggested. |
| Page 19 lines 4,5: Confusion here. Earlier you said you used GFED and GFED 4s for fire occurrence and emissions, correctly referencing van der Werf et al. 2017. But, as referenced in van der Werf et al. 2017, GFED 4s derives from MODIS MCD64A1 c6 burned area determinations. Here, however, we find a different MODIS product, MCD14ML, accompanied by both the Giglio and van der Werf citations. Please check! Different MODIS product or typo? There is also a right parenthesis ')' missing here somewhere. | The sentence was clarified as follows: "Peat burning as well as tropical deforestation and degradation are estimated using active fire data (MCD14ML; Giglio et al. (2016)), which scales almost linearly with GFED (van der Werf et al., 2017), and thus allows for tracking fire emissions in deforestation and tropical peat zones in near-real time." |
| Page 19, line 6: "fires season" or fire seasons? Following in line 7, most burning, at least as detected by satellite, occurs in tropical Africa. Deforestation and associated emissions do or do not show the same geographic pattern as burned areas? Why or why not? | Fire season (singular). While most burning indeed occurs in Africa, those fires are mostly savanna fires which are not considered here as they are not a net source of CO2. Here we only address fires associated with deforestation and tropical peatland burning. About 80% of deforestation emissions over the last two decades have occurred in tropical regions of Asia and America and only 20% in Africa (https://www.geo.vu.nl/~gwerf/GFED/GFED4/tables/GFED4.1s_C.txt), all of the tropical peat emissions occur in Asia. Therefore, capturing tropical Asia and America correctly, which is possible given that by October we cover most of the fire season in these regions, is a good approximation of total emissions by deforestation/degradation fires and land-use-induced peat fires globally. |
| Page 19, Section 2.3.1. The legend to Figure 1 quotes Scripps as 1958 to 1979 followed by NOAA from 1980 to 2018. Those dates differ slightly from what we read here? Figure 1 legend also refers to an essential overlap intercalibration period of more than 20 years, not mentioned here? | We harmonised the 1979/1980 date in the text. For the start year, the concentration data starts in 1958 while the growth rate starts in 1959 so these dates are correct. The 20 years is invoked in Fig.1 because we adjusted the concentrations ourselves to account for the different networks before and after 1980. For the growth rate, this was done directly by Ballantyne et al. 2012 updated by Dlugokencky and Tans (2018) which we cite. |
| Page 19 line 29: "Marine Boundary Layer" here but "marine boundary layer" earlier in line 16. Chose one appropriate capitalisation scheme and use it consistently? | Done. |
| Page 20 lines 2,3: Replace existing with: 'The second and third uncertainties, summed in quadrature, add up to 0.085 on average'? | Done. |

| | |
|---|---|
| Page 20 lines 6 to 12: Confusing, redundant and disorderly. Small changes could give a substantially better accounting: '… assess the total atmospheric CO2 burden. An excess measured at any station will not exactly track changes in total atmospheric burden, with offsets in magnitude and phasing owing to finite rates of vertical mixing and stratosphere-troposphere exchange. For example, excess CO2 from tropical emissions may arrive at distant stations in the network after a delay of months or more and the signals will continue to evolve as the excess mixes throughout the troposphere and stratosphere. This effect must be very small on …' | We have replaced the text with a shorter version of the same argument: "In reality, $CO_2$ variations measured at the stations will not exactly track changes in total atmospheric burden, with offsets in magnitude and phasing due to vertical and horizontal mixing. This effect must be very small on decadal and longer time scales, when the atmosphere can be considered well mixed." |
| Page 20 line 20: for years prior to 1980? | Corrected. |
| Page 21 line 17: 'first halves of years show' or 'first half of the year shows'? | Clarified. |
| Page 22 line 1: statistical "spread" or geographic "spread" or both? Following to line 2, do we need a citation to support "… methods that are deemed most reliable for the assessment …"? Following further from line 2 into line 3 "IPCC did not revise its assessment in 2013" - what does this mean? No further attention to the issue? I think you mean that IPCC confirmed their confidence in these ocean sink estimates as recently as 2013? | It is indeed the uncertainty, not the spread. Reference added to Denman et al (2007) for 'reliable methods'. Yes, IPCC confirmed in 2013, we reformulated and cited. |
| Page 22 line 7 "This estimate" Which estimate? The sentence follows an extensive list of observational constraints, but you expect a reader to remember the earlier IPCC estimate? Give the reader a little more guidance here? | Clarified. |
| Page 22 lines 18,19: Confusing. Suggest instead 'Several other ocean sink products based on observations are also available but they continue to show large unresolved discrepancies with observed variability.' | Modified as suggested. |
| Page 23 line 9: '… in the source of the atmospheric forcing data, spin up strategies, and in the resolution of the oceanic physical processes …'? | We are not sure what the reviewer is suggesting. Yet, we have clarified the text. This is not a model intercomparison exercise, but independent best estimates of the ocean carbon sink by different modelling groups. Models may differ in their choice of the atmospheric forcing data set (JRA55, NCEP,...), in their grid resolutions and in the length of the spin-up. We note though, that even in model intercomparison exercises, such as CMIP5 or 6, models differ in their grid resolutions and spin-up strategies. |
| Page 23 lines 12,13 'changes in river organic carbon (discussed in Section 2.7.3)'? | Modified as suggested. |
| Page 23 line 26 '… variability in ocean biogeochemistry models.'? | Modified as suggested. |
| Page 25 line 12: 16 DGVM models (also 16 listed in Table 3 on page 63) but Figure S2 shows only 15? | At the time of resubmission, we had just received the gridded results from the last model. We are working to complete this figure. |
| Page 27 line 6: instead of rivers, suggest 'river's' or 'riverine' or 'fluvial' | Modified as suggested. |
| Page 27 line 18: need a change in wording. Instead of "is missing in the combination of approached used here to estimate both" suggest 'is missing in the combination of approaches' or more simply 'is missing in the approaches used here …'. | Modified as suggested. |
| Page 29 line 16: transferred in the open ocean or transferred to the open ocean? | Clarified. |
| Page 30 line 2: readers will perhaps know CMIP but likely will not know TRENDY for DGVM. Include some acronym definitions or references here? | Clarified. |
| Page 30 lines 7 to 9. 0.4 GtC per year over decade 2005 to 2014 gives 4 GtC per recent decade. Extrapolated over nearly 15 decades (1870 to 2017) does not give 20 GtC accumulated as stated here. Instead, closer to 60 GtC? The 20 GtC comes not from simple mathematical extrapolation of the most recent rate but from the time history of the OSCAR model ensemble? Including some scaling of the rate in past decades? Please check and explain. As written could allow or encourage errors by this reader and others? | Clarified. |
| Page 31 line 21: Estimates of total land fluxes (SLAND – ELUC) from the DGVMs is consistent or are consistent? | Clarified. |
| Page 32 line 3: 193%? | Corrected. |
| Page 36 line 15: were, not where | Corrected. |
| Page 39 line 19: total LUC emissions? total fire emissions? Help us out here, we can't tell what you mean. | Revised to clarify (it's ELUC) |
| Page 40 line 17: would further exacerbate or further exacerbates | Corrected. |
| Page 41 line 9: You have shown repeatedly that mean BIM remains small without a trend; at one point earlier you discussed the generally positive implications of a small imbalance, e.g. that all elements from all sources result in fundamental agreement on basic terms (Section 3.1.4 on Page 33). In the previous sentence you point out occasional higher BIM values and substantial unexplained variability. This represents a valid but still somewhat mixed message. Therefore, in this particular sentence, I suggest perhaps the word 'persistent' to replace the word "large": 'A persistent budget imbalance …' instead of "Such large budget imbalance …". | Modified as suggested. |
| Page 41 lines 10, 11: This statement seems more negative or cautious than the assessment back in Section 3.1.4. | The discussion was complete revised. |
| Page 41 lines 23 to 25: I understand what you mean here but the language tends toward opaque jargon. Could you write instead (shorter, clearer): '… is not new and highlights the difficulty to quantify complex processes (CO2 fertilisation, nitrogen deposition, climate change and variability, land management, etc.) that collectively determine the net land CO2 flux.'? | Yes indeed this is much clearer. Modified as suggested. |
| Page 41 line 27: … uncertainties that have been [deleted: a demonstrated documented] | Corrected. |
| Page 41 line 31 to Page 42 line 1: Not a proper sentence. Write instead something like: 'When assessing SLAND using DGVMs, uncertainties relate mostly to limitations in understanding and representation of fundamental processes as evidenced by …'? | Modified as suggested. |
| Page 42 line 2: multiple studies | Corrected. |
| Page 42 line 10: 'quality of the energy statistics and of the emissions factors remain the largest source of uncertainties for' | Modified as suggested. |
| Page 42 line 11: again, 'documented' instead of "demonstrated"? | Modified as suggested. |

| | |
|---|---|
| Page 42 line 13: "atmosphere introduces additional errors that have not yet been" If they haven't been quantified, we can't know if they represent significant additional errors? | This is correct, although preliminary analysis by the author team suggest that this specific source of uncertainty might be close to or larger than 0.3. We have not yet done this estimate formally but would like to keep highlighting the possibility here. |
| Page 42 lines 17 to 20: awkwardly written paragraph with very mixed messages. For the first half, I think you want to say something like 'changes and improvements to the budget might either increase or decrease accuracy or variability and in any case might represent process that act neither globally nor simultaneously'? I can not agree with the next sentence as written "It is also possible that further yet unknown processes are not taken into account." Surely this statement, written so blandly and open-ended, invites confusion and criticism; it serves to undermine all that you have presented? Thinking in budget terms, we have almost absolute confidence in atmospheric CO2 concentrations - what better examined, reproduced and reproducible, tested, intercalibrated, verified, re-verified dataset do we have about our planet? From the remaining budget terms we can then estimate best and worst cases. Best cases, realised in many past years, show a budget imbalance of essentially zero. Once or twice we might experience the ignorant good luck that emission errors exactly compensate sink errors - thus giving a false budget imbalance close to zero - but as Figure 3 clearly shows for perhaps as many as 50 or 60 or more years of the past 120, we have zero budget imbalance. You say this several times: mean budget imbalance essentially zero with no trends. Either we have confidence in that record or we don't, but we find nothing there to suggest fundamental missing components of the carbon cycle? Taking now the worst case, based on the most recent decade, we could envision a budget imbalance as large as 3 GtC y-1 (working from Table 6 and assuming we err on the high side of all emissions and on the low side of all sinks). Setting aside for the moment that only in the late 1940s before Moana Loa CO2 records start, we might imagine one or two years where budget imbalance approached 3 GtC while in every other year of a nearly 120-year record we never measured an imbalance anywhere close to 3 GtC, what "unknown processes … not taken into account" can we imagine that would force a 30% error in either emissions or sinks? I know the sink side better than the emission side but I can not envision an omission or error that large in emissions or sinks! Some widespread but hidden emission source? Inversions and isotope analyses would already have identified such a source? Processes in soil or ocean, out of sight of satellites and difficult to reach by current sampling methods, that consume Gt of carbon per year? Dozens of researchers over at least four decades will have tried to attack current accounting and propose or discover new sinks, but (apparently) our understanding of basic carbon cycle processes remains valid. Do we have continuing errors - certainly! Does unexplained variability persist, to a degree that clouds our predictive skill - absolutely! But do we miss a fundamental term in the carbon cycle of today's planet? No, and everything GCP and others have published and reported for more than a decade reinforces that point. I suggested at the top several interesting topics for discussion; I contend that this paragraph and particularly this sentence need serious and substantial revision. | We have re-written the discussion and deleted this paragraph. |
| Page 42 paragraph beginning at line 21: I applaud this inclusion but this paragraphs repeats almost exactly what you say again on Page 43 in lines 17 to 20, and you say it better (shorter, clearer) there. | The discussion was reformulated and the two sentences merged in the same paragraph. |
| Page 43 lines 2 to 4: Awkward, imprecise. Suggest something such as: 'Merging these terms has limited usefulness, however, as it involves mixing direct and indirect processes and bringing in errors from other components; source, sink and uncertainty signals become more difficult to interpret.' | We rephrased the sentence building on the reviewer's suggestion. |
| Page 43 line 8: "different boundary limits among models." What does this mean? Geographic boundaries, e.g. at coastlines between land and ocean? Process boundaries, e.g. how various model treat (or do not treat) soil carbon respiration or deforestation/reforestation? Component boundaries among atmosphere, land and ocean? Specificity and clarification necessary. | Clarified. |
| Page 43, lines 9 and 10: Does this sentence refer to land fluxes only or to carbon fluxes (land and ocean) generally? | Clarified (all fluxes in the budget). |
| Page 43, line 11: Annual estimates, which continue to generate more uncertainty than longer-term means of trends, may improve with improvements in data quality and timeliness. | Modified as suggested. |
| Page 43 line 17: If you had truly large discrepancies you wouldn't construct long-term much less annual budgets. You should eliminate the word 'large'? | Deleted as suggested. |
| Page 43 lines 17 to 20: Good sentence about why you introduced model metrics. Keep this and eliminate the highly similar paragraph on page 42? | The discussion was reformulated and the two sentences merged in the same paragraph. |
| Page 44 line 13: Extraneous characters at the end of that line. | Corrected. |
| Page 44 line 29: that requires a careful compilation and synthesis of observations, statistical reports and model products. | Modified building on suggestion. |
| Page 45 line 8: change of which sink? Ocean, land, both? Also, you should refer to ocean heat content rather than ocean temperatures? | Clarified. |
| Page 45 line 9: better = improved, more frequent, more precise? | Clarified (although we do mean that with more frequent assessment we *should* improve our understanding (though admitedly it would not be only the process of updating that would improve the understanding, but all the effort around it to improve the assessment)) |
| Page 45 line 12: If your budget is correct, the sinks do more than affect, they determine? | Yes, modified as suggested. |
| Page 45 line 15: add some urgency here? 'Building this scientific understanding to meet urgent social needs depends on more frequent, robust and transparent …'? | Yes, modified as suggested. |
| Page 45 line 23, 24; here; and F Joos … Likewise line 25: measurements; and V. Kitidis … | Corrected. |
| Page 46 line 3: profile data; and Japan Met…. | Corrected. |
| Page 46 line 5: evaluations | Corrected. |
| Page 47 line 22: something wrong or missing with the citation here? Using doi would fix these problems? | Done |
| Page 48 line 5: "submitted" used twice | Done |
| Page 49 line 3: spelling and formatting errors in Collier | Done |